# Federated Learning under Covariate Shifts with Generalization Guarantees

**Ali Ramezani-Kebrya**[*]                                                                 *ali@uio.no*
*Department of Informatics, University of Oslo and Visual Intelligence Centre*

**Fanghui Liu**[*]                                                                 *fanghui.liu@epfl.ch*
*Laboratory for Information and Inference Systems (LIONS), EPFL*

**Thomas Pethick**[*]                                                                 *thomas.pethick@epfl.ch*
*Laboratory for Information and Inference Systems (LIONS), EPFL*

**Grigorios Chrysos**                                                                 *grigorios.chrysos@epfl.ch*
*Laboratory for Information and Inference Systems (LIONS), EPFL*

**Volkan Cevher**                                                                 *volkan.cevher@epfl.ch*
*Laboratory for Information and Inference Systems (LIONS), EPFL*

**Reviewed on OpenReview:** *https://openreview.net/forum?id=N7lCDaeNiS*

## Abstract

This paper addresses intra-client and inter-client covariate shifts in federated learning (FL) with a focus on the overall generalization performance. To handle covariate shifts, we formulate a new global model training paradigm and propose Federated Importance-Weighted Empirical Risk Minimization (FTW-ERM) along with improving density ratio matching methods without requiring perfect knowledge of the supremum over true ratios. We also propose the communication-efficient variant FITW-ERM with the same level of privacy guarantees as those of classical ERM in FL. We theoretically show that FTW-ERM achieves smaller generalization error than classical ERM under certain settings. Experimental results demonstrate the superiority of FTW-ERM over existing FL baselines in challenging imbalanced federated settings in terms of data distribution shifts across clients.

## 1 Introduction

Federated learning (FL) (Li et al., 2020; Kairouz et al., 2021; Wang et al., 2021) is an efficient and powerful paradigm to collaboratively train a shared machine learning model among multiple clients, such as hospitals and cellphones, without sharing local data.

Existing FL literature mainly focuses on training a model under the classical empirical risk minimization (ERM) paradigm in learning theory, with implicitly assuming that the training and test data distributions of each client are the same. However, this stylized setup overlooks the specific requirements of each client. Statistical heterogeneity is a major challenge for FL, which has been mainly studied in terms of non-identical data distributions across clients, i.e., inter-client distribution shifts (Li et al., 2020; Kairouz et al., 2021; Wang et al., 2021). Even for a single client, the distribution shift between training and test data, i.e., intra-client distribution shift, has been a major challenge for decades (Wang & Deng 2018; Kouw & Loog 2019, and references therein). For instance, scarce disease data for training and test in a local hospital can be different. To adequately address the statistical heterogeneity challenge in FL, we need to handle both intra-client and inter-client distribution shifts under stringent requirements in terms of privacy and communication costs.

---

[*]These authors contributed equally to this work. This work was partially done while the first author was at LIONS, EPFL.

We focus on the *overall generalization performance* on multiple clients by considering both intra-client and inter-client distribution shifts. There exist three major challenges to tackle this problem: 1) how to modify the classical ERM to obtain an unbiased estimate of an overall true risk minimizer under intra-client and inter-client distribution shifts; 2) how to develop an efficient density ratio estimation method under stringent privacy requirements of FL; 3) are there theoretical guarantees for the modified ERM under the improved density ratio method in FL?

We aim to address the above challenges in our new paradigm for FL. For description simplicity, in our problem setting, we focus on covariate shift, which is the *most commonly used and studied* in *theory* and *practice* in distribution shifts (Sugiyama et al., 2007; Kanamori et al., 2009; Kato & Teshima, 2021; Uehara et al., 2020; Tripuraneni et al., 2021; Zhou & Levine, 2021).[1] To be specific, for any client $k$, covariate shift assumes marginal train distributions $p_k^{\text{tr}}(\mathbf{x})$ and marginal test distributions $p_k^{\text{te}}(\mathbf{x})$ can be arbitrarily different; while the conditional distribution $p_k^{\text{tr}}(\mathbf{y}|\mathbf{x}) = p_k^{\text{te}}(\mathbf{y}|\mathbf{x}) := p(\mathbf{y}|\mathbf{x})$ remains the same, which gives rise to intra-client and inter-client *covariate shifts*. Handling covariate shift is a challenging issue, especially in federated settings (Kairouz et al., 2021).

To this end, motivated by Sugiyama et al. (2007) under the classical covariate shift setting, we propose

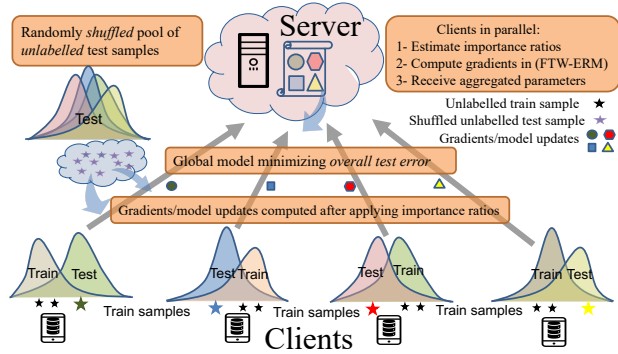

**Figure 1:** An overview of FTW-ERM. Marginal train and test distributions of clients are arbitrarily different leading to intra-client and inter-client *covariate shifts*. To control privacy leakage, the server *randomly shuffles* unlabelled test samples and broadcasts to the clients.

Federated Importance-Weighted Empirical Risk Minimization (FTW-ERM), that considers covariate shifts across multiple clients in FL. We show that the learned global model under intra/inter-client covariate shifts is still unbiased in terms of minimizing the overall true risk, i.e., FTW-ERM is *consistent* in FL. To handle covariate shifts accurately, we propose a histogram-based density ratio matching method (DRM) under both intra/inter-client distribution shifts. Our method unifies well-known DRMs in FL, which has its own interest in the distribution shift community for ratio estimation (Zadrozny, 2004; Huang et al., 2006; Sugiyama et al., 2007; Kanamori et al., 2009; Sugiyama et al., 2012; Zhang et al., 2020; Kato & Teshima, 2021). To fully eliminate *any privacy risks*, we introduce another variant of FTW-ERM, termed as Federated Independent Importance-Weighted Empirical Risk Minimization (FITW-ERM). It does not require any form of data sharing among clients and preserves the *same level of privacy* and *same communication costs* as those of baseline federated averaging (FedAvg) (McMahan et al., 2017). An overview of FTW-ERM is shown in Fig. 1.

## 1.1 Technical challenges and contributions

Learning on multiple clients in FL under covariate shifts via importance-weighted ERM is challenging due to multiple data owners with own learning objectives, multiple potential but unpredictable train/test shift scenarios, privacy, and communication costs (Kairouz et al., 2021). To be specific,
1) It is non-trivial to control privacy leakage to other clients while estimating ratios and relax the requirement to have perfect estimates of the supremum over true ratios, which is a key step for non-negative Bregman divergence (nnBD) DRM. Our work handles inter/inter-client distribution shifts in FL;
2) It is challenging to obtain per-client bounds on ratio estimation error for a general nnBD DRM with multiple clients and imperfect estimates of the supremum due to intra/inter-client couplings in ratios. Note that, even if we have access to perfect estimates of density ratios, it is still unclear whether importance-weighted ERM results in smaller excess risk compared to classical ERM. Our work gives an initial attempt by providing an affirmative answer for ridge regression;
3) While well-established benchmarks for multi-client FL have been used, they are usually designed in a way

---

[1]Our results can be extended to other typical distribution shifts, e.g., target shift (Azizzadenesheli, 2022). We provide experimental results on target shift in Section 5.

that each client's test samples are drawn uniformly from a set of classes. However, we believe this might not be the case in real-world applications and then design realistic experimental settings in our work.

To address those technical challenges, we

- Algorithmically propose an intuitive framework to minimize average *test error* in FL, design efficient mechanisms to control privacy leakage while estimating ratios (FTW-ERM) along with a privacy-preserving and communication-efficient variant (FITW-ERM), and improve nnBD DRM under FL without requiring perfect knowledge of the supremum over true ratios.

- Theoretically establish high-probability guarantees on ratio estimation error for general nnBD DRM with multiple clients under imperfect estimates of the supremum, which unifies a number of DRMs, and show benefits of importance weighting in terms of excess risk decoupled from density ratio estimation through bias-variance decomposition.

- Experimentally demonstrate more than 16% overall test accuracy improvement over existing FL baselines when training ResNet-18 (He et al., 2016) on CIFAR10 (Krizhevsky) in challenging imbalanced federated settings in terms of data distribution shifts across clients.

In conclusion, we expand the concept and application scope of FL to a general setting under intra/inter-client covariate shifts, provide an in-depth theoretical understanding of learning with FTW-ERM via a general DRM, and experimentally validate the utility of the proposed framework. We hope that our work opens the door to a new FL paradigm.

## 1.2 Related work

In this section, we overview a summary of related work. See Appendix B for complete discussion.

**Federated learning.** The current FL literature largely focuses on minimizing the empirical risk, under the same training/test data distribution assumption over each client (Li et al., 2020; Kairouz et al., 2021; Wang et al., 2021). Statistical heterogeneity across clients in training-time is handled using heuristics-based personalization methods that typically do not have a statistical learning theoretical support (Smith et al., 2017; Khodak et al., 2019; Li et al., 2021b). In contrast, we focus on learning under both intra-client and inter-client covariate shifts. Communication-efficient, robust, and secure aggregations can be viewed as complementary technologies, which can be used along with FTW-ERM to improve FTW-ERM's scalability and security while addressing overall generalization. Our theory focuses on cross-silo FL where a number of trustworthy and available clients under intra/inter-client covariate shifts learn a global model collaboratively, and our experiments extend to scenarios with 100 clients and client partial participation.

Wang et al. (2020) tackle update drifts considering variations in the number of local updates performed by each client in each communication round and focuses on minimizing the empirical risk under the same training/test data distribution assumption over each client. Li et al. (2021c) propose FedBD to tackle inter-client feature shift by updating Batch Normalization (BN) layers locally and updating non-BN layers using FedAvg. They consider both inter-client covariate shift and concept shift but under the same training/test data distribution assumption over each client. de Luca et al. (2022) consider Federated Domain Generalization and propose data augmentation to learn a model that generalizes to in-domain datasets of the participating clients and an out-of-domain dataset of a non-participating client. They propose to use FedAvg after proper data augmentation, which is orthogonal to the algorithmic design, e.g., our work. Gupta et al. (2022) propose FL Games, a game-theoretic framework for learning causal features that are invariant across clients by using ensembles over clients' historical actions and increasing the local computation under the same training/test data distribution assumption over each client. Different from these work, we focus on learning and overall generalization performance, i.e., minimizing the average test error over all clients, under both intra/inter-client covariate shifts.

**Importance-weighted ERM and density ratio matching.** Shimodaira (2000) introduce covariate shift where the input train and test distributions are different while the conditional distribution of the output

variable given the input variable remains unchanged. Importance-weighted ERM is widely used to improve generalization performance under covariate shift (Zadrozny, 2004; Sugiyama & Müller, 2005; Huang et al., 2006; Sugiyama et al., 2007; Kanamori et al., 2009; Sugiyama et al., 2012; Fang et al., 2020; Zhang et al., 2020; Kato & Teshima, 2021). Sugiyama et al. (2012) propose a BD-based DRM, which unifies various DRMs. Kato & Teshima (2021) propose an nnBD-based DRM when using deep neural networks for density ratio estimation. Our work largely differs from Kato & Teshima (2021) in our *problem setting* that allows multiple clients, *algorithm design* to estimate different ratios across clients and relax the requirement to have perfect estimates of the supremum over true ratios while controlling privacy leakage, and *theoretical analyses* to show the benefit of importance weighting in generalization.

**Domain adaptation.** Distribution shifts between a source and a target domain have been a prominent problem in machine learning for several decades (Wang & Deng, 2018; Kouw & Loog, 2019). The premise behind such shifts is that data is frequently biased, and this results in distribution shifts that can be estimated by assuming some (unlabelled) knowledge of the target distribution. The following two categories of domain adaptation methods are most closely related to our work: a) sample-based, and b) feature-based methods. In feature-based methods, the goal is to find a transformation that maps the source samples to target samples (Ganin et al., 2016; Bousmalis et al., 2017; Das & Lee, 2018; Damodaran et al., 2018). Sample-based methods aim at minimizing the target risk through data in the source domain. Importance weighting is often used in sample-based methods (Shimodaira, 2000; Jiang & Zhai, 2007; Baktashmotlagh et al., 2014). However, the focus on domain adaptation has been mainly to adapt to a single target distribution, not the overall generalization performance on multiple clients, which is addressed in this paper.

**Statistical generalization and excess risk bounds.** Understanding generalization performance of learning algorithms is one essential topic in modern machine learning. Typical techniques to establish generalization guarantees include uniform convergence by Rademacher complexity (Bartlett, 1998), and its variants (Bartlett et al., 2005), bias-variance decomposition (Geman et al., 1992; Adlam & Pennington, 2020), PAC-Bayes (McAllester, 1999), and stability-based analysis (Bousquet & Elisseeff, 2002; Shalev-Shwartz et al., 2010). Our work employs the first two techniques to analyze our density ratio estimation method in a federated setting and establish generalization guarantees for FTW-ERM, respectively. Rademacher complexity has been used in FL to obtain theoretical guarantees on the centralized model (Mohri et al., 2019) and personalized model (Mansour et al., 2020). These work are different from our setting where we consider multiple test distributions under different training/test data distributions for clients and focus on the overall test error. Bias-variance decomposition is typically studied in two settings, i.e., the fixed and random design setting, which is categorized by whether the (training) data are fixed or random. This technique has been extensively applied in least squares (Hsu et al., 2012; Dieuleveut et al., 2017), analysis of SGD (Jain et al., 2018; Zou et al., 2021), and double descent (Adlam & Pennington, 2020).

**Notation:** We use $\mathbb{E}[\cdot]$ to denote the expectation and $\|\cdot\|$ to represent the Euclidean norm of a vector. We use lower-case bold font to denote vectors. Sets and scalars are represented by calligraphic and standard fonts, respectively. We use $[n]$ to denote $\{1, \ldots, n\}$ for an integer $n$. We use $\lesssim$ to ignore terms up to constants and logarithmic factors.

## 2 Covariate shift and FTW-ERM for FL

We first provide the problem setting under intra/inter client covariate shifts, and then describe the proposed FTW-ERM as an unbiased estimate in terms of minimizing the overall true risk[2].

### 2.1 Problem setting

Let $\mathcal{X} \subseteq \mathbb{R}^{d_\mathbf{x}}$ be a compact metric space, $\mathcal{Y} \subseteq \mathbb{R}^{d_\mathbf{y}}$, and $K$ be the number of clients in an FL setting. Let $\mathcal{S}_k = \{(\mathbf{x}^{\text{tr}}_{k,i}, \mathbf{y}^{\text{tr}}_{k,i})\}_{i=1}^{n^{\text{tr}}_k}$ denote the training set of client $k$ with $n^{\text{tr}}_k$ samples drawn i.i.d. from an unknown probability distribution $p^{\text{tr}}_k$ on $\mathcal{X} \times \mathcal{Y}$.[3] The test data of client $k$, is drawn from another unknown probability

---

[2]Notations are provided in Appendix A.

[3]For notational simplicity, we use the same notation for probability distributions and density functions.

distribution $p_k^{\text{te}}$ on $\mathcal{X} \times \mathcal{Y}$. Under the covariate shift setting (Sugiyama et al., 2007; Kanamori et al., 2009; Kato & Teshima, 2021; Uehara et al., 2020; Tripuraneni et al., 2021; Zhou & Levine, 2021), the conditional distribution $p_k^{\text{tr}}(\mathbf{y}|\mathbf{x}) = p_k^{\text{te}}(\mathbf{y}|\mathbf{x}) := p(\mathbf{y}|\mathbf{x})$ is assumed to be the same for all $k$, while $p_k^{\text{tr}}(\mathbf{x})$ and $p_k^{\text{te}}(\mathbf{x})$ can be arbitrarily different, which gives rise to intra-client and inter-client covariate shifts. We consider supervised learning where the goal is to find a hypothesis $h_{\mathbf{w}} : \mathcal{X} \to \mathcal{Y}$, parameterized by $\mathbf{w} \in \mathbb{R}^d$ e.g., weights and biases of a neural network, such that $h_{\mathbf{w}}(\mathbf{x})$ (for short $h(\mathbf{x})$) is a good approximation of the label $\mathbf{y} \in \mathcal{Y}$ corresponding to a new sample $\mathbf{x} \in \mathcal{X}$. Let $\ell : \mathcal{X} \times \mathcal{Y} \to \mathbb{R}_+$ denote a loss function. In our FL setting, the true (expected) risk of client $k$ is given by $R_k(h_{\mathbf{w}}) = \mathbb{E}_{(\mathbf{x},\mathbf{y}) \sim p_k^{\text{te}}(\mathbf{x},\mathbf{y})}[\ell(h_{\mathbf{w}}(\mathbf{x}), \mathbf{y})]$.

## 2.2 FTW-ERM for FL under covariate shift

We assume that $p_k^{\text{tr}}(\mathbf{x}^{\text{tr}}) > 0$ for $k \in [K]$ and all $\mathbf{x}^{\text{tr}} \in \mathcal{X}^{\text{tr}} \subseteq \mathcal{X}$ with $\mathcal{X}^{\text{te}} \subseteq \mathcal{X}^{\text{tr}}$, i.e., we need a common data domain with strictly positive train density, which is a common assumption (Kanamori et al., 2009; Kato & Teshima, 2021). For a scenario with $K$ clients, we first focus on minimizing $R_l$ ($l \in [K]$) under intra/inter-client covariate shifts, i.e., $p_k^{\text{tr}}(\mathbf{x}) \neq p_l^{\text{te}}(\mathbf{x})$ for all $k$. We then formulate FTW-ERM to minimize the average test error over $K$ clients under covariate shifts by optimizing a global model under our FL setting.

**FTW-ERM for one client.** Under $p_k^{\text{tr}}(\mathbf{x}) \neq p_l^{\text{te}}(\mathbf{x}) \ \forall k$, FTW-ERM focusing on minimizing $R_l$ is given by:

$$\min_{\mathbf{w} \in \mathbb{R}^d} \sum_{k=1}^{K} \frac{1}{n_k^{\text{tr}}} \sum_{i=1}^{n_k^{\text{tr}}} \frac{p_l^{\text{te}}(\mathbf{x}_{k,i}^{\text{tr}})}{p_k^{\text{tr}}(\mathbf{x}_{k,i}^{\text{tr}})} \ell(h_{\mathbf{w}}(\mathbf{x}_{k,i}^{\text{tr}}), \mathbf{y}_{k,i}^{\text{tr}}) . \tag{2.1}$$

In Appendix C, we elaborate on four special cases of the above scenario, i.e., $p_k^{\text{tr}}(\mathbf{x}) \neq p_l^{\text{te}}(\mathbf{x}) \ \forall k$, focusing on one client under various covariate shifts and formulate their FTW-ERM's.

**Proposition 1.** *Let $l \in [K]$. FTW-ERM in Eq. (2.1) is consistent. i.e., the learned function converges in probability to the optimal function in terms of minimizing $R_l$.*

See Appendix C for the proof. Proposition 1 implies that, under intra/inter-client covariate shifts, FTW-ERM outputs an unbiased estimate of a true risk minimizer of client $l$. In Appendix C.1, we show usefulness of importance weighting under no intra-client covariate shifts but inter-client covariate shifts, which is a special and important case of our setting.

Building on Eq. (2.1) that aims to minimize $R_l$, we now formulate FTW-ERM to minimize the average test error over all clients and explain its costs and benefits for federated settings.

**FTW-ERM for $K$ clients.** Let $\mathbf{w}$ be the global model. For $K$ clients under intra/inter-clinet covariate shifts, FTW-ERM minimizes the average test error over all clients and is formulated as:

$$\min_{\mathbf{w} \in \mathbb{R}^d} F(\mathbf{w}) := \sum_{k=1}^{K} F_k(\mathbf{w}) \tag{FTW-ERM}$$

where

$$F_k(\mathbf{w}) = \frac{1}{n_k^{\text{tr}}} \sum_{i=1}^{n_k^{\text{tr}}} \frac{\sum_{l=1}^{K} p_l^{\text{te}}(\mathbf{x}_{k,i}^{\text{tr}})}{p_k^{\text{tr}}(\mathbf{x}_{k,i}^{\text{tr}})} \ell(h_{\mathbf{w}}(\mathbf{x}_{k,i}^{\text{tr}}), \mathbf{y}_{k,i}^{\text{tr}}) . \tag{2.2}$$

Each client requires an estimate of a ratio in the form of sum of test densities over own train density, e.g., $\sum_{l=1}^{K} p_l^{\text{te}}/p_k^{\text{tr}}$ for client $k$. We emphasize that $F_k(\mathbf{w})$ should *not* be viewed as the local loss function of client $k$. Our formulation FTW-ERM is meant to minimize the overall test error over all clients given intra/inter-client covariate shifts. To solve FTW-ERM, we employ the stochastic gradient descent (SGD) algorithm for $T$ iterations starting from an initial parameter $\mathbf{w}_0$: $\mathbf{w}_{t+1} = \mathbf{w}_t - \eta_t \sum_{k=1}^{K} \mathbf{g}_k(\mathbf{w}_t)$ where $\eta_t > 0$ is the step size, $\mathbf{g}_k(\mathbf{w}_t)$ is an unbiased estimate of $\nabla_{\mathbf{w}} F_k(\mathbf{w}_t)$, and $\mathbf{w}_T$ is the output.

Under no covariate shift, both FTW-ERM with true ratios and classical ERM result in the same solution, which is a minimizer of the overall *empirical risk*. The main difference happens under intra-client and inter-client covariate shifts. In those challenging settings, FTW-ERM's solution is an unbiased estimate of a minimizer of the overall *true risk*, while the solution of ERM minimizes the overall *empirical risk*.

## 2.3 Privacy, communication, and computation in FL

Privacy and communication efficiency are major concerns in FL (Kairouz et al., 2021). We elaborate on them and introduce another variant of FTW-ERM with the same guarantees and costs as FedAvg.

**Communication/computational costs and security benefits.** Compared to classical ERM, the communication/computational overhead of FTW-ERM is negligible.[4] To solve FTW-ERM, client $k$ should compute an unbiased estimate of the weighted gradient $\nabla_{\mathbf{w}} F_k(\mathbf{w}_t)$, which requires a single backward pass at a single parameter $\mathbf{w} = \mathbf{w}_t$. Hence, given the ratios, there is no extra computational/communication overhead compared to classical ERM. Clients compute the ratios in parallel. In Appendix E, we provide a concrete example and show that the number of communication bits needed during training in standard FL is usually many orders of magnitudes larger than the size of samples shared for estimating the ratios. To further reduce communication costs of density ratio estimation and gradient aggregation, compression methods such as quantization, sparsification, and local updating rules, can be used along with FTW-ERM on the fly. More importantly, due to *importance weighting*, $\mathbf{g}_k(\mathbf{w})$ can be arbitrarily different from an unbiased stochastic gradient of classical ERM for client $k$, *i.e.*, $\frac{1}{n_k^{\mathrm{tr}}} \sum_{i=1}^{n_k^{\mathrm{tr}}} \nabla_{\mathbf{w}} \ell(h_{\mathbf{w}}(\mathbf{x}_{k,i}^{\mathrm{tr}}), \mathbf{y}_{k,i}^{\mathrm{tr}})$. The formulation FTW-ERM makes it impossible for an adversary to apply gradient inversion attack and obtain private training data of clients (Zhu et al., 2019). In particular, the attacker cannot formulate the correct optimization problem and reconstruct client $k$'s data unless the attacker has a perfect knowledge of the ratio $r_k(\mathbf{x}) = \sum_{l=1}^{K} p_l^{\mathrm{te}}(\mathbf{x})/p_k^{\mathrm{tr}}(\mathbf{x})$ that client $k$ applies when computing (stochastic) gradients in Eq. (2.2).

**Privacy.** Given $\{r_k(\mathbf{x})\}_{k=1}^{K}$, FTW-ERM efficiently minimizes the overall test error over all clients in a privacy-preserving manner. To estimate those ratios, if clients can tolerate some level of privacy leakage, clients send unlabelled samples $\mathbf{x}_{l,j}^{\mathrm{te}}$ for $l \in [K]$ and $j \in [n^{\mathrm{te}}]$ from their test distributions. To control privacy leakage to other clients, we propose that the server *randomly shuffles* these unlabelled samples before broadcasting to clients. In Appendix Q, we discuss an alternative method instead of sending original unlabelled samples and discuss its limitations.

To fully *eliminate any privacy risks* compared to classical ERM, clients may opt to minimize the following surrogate objective, which we name Federated Independent Importance-Weighted Empirical Risk Minimization (FITW-ERM):

$$\min_{\mathbf{w} \in \mathbb{R}^d} \tilde{F}(\mathbf{w}) := \sum_{k=1}^{K} \frac{1}{n_k^{\mathrm{tr}}} \sum_{i=1}^{n_k^{\mathrm{tr}}} \frac{p_k^{\mathrm{te}}(\mathbf{x}_{k,i}^{\mathrm{tr}})}{p_k^{\mathrm{tr}}(\mathbf{x}_{k,i}^{\mathrm{tr}})} \ell(h_{\mathbf{w}}(\mathbf{x}_{k,i}^{\mathrm{tr}}), \mathbf{y}_{k,i}^{\mathrm{tr}}). \tag{FITW-ERM}$$

The formulation FITW-ERM preserves the *same level of privacy* and *same communication costs* as those of classical ERM, e.g., FedAvg.[5] Ratios for FITW-ERM are obtained using local data and clients share only gradient information without sharing any data. Clients estimate and apply ratios using their own local data, which essentially modifies their local loss function. This modified local loss for FITW-ERM can be directly substituted in any formal differential privacy results for ERM such as those in (Kairouz et al., 2021). However, to exploit the entire data distributed among all clients and achieve the optimal global model in terms of overall test error, clients need to compromise some level of privacy and share unlabelled test samples with the server. Hence, in this paper, we focus on the original objective in FTW-ERM.

## 3 Ratio estimation for FL under covariate shift

To solve FTW-ERM, client $k$ should have access to an accurate estimate of this ratio

$$r_k(\mathbf{x}) = \frac{\sum_{l=1}^{K} p_l^{\mathrm{te}}(\mathbf{x})}{p_k^{\mathrm{tr}}(\mathbf{x})}. \tag{3.1}$$

---

[4]The analyses of computational/communication overheads are provided in Appendices P and E, respectively.
[5]By estimating the ratios locally and absorbing into local losses, FITW-ERM can be viewed as a variant of classical ERM.

Ratio estimation is a key step for importance weighting (Sugiyama et al., 2007; 2012). The discrepancy between the true ratio $r_k^*$ for client $k$ in Eq. (3.1) and the estimated one $r_k$ using our ratio model can be measured by $\mathbb{E}_{p_k^{\mathrm{tr}}}[\mathrm{BD}_f(r_k^*(\mathbf{x}) \| r_k(\mathbf{x}))]$ where the Bregman divergence (BD) associated with a strictly convex $f$ leads to BD-based DRMs (Kato & Teshima, 2021; Kiryo et al., 2017):

**Definition 1** (Bregman 1967). *Let $\mathcal{B}_f \subset [0, \infty)$ be bounded and $f : \mathcal{B}_f \to \mathbb{R}$ be a strictly convex function with bounded gradient. The BD associated with $f$ from $\tilde{z}$ to $z$ is given by $\mathrm{BD}_f(\tilde{z} \| z) = f(\tilde{z}) - f(z) - \nabla f(z)(\tilde{z} - z)$.*

Note that $\mathrm{BD}_f(\tilde{z} \| z)$ is a convex function w.r.t. $\tilde{z}$; however, it is not necessarily convex w.r.t. $z$. The bounded $\mathcal{B}_f$ is a standard assumption (Kato & Teshima, 2021), which holds in our problem since the density ratios that are inputs of BD are bounded following the assumption in Section 2.2. We estimate the supremum over true ratios in Section 3.2 and provide examples of $f$ commonly used for BD-based methods in Table 4 of Appendix A. Motivated by Kato & Teshima (2021); Kiryo et al. (2017), we propose a new histogram-based DRM (HDRM) for FL with multiple clients. HDRM overcomes the over-fitting issue (Kiryo et al., 2017; Kato & Teshima, 2021) while providing an estimate for the upper bound $\overline{r}_k = \sup_{\mathbf{x} \in \mathcal{X}^{\mathrm{tr}}} r_k^*(\mathbf{x})$, which is a key step for non-negative BD (nnBD) DRM. We now extend nnBD DRM to FL settings.

## 3.1 Extension of nnBD DRM to FL

Let $\mathcal{H}_r \subset \{r : \mathcal{X} \to \mathcal{B}_f\}$ denote a hypothesis class for our ratios $r_k$, e.g., neural networks with a given architecture. Our goal is to estimate $r_k$ by minimizing the discrepancy $\mathbb{E}_{p_k^{\mathrm{tr}}}[\mathrm{BD}_f(r_k^*(\mathbf{x}) \| r_k(\mathbf{x}))]$, which leads to BD-based DRM for FL and is formulated in Appendix D.2. Let $\{\mathbf{x}_{k,i}^{\mathrm{tr}}\}_{i=1}^{n_k^{\mathrm{tr}}}$ and $\{\mathbf{x}_{l,j}^{\mathrm{te}}\}_{j=1}^{n^{\mathrm{te}}}$ denote unlabelled samples drawn i.i.d. from distributions $p_k^{\mathrm{tr}}$ and $p_l^{\mathrm{te}}$, respectively, for $l \in [K]$. Standard BD-based DRM is shown to suffer from an over-fitting issue where $-\frac{1}{n^{\mathrm{te}}} \sum_{j=1}^{n^{\mathrm{te}}} \nabla f(r_k(\mathbf{x}_{l,j}^{\mathrm{te}}))$ diverges if there is no lower bound on this term (Kiryo et al., 2017; Kato & Teshima, 2021). To resolve this issue in FL, we consider non-negative BD (nnBD) DRM for client $k$, *i.e.*, $\min_{r_k \in \mathcal{H}_r} \hat{\mathcal{E}}_f^+(r_k)$ where

$$\hat{\mathcal{E}}_f^+(r_k) = \mathrm{ReLU}\Big(\frac{1}{n_k^{\mathrm{tr}}} \sum_{i=1}^{n_k^{\mathrm{tr}}} \ell_1(r_k(\mathbf{x}_{k,i}^{\mathrm{tr}})) - \frac{C_k}{n^{\mathrm{te}}} \sum_{j=1}^{n^{\mathrm{te}}} \sum_{l=1}^{K} \ell_1(r_k(\mathbf{x}_{l,j}^{\mathrm{te}}))\Big) + \frac{1}{n^{\mathrm{te}}} \sum_{j=1}^{n^{\mathrm{te}}} \sum_{l=1}^{K} \ell_2(r_k(\mathbf{x}_{l,j}^{\mathrm{te}})), \tag{3.2}$$

$\mathrm{ReLU}(z) = \max\{0, z\}$, $0 < C_k < \frac{1}{\overline{r}_k}$, $\overline{r}_k = \sup_{\mathbf{x} \in \mathcal{X}^{\mathrm{tr}}} r_k^*(\mathbf{x})$, $\ell_1(z) = \nabla f(z)z - f(z)$, and $\ell_2(z) = C(\nabla f(z)z - f(z)) - \nabla f(z)$. Intuitively, ReLU is used for non-negativity and $0 < C_k < \frac{1}{\overline{r}_k}$ acts as a regularization parameter. Substituting different $f$'s into Eq. (3.2) leads to different variants of nnBD, which covers previous work (Basu et al., 1998; Hastie et al., 2001; Gretton et al., 2009; Nguyen et al., 2010; Kato et al., 2019). We provide explicit expressions of those variants for client $k$ in Appendix H. In this work, we focus on $f(z) = \frac{(z-1)^2}{2}$ leading to the well-known least-squares importance fitting (LSIF) variant of nnBD for client $k$.

## 3.2 Estimation of the upper bound $\overline{r}_k$

Estimating $\overline{r}_k = \sup_{\mathbf{x} \in \mathcal{X}^{\mathrm{tr}}} r_k^*(\mathbf{x})$ is a key step for nnBD DRM. For a single train and test distribution, it is shown that overestimating $\overline{r}$ leads to significant performance degradation (Kato & Teshima, 2021, Section 5). Kato & Teshima (2021) considered $0 < C < \frac{1}{\overline{r}}$ as a hyper-parameter, which can be tuned. However, obtaining an efficient estimate of $\overline{r}_k$ is desirable, in particular when training a deep model. Here we propose a histogram-based method for estimation of $\overline{r}_k$.

Let $\mathcal{B} \subset \mathcal{X}^{\mathrm{tr}}$, and assume $p_k^{\mathrm{tr}}$ and $p_l^{\mathrm{te}}$ are continuous for $l \in [K]$. Since $\mathcal{B}$ is connected and Lebesgue-measurable with finite measure, by applying intermediate value theorem (Russ, 1980), there exist $\tilde{\mathbf{x}}^{\mathrm{tr}}$ and $\hat{\mathbf{x}}^{\mathrm{te}}$ such that $\Pr\{X_k^{\mathrm{tr}} \in \mathcal{B}\} = p_k^{\mathrm{tr}}(\tilde{\mathbf{x}}^{\mathrm{tr}})\mathrm{Vol}(\mathcal{B})$ and $\sum_{l=1}^{K} \Pr\{X_l^{\mathrm{te}} \in \mathcal{B}\} = \sum_{l=1}^{K} p_l^{\mathrm{te}}(\hat{\mathbf{x}}^{\mathrm{te}})\mathrm{Vol}(\mathcal{B})$ where $\mathrm{Vol}(\mathcal{B}) = \int_{\mathbf{x} \in \mathcal{B}} \mathrm{d}\mathbf{x}$. We note that $\sup_{\mathbf{x} \in \mathcal{B}} r_k^*(\mathbf{x}) \leq \frac{\sup_{\mathbf{x} \in \mathcal{B}} \sum_{l=1}^{K} p_l^{\mathrm{te}}(\mathbf{x})}{\inf_{\mathbf{x} \in \mathcal{B}} p_k^{\mathrm{tr}}(\mathbf{x})}$ and $\frac{\sum_{l=1}^{K} p_l^{\mathrm{te}}(\hat{x}^{\mathrm{te}})}{p_k^{\mathrm{tr}}(\tilde{x}^{\mathrm{tr}})} \leq \frac{\sup_{\mathbf{x} \in \mathcal{B}} \sum_{l=1}^{K} p_l^{\mathrm{te}}(\mathbf{x})}{\inf_{\mathbf{x} \in \mathcal{B}} p_k^{\mathrm{tr}}(\mathbf{x})}$. To estimate $\overline{r}_k$, we first partition $\mathcal{X}^{\mathrm{tr}}$ into $M$ bins where for each bin $\mathcal{B}_m$, if there exists some $\mathbf{x}_{k,i}^{\mathrm{tr}} \in \mathcal{B}_m$, then we define

$$\tilde{r}_{k,m} := \frac{\sum_{l=1}^{K} \Pr\{X_l^{\mathrm{te}} \in \mathcal{B}_m\}}{\Pr\{X_k^{\mathrm{tr}} \in \mathcal{B}_m\}} \simeq \frac{\frac{1}{n^{\mathrm{te}}} \sum_{j=1}^{n^{\mathrm{te}}} \sum_{l=1}^{K} \mathbb{1}(\mathbf{x}_{l,j}^{\mathrm{te}} \in \mathcal{B}_m)}{\frac{1}{n_k^{\mathrm{tr}}} \sum_{i=1}^{n_k^{\mathrm{tr}}} \mathbb{1}(\mathbf{x}_{k,i}^{\mathrm{tr}} \in \mathcal{B}_m)} \text{ for } m \in [M]. \text{ Otherwise, } \tilde{r}_{k,m} = 0. \text{ Finally, we propose}$$

**Input:** Samples $\{\{\mathbf{x}_{k,i}^{\mathrm{tr}}\}_{i=1}^{n_k^{\mathrm{tr}}}\}_{k=1}^{K}$, $\{\{\mathbf{x}_{l,j}^{\mathrm{te}}\}_{j=1}^{n^{\mathrm{te}}}\}_{l=1}^{K}$, learning rate $\alpha$, regularization $\Lambda(r)$ and regularization coefficient $\lambda$.

**Output:** Ratio model parameters $\{\boldsymbol{\theta}_{r_k}\}_{k=1}^{K}$.

**1** **for** $k = 1$ **to** $K$ *(in parallel)* **do**

**2** $\quad$ Send $n^{\mathrm{te}}$ samples to the server ;

**3** Server randomly shuffles and broadcasts samples $\{\{\mathbf{x}_{l,j}^{\mathrm{te}}\}_{j=1}^{n^{\mathrm{te}}}\}_{l=1}^{K}$ to clients ;

**4** **for** $k = 1$ **to** $K$ *(in parallel)* **do**

**5** $\quad$ Create $M$ bins and compute $\tilde{r}_{k,m} = \frac{1/n^{\mathrm{te}} \sum_{j=1}^{n^{\mathrm{te}}} \sum_{l=1}^{K} \mathbb{1}(\mathbf{x}_{l,j}^{\mathrm{te}} \in \mathcal{B}_m)}{1/n_k^{\mathrm{tr}} \sum_{i=1}^{n_k^{\mathrm{tr}}} \mathbb{1}(\mathbf{x}_{k,i}^{\mathrm{tr}} \in \mathcal{B}_m)}$ ;

**6** $\quad$ Estimate $C_k = \frac{1}{\max\{\tilde{r}_{k,1},\ldots,\tilde{r}_{k,M}\}}$ ;

**7** **for** $t = 1$ **to** $T$ **do**

**8** $\quad$ **for** $k = 1$ **to** $K$ *(in parallel)* **do**

**9** $\quad\quad$ **for** $n = 1$ **to** $N_k$ **do**

**10** $\quad\quad\quad$ **if** $\frac{1}{B_k^{\mathrm{tr}}} \sum_{i=1}^{B_k^{\mathrm{tr}}} \ell_1(r_k(\mathbf{x}_{k,n,i}^{\mathrm{tr}})) - \frac{KC_k}{B_k^{\mathrm{te}}} \sum_{j=1}^{B_k^{\mathrm{te}}} \ell_1(r_k(\mathbf{x}_{k,n,j}^{\mathrm{te}})) \geq 0$ **then**

**11** $\quad\quad\quad\quad$ $\mathbf{g}_k = -\nabla_{\boldsymbol{\theta}_r}\Big(\frac{1}{B_k^{\mathrm{tr}}} \sum_{i=1}^{B_k^{\mathrm{tr}}} \ell_1(r_k(\mathbf{x}_{k,n,i}^{\mathrm{tr}})) - \frac{KC_k}{B_k^{\mathrm{te}}} \sum_{j=1}^{B_k^{\mathrm{te}}} \ell_1(r_k(\mathbf{x}_{k,n,j}^{\mathrm{te}})) + \frac{K}{B_k^{\mathrm{te}}} \sum_{j=1}^{B_k^{\mathrm{te}}} \ell_2(r_k(\mathbf{x}_{k,n,j}^{\mathrm{te}})) +$
$\quad\quad\quad\quad \frac{\lambda}{2}\Lambda(r_k)\Big)$ ;

**12** $\quad\quad\quad$ **else**

**13** $\quad\quad\quad\quad$ $\mathbf{g}_k = \nabla_{\boldsymbol{\theta}_r}\Big(\frac{1}{B_k^{\mathrm{tr}}} \sum_{i=1}^{B_k^{\mathrm{tr}}} \ell_1(r_k(\mathbf{x}_{k,n,i}^{\mathrm{tr}})) - \frac{KC_k}{B_k^{\mathrm{te}}} \sum_{j=1}^{B_k^{\mathrm{te}}} \ell_1(r_k(\mathbf{x}_{k,n,j}^{\mathrm{te}})) + \frac{\lambda}{2}\Lambda(r_k)\Big)$ ;

**14** $\quad\quad\quad$ Update ratio model parameters $\boldsymbol{\theta}_{r_k} = \boldsymbol{\theta}_{r_k} + \alpha\mathbf{g}_k$ ;

**Algorithm 1:** Histogram-based density ratio matching. Loops are executed in parallel on each client.

to use $C_k = \frac{1}{\tilde{r}_k}$ where $\tilde{r}_k = \max\{\tilde{r}_{k,1},\ldots,\tilde{r}_{k,M}\}$. Convergence of $\tilde{r}_k$ to $\bar{r}_k$ is established in Appendix G. Furthermore, for high-dimensional data, an efficient implementation of HDRM using $k$-means clustering is provided in Appendix G. We note that the number of elementary operations for computing (3.2) and its gradients per step dominates that of running the efficient $k$-means clustering to estimate $C_k$.

In HDRM, $K$ clients estimate their ratios in parallel. To be specific, clients first share unlabelled test samples with the server. The server returns the randomly shuffled pool of samples to all clients. Then clients find $C_k$'s in parallel. Given $C_k$'s, clients estimate their corresponding ratios in parallel. To handle high-dimensional data samples and deep ratio estimation models, we adopt a variant of SGD. For client $k$, we divide unlabelled samples $\{\mathbf{x}_{k,i}^{\mathrm{tr}}\}_{i=1}^{n_k^{\mathrm{tr}}}$ and $\{\mathbf{x}_{l,j}^{\mathrm{te}}\}_{j=1}^{n^{\mathrm{te}}}$ for $l \in [K]$ into $N_k$ batches $\{\mathbf{x}_{k,n,i}^{\mathrm{tr}}\}_{i=1}^{B_k^{\mathrm{tr}}}$ and $\{\mathbf{x}_{k,n,j}^{\mathrm{te}}\}_{j=1}^{B_k^{\mathrm{te}}}$ for $n \in [N_k]$. Client $k$ first computes $\frac{1}{B_k^{\mathrm{tr}}} \sum_{i=1}^{B_k^{\mathrm{tr}}} \ell_1(r_k(\mathbf{x}_{k,n,i}^{\mathrm{tr}})) - \frac{KC_k}{B_k^{\mathrm{te}}} \sum_{j=1}^{B_k^{\mathrm{te}}} \ell_1(r_k(\mathbf{x}_{k,n,j}^{\mathrm{te}}))$. If it becomes negative, then we apply a gradient ascent step to increase this term. We may also opt to apply 1-norm or 2-norm regularizations. The details of the HDRM algorithm are shown in Algorithm 1.

## 4 Theoretical guarantees

To address learning on multiple clients in FL, it is essential to obtain per-client generalization bounds for a general nnBD DRM with imperfect estimates of $\bar{r}_k$'s. Even if we have access to perfect estimates of density ratios, it is still unclear the usefulness of importance weighting. In this section, we firstly study the high-probability guarantees on ratio estimation error of nnBD DRM under imperfect estimate of $\bar{r}_k$ in terms of BD risk. We then show the benefit of importance weighting in term of excess risk through a refined bias-variance decomposition on a ridge regression problem. Theorem 1, Lemma 1, Theorem 2 are proved in Appendix I, Appendix L, and Appendix M, respectively.

### 4.1 Ratio estimation error in terms of BD risk

We establish a high-probability bound on the ratio estimation error of nnBD DRM with an arbitrary $f$ for client $k$ in terms of BD risk given by

$$\mathcal{E}_f(r_k) = \tilde{\mathbb{E}}_k(\mathbf{x})[\ell_1(r_k(\mathbf{x}))] + \sum_{l=1}^{K} \mathbb{E}_{p_l^{\mathrm{te}}}[\ell_2(r_k(\mathbf{x}))] \tag{4.1}$$

where $\tilde{\mathbb{E}}_k := \mathbb{E}_{p_k^{\mathrm{tr}}} - C_k \sum_{l=1}^{K} \mathbb{E}_{p_l^{\mathrm{te}}}$. Our bound for client $k$ depends on the Rademacher complexity (Koltchinskii, 2001) of the hypothesis class for our density ratio model $\mathcal{H}_r \subset \{r : \mathcal{X} \to \mathcal{B}_f\}$ w.r.t. client $k$ train distribution $p_k^{\mathrm{tr}}$ and all client's test distributions $p_l^{\mathrm{te}}$ for $l \in [K]$. Let $R_n^p(\mathcal{H})$ denotes the Rademacher complexity of function class $\mathcal{H}$ w.r.t. distribution $p$, formally defined as follows:

**Definition 2.** *Let $n \in \mathbb{Z}_+$ and $p$ be a distribution, $\mathcal{S} = \{\mathbf{x}_1, \ldots, \mathbf{x}_n\}$ be i.i.d. random variables drawn from $p$, and $\mathcal{H}$ be a function class. The Rademacher complexity of $\mathcal{H}$ w.r.t. $p$ is given by:*

$$R_n^p(\mathcal{H}) = \mathbb{E}_{\mathcal{S}} \mathbb{E}_{\boldsymbol{\sigma}} \left[ \sup_{r \in \mathcal{H}} \left| \frac{1}{n} \sum_{i=1}^{n} \sigma_i r(\mathbf{x}_i) \right| \right]$$

*where $\{\sigma_i\}_{i=1}^{n}$ are Rademacher variables uniformly chosen from $\{-1, 1\}$.*

We first make the following assumptions on $\ell_1(z) = \nabla f(z)z - f(z)$ and $\ell_2(z) = C(\nabla f(z)z - f(z)) - \nabla f(z)$.

**Assumption 1** (Basic assumptions on $\ell_1$ and $\ell_2$). We assume 1) $\sup_{z \in \mathcal{B}_f} \max_{i \in \{1,2\}} |\ell_i(z)| < \infty$; 2) $\ell_1$ is $L_1$-Lipschitz and $\ell_2$ is $L_2$-Lipschitz on $\mathcal{X}$; 3) $\inf_{r \in \mathcal{H}_r} \tilde{\mathbb{E}}_k[\ell_1(r_k(\mathbf{x}))] > 0$ for $k \in [K]$.

The first two assumptions are satisfied if $\inf\{z | z \in \mathcal{B}_f\} > 0$ for commonly used loss functions, e.g., unnormalized Kullback–Leibler and logistic regression. The third assumption is mild, commonly used in DRM literature (Kiryo et al., 2017; Lu et al., 2020; Kato & Teshima, 2021).

**Theorem 1** (High-probability ratio estimation error bound for client $k$). *Let $f$ be a strictly convex function with bounded gradient. Denote $\Delta_\ell := \sup_{z \in \mathcal{B}_f} \max_{i \in \{1,2\}} |\ell_i(z)|$, $\hat{r}_k := \arg\min_{r_k \in \mathcal{H}_r} \hat{\mathcal{E}}_f^+(r_k)$ and $r_k^* := \arg\min_{r_k \in \mathcal{H}_r} \mathcal{E}_f(r_k)$ where $\hat{\mathcal{E}}_f^+$ and $\mathcal{E}_f$ are defined in Eqs. (3.2) and (4.1), respectively. Suppose that $\ell_1$ and $\ell_2$ satisfy Assumption 1, then for any $0 < \delta < 1$, with probability at least $1 - \delta$:*

$$\mathcal{E}_f(\hat{r}_k) - \mathcal{E}_f(r_k^*) \lesssim R_{n_k^{\mathrm{tr}}}^{p_k^{\mathrm{tr}}}(\mathcal{H}_r) + C_k \sum_{l=1}^{K} R_{n^{\mathrm{te}}}^{p_l^{\mathrm{te}}}(\mathcal{H}_r) + \sqrt{\Upsilon \log \frac{1}{\delta}} + K C_k \Delta_\ell \exp\left(\frac{-1}{\Upsilon}\right) \tag{4.2}$$

*where $\Upsilon = \Delta_\ell^2 (1/n_k^{\mathrm{tr}} + C_k^2 K / n^{\mathrm{te}})$.*

*Remark* 1. Theorem 1 provides generalization guarantees for a general nnBD DRM in a federated setting under a strictly convex $f$ with bounded gradient. We make the following remarks.
1) Typically, the required number of samples to accurately estimate density ratios scales exponentially with the dimensionality of data due to the curse of dimensionality. Theorem 1 bounds estimation error without considering approximation error. The curse of dimensionality is avoided when e.g., the ratio model $\mathcal{H}_r$ is rich enough and contains the true ratio that is smooth enough.
2) Our results are general to cover various ratio models. For example, in Corollary 1 of Appendix J, we consider neural networks with depth $L$ and bounded Frobenius norm $\|\mathbf{W}_i\|_F \leq \Delta_{\mathbf{W}_i}$ and establish explicit ratio estimation error bounds for client $k$ in $\mathcal{O}\left(\sqrt{L} \prod_{i=1}^{L} \Delta_{\mathbf{W}_i}(1/\sqrt{n_k^{\mathrm{tr}}} + K/\sqrt{n^{\mathrm{te}}}) + \sqrt{\Upsilon \log \frac{1}{\delta}} + K C_k \Delta_\ell \exp\left(\frac{-1}{\Upsilon}\right)\right)$.
3) If the additional error due to estimation of $\bar{r}_k$ with HDRM in Section 3 using $M$ bins is considered, it leads to $\mathcal{O}(K \Delta_\ell(\frac{1}{M} + \sqrt{\frac{M}{n_k^{\mathrm{tr}}}}))$ under mild assumptions. Refer to Appendix K for details.
4) Our error bound increases with $K$ due to the structure of BD risk. Note that $K$ is in a constant order. Our goal is to show that nnBD DRM is guaranteed to generalize in a general federated setting.

## 4.2 Excess risk and benefit of FTW-ERM

In this section, we aim to demonstrate the benefit of importance weighting in term of excess risk through bias-variance decomposition. We consider the classical least squares problem, a good starting point to understand the superiority of FTW-ERM over ERM with generalization guarantees. We consider the single client setting $K = 1$ for the ease of description, and our results can be extended to the multiple clients setting.

Let $(\mathbf{x}, y)$ denote the (test) data sampled from an unknown probability measure $\rho$. The least squares problem is to estimate the true parameter $\boldsymbol{\theta}_*$, which is assumed to be the unique solution that minimizes the *population risk* in a Hilbert space $\mathcal{H}$: $L(\boldsymbol{\theta}_*) = \min_{\boldsymbol{\theta} \in \mathcal{H}} L(\boldsymbol{\theta})$ where $L(\boldsymbol{\theta}) := \frac{1}{2}\mathbb{E}_{(\mathbf{x},y)\sim\rho}[(y - \boldsymbol{\theta}^\top \mathbf{x})]^2$. Moreover, we have $L(\boldsymbol{\theta}_*) = \sigma_\epsilon^2$ corresponding to the noise level. For an estimate $\boldsymbol{\theta}$ found by a learning algorithm such as ridge regression, its performance is measured by the expected *excess risk*, $R(\boldsymbol{\theta}) := \mathbb{E}[L(\boldsymbol{\theta})] - L(\boldsymbol{\theta}_*)$, where the expectation is over the random noise, randomness of the algorithm, and training data. In the following, we consider two settings: random-design setting and fixed-design settings where the training data matrix is random and given, respectively.

**Bias variance decomposition.** We need the following noise assumption for our proof.

**Assumption 2** (Dhillon et al. 2013; Zou et al. 2021, bounded variance). Let $\epsilon := y - \boldsymbol{\theta}_*^\top \mathbf{x}$. We assume that $\mathbb{E}[\epsilon] = 0$ and $\mathbb{E}[\epsilon^2] = \sigma_\epsilon^2$.

We have the following lemma on the bias-variance decomposition of the ridge regression FTW-ERM estimate in the random-design setting.

**Lemma 1.** *Let $\mathbf{X} \in \mathbb{R}^{n \times d}$ be the training data matrix. Let $\mathbf{W} = \operatorname{diag}(w_1, \ldots, w_n)$ with $w_i = p^{\text{te}}(\mathbf{x}_i)/p^{\text{tr}}(\mathbf{x}_i)$ for $i \in [n]$, $\hat{\boldsymbol{\theta}}$ be the regularized least square estimate with importance weighting: $\hat{\boldsymbol{\theta}} = \arg\min_{\boldsymbol{\theta}} \sum_{i=1}^n w_i(\boldsymbol{\theta}^\top \mathbf{x}_i - y_i)^2 + \lambda\|\boldsymbol{\theta}\|_2^2$ where $\lambda$ is the regularization parameter. Denote $\boldsymbol{\theta}_*$ be the true estimate, then the excess risk can be decomposed as the bias $\mathtt{B}$ and the variance $\mathtt{V}$: $\mathbb{E}[L(\hat{\boldsymbol{\theta}})] - L(\boldsymbol{\theta}_*) = \mathtt{B} + \mathtt{V}$, with*

$$\mathtt{B} := \lambda^2 \mathbb{E}\left[\boldsymbol{\theta}_*^\top \boldsymbol{\Sigma}_{\mathbf{W},\lambda}^{-1} \boldsymbol{\Sigma}^{\text{te}} \boldsymbol{\Sigma}_{\mathbf{W},\lambda}^{-1} \boldsymbol{\theta}_*\right], \quad \mathtt{V} := \sigma_\epsilon^2 \mathbb{E}\left[\operatorname{tr}\left(\boldsymbol{\Sigma}_{\mathbf{W},\lambda}^{-1} \mathbf{X}^\top \mathbf{W}^2 \mathbf{X} \boldsymbol{\Sigma}_{\mathbf{W},\lambda}^{-1} \boldsymbol{\Sigma}^{\text{te}}\right)\right]$$

*where $\boldsymbol{\Sigma}_{\mathbf{W},\lambda} := \mathbf{X}^\top \mathbf{W} \mathbf{X} + \lambda \mathbf{I}$ and $\boldsymbol{\Sigma}^{\text{te}} = \mathbb{E}_{\mathbf{x}}[\mathbf{x}\mathbf{x}^\top]$. Note that the expectation is taken over the randomness of the training data matrix $\mathbf{X}$ and label noise.*

*Remark* 2. Our results in Lemma 1 hold under the fixed-design setting where the training data are given (Dhillon et al., 2013; Hsu et al., 2012), by omitting the expectations from $\mathtt{B}$ and $\mathtt{V}$.

**One-hot case.** To theoretically prove that FTW-ERM outperforms ERM in non-trivial settings, we start from the one-hot case, along the lines of Zou et al. (2021), and strictly show that, under which level of covariate shift, the excess risk of FTW-ERM is always smaller than the classical ERM.

To be specific, in the one-hot case, every training data $\mathbf{x}$ is sampled from the set of natural basis $\{\mathbf{e}_1, \mathbf{e}_2, \ldots, \mathbf{e}_d\}$ according to the data distribution given by $\Pr\{\mathbf{x} = \mathbf{e}_i\} = \lambda_i$ where $0 < \lambda_i \leq 1$ and $\sum_i \lambda_i = 1$. The class of one-hot least square instances is characterized by the following problem set: $\{(\boldsymbol{\theta}_*; \lambda_1, \ldots, \lambda_d) : \boldsymbol{\theta}_* \in \mathcal{H}, \sum_i \lambda_i = 1\}$. It is not difficult to show that the population second momentum matrix is $\boldsymbol{\Sigma}^{\text{tr}} = \mathbb{E}[\mathbf{x}_i \mathbf{x}_i^\top] = \operatorname{diag}(\lambda_1, \ldots, \lambda_d)$ for $i \in [n]$. Similarly, we assume that each test data follows the same scheme but with different probabilities $\Pr\{\mathbf{x} = \mathbf{e}_i\} = \lambda_i'$, and hence, we have $\boldsymbol{\Sigma}^{\text{te}} = \operatorname{diag}(\lambda_1', \ldots, \lambda_d')$. This is a relatively simple setting, which admits covariate shift. Take $\{\mu_1, \mu_2, \ldots, \mu_d\}$ as the eigenvalues of $\mathbf{X}^\top \mathbf{X}$. Since $\mathbf{x}_i$ can only take on natural basis, the eigenvalue $\mu_i$ can be understood as the number of training data that equals $\mathbf{e}_i$. For notational simplicity, we rearrange the order of the training data following the decreasing order of the ratio, such that the $i$-th sample $\mathbf{x}_i$ corresponds to the ratio $w_i$ as the exact $i$-th largest value.

**Theorem 2.** *Let $\hat{\boldsymbol{\theta}}$ be the estimate of FTW-ERM, $\boldsymbol{\theta}^{\text{v}}$ be the classical ERM, and $\xi_i := \frac{\lambda}{\lambda + \mu_i}$. Under the fixed-design setting in the one-hot case, label noise assumption, and data correlation assumption, if the ratio $w_i := p^{\text{te}}(\mathbf{x}_i)/p^{\text{tr}}(\mathbf{x}_i)$ satisfies*

$$\sqrt{\frac{\lambda_i'}{\lambda_i}} - 1 \leq w_i \leq \xi_i \sqrt{\frac{\lambda_i}{\lambda_i'}}, \tag{4.3}$$

*then we have $R(\hat{\boldsymbol{\theta}}) \leqslant R(\boldsymbol{\theta}^{\text{v}})$.*

**Table 1:** Fashion MNIST with label shift across five clients, where each client receives different fractions of examples from each class. In this case, FTW-ERM achieves a better average accuracy than the baselines.

|  | FTW-ERM | FITW-ERM | FedAvg |
|---|---|---|---|
| Average accuracy | **0.8245** $\pm$ 0.0111 | 0.7942 $\pm$ 0.0096 | 0.5475 $\pm$ 0.0093 |
| Client 1 accuracy | **0.8627** $\pm$ 0.0175 | 0.8336 $\pm$ 0.0066 | 0.3978 $\pm$ 0.0215 |
| Client 2 accuracy | **0.9308** $\pm$ 0.0057 | 0.8896 $\pm$ 0.0124 | 0.9143 $\pm$ 0.0048 |
| Client 3 accuracy | **0.7742** $\pm$ 0.0618 | 0.7275 $\pm$ 0.0261 | 0.3677 $\pm$ 0.0297 |
| Client 4 accuracy | 0.7933 $\pm$ 0.0598 | **0.8204** $\pm$ 0.0152 | 0.6566 $\pm$ 0.0447 |
| Client 5 accuracy | **0.7616** $\pm$ 0.0593 | 0.6998 $\pm$ 0.0649 | 0.4009 $\pm$ 0.0642 |

**Table 2:** Average, worst-case, and best-case client accuracies of CIFAR10 target shift experiment across 100 clients where 5 randomly sampled clients participate in every round of training.

|  | FTW-ERM | FedAvg | FedBN |
|---|---|---|---|
| Average client accuracy | **0.7658** | 0.7237 | 0.4934 |
| Worst client accuracy | **0.6163** | 0.5403 | 0.1678 |
| Best client accuracy | **0.9016** | 0.8904 | 0.8233 |

*Remark* 3. We have the following remarks:

1) The condition (4.3) is equivalent to $\sqrt{\frac{\lambda_i'}{\lambda_i}} \in \left(0, \frac{1+\sqrt{1+4\xi_i}}{2}\right)$, which requires the training and test data to behave similarly in terms of eigenvalues, avoiding significant differences under distribution shifts for learnability. Other metrics, e.g., similarity on eigenvectors (Tripuraneni et al., 2021) also coincide with the spirit of our assumption.

2) The ratio matrix is $\mathbf{W} \in \mathbb{R}^{n \times n}$. However, we only need its top $d$ eigenvalue, i.e., the top-$d$ ratios. In particular, the last $n - d$ ratios have no effect on the final excess risk. This makes our algorithm robust to noise and domain shift.

3) For the special case by taking the ratio as $w_i := \sqrt{\frac{\lambda_i'}{\lambda_i}}$, we have

$$\mathtt{B}(\hat{\boldsymbol{\theta}}) = \lambda^2 \sum_{i=1}^{d} \frac{[(\boldsymbol{\theta}_*)_i]^2 \lambda_i'}{[\mu_i w_i + \lambda]^2} = \lambda^2 \sum_{i=1}^{d} \frac{[(\boldsymbol{\theta}_*)_i]^2 \lambda_i}{\left[\mu_i + \sqrt{\frac{\lambda_i}{\lambda_i'}}\lambda\right]^2},$$

which implies that the ratio can be regarded as an implicit regularization (Zou et al., 2021).

## 5 Experimental evaluation

In this section, we illustrate conditions under which FTW-ERM is favored over both Federated Averaging (FedAvg) (McMahan et al., 2017), FedBN (Li et al., 2021c), and FITW-ERM. For MNIST-based experiments we use a LeNet (LeCun et al., 1989) with cross entropy loss and compute standard deviations over *5 independent executions*. For CIFAR10-based experiments we use the larger ResNet-18 network (He et al., 2016). Further implementation details can be found in Appendix O.

**Target shift.** We consider the case of target shift where the label distribution $p(y)$ changes but the conditional distribution $p(\mathbf{x}|y)$ remains invariant. We split the 10-class Fashion MNIST dataset between 5 clients and simulate a target shift by including different fractions of examples from each class across the training data and test data. We further consider the separable case in order to compute the exact ratio for FTW-ERM and FITW-ERM in closed form. The specific distribution and the construction of the ratio can be found in Appendix O.1. The results in Table 1 illustrate that FITW-ERM can outperform FedAvg on average while preserving the same level of privacy. By relaxing the privacy slightly the proposed FTW-ERM improves on FedAvg uniformly across all clients. Even though the proportions of the classes have been artificially

**Table 3:** A challenging binary classification task on Colored MNIST with covariate shift across two clients. FTW-ERM is close to the idealised baseline that ignores the spurious correlation (Grayscale).

|  | Upper Bound (Grayscale) | FTW-ERM | FITW-ERM | FedAvg |
|---|---|---|---|---|
| Average accuracy | $0.68 \pm 0.01$ | $\mathbf{0.66} \pm 0.01$ | $0.63 \pm 0.00$ | $0.58 \pm 0.01$ |

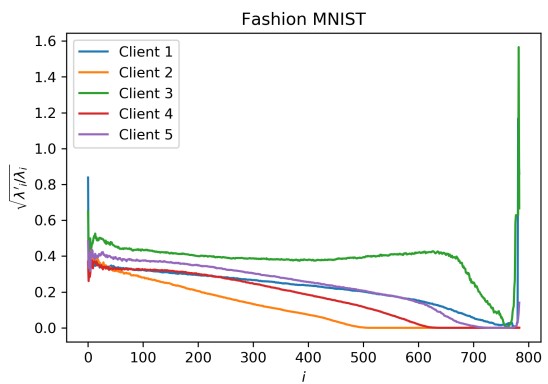 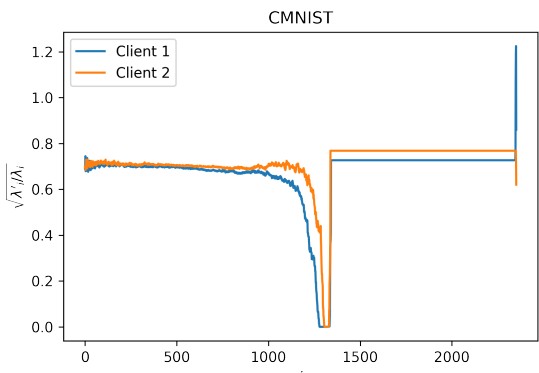

**Figure 2:** The squared ratio of eigenvalues ordered in descending order are all below 1 thus satisfying $\sqrt{\frac{\lambda_i'}{\lambda_i}} \in \left(0, \frac{1+\sqrt{1+4\xi_i}}{2}\right)$ in Theorem 2. The sudden increase in the ratio for the lowest eigenvalues are most likely due to numerical error occuring when the eigenvalues are close to zero (cf. Figure 3).

created, we believe that this demonstrates a realistic scenario where clients have a different fraction of samples per class. Additional experiments using larger models on the CIFAR10 dataset under a challenging target shift setting can be found in Appendix O.1 where FTW-ERM is observed to improve uniformly over FedAvg.

To model a scenario closer to real-world FL, we consider a setting with 100 clients on CIFAR10 under challenging distribution shifts and partial participation of clients, which is a requirement for cross-device FL (Kairouz et al., 2021; Wang et al., 2021). We sub-sample 5 clients uniformly at random at every round for 200,000 iterations. The target distribution is described in Table 6 and experimental results can be found in Table 2. We observe that FTW-ERM uniformly improves the test accuracy when compared with FedAvg and FedBN Li et al. (2021c) and that the gap is especially large between the worst-performing clients. The difficulty of FedBN under partial participation is most likely due how the method performs batch normalization. The batch normalization parameters are only maintained locally on each client and are consequently only updated when the given client is sampled. For experiments under full participation see Table 7.

**Covariate shift.** We now focus on covariate shift, where $p(\mathbf{x})$ undergoes a shift while $p(y|\mathbf{x})$ remains unchanged. For this setting, we extend the Colored MNIST dataset in Arjovsky et al. (2019) to the multi-client setting. The dataset is constructed by first assigning a binary label 0 to digits from 0-4 and label 1 for digits between 5-9. The label is then flipped with probability 0.25 to make the dataset non-separable. A spurious correlation is introduced by coloring the digits according to their assigned labels and then flipping the colors according to a different probability for each distribution (see Appendix O.2). For this experimental setup, we introduce an idealized scheme, which ignores the color and thus the spurious correlation, i.e., provides an upper bound, and is referred to as *Grayscale*. FTW-ERM outperforms both baselines in terms of the average accuracy even in a two-client setting. FTW-ERM is also close to Grayscale upper bound that by construction ignores the spurious correlations.

**Verifying assumptions.** Consider the two datasets used for the main experiments in Table 1 and Table 3. We verify in Figure 2 that the eigenvalues of the training distribution and test distribution for each client satisfy $\sqrt{\frac{\lambda_i'}{\lambda_i}} \in \left(0, \frac{1+\sqrt{1+4\xi_i}}{2}\right)$ in Theorem 2.

# 6 Conclusions and future work

In this work, we focus on FL under both intra-client and inter-client distribution shifts and propose FTW-ERM to improve the overall generalization performance. We establish high-probability ratio estimation guarantees for a general DRM method in a federated setting. We further show the benefit of importance weighting in term of excess risk through bias-variance decomposition in a ridge regression problem. Our theoretical guarantees indicate how FTW-ERM can provably solve a learning task under distribution shifts. We experimentally evaluate FTW-ERM under both label shift and covariate shift cases. Our experimental results validate that under certain covariate and target shifts, the proposed method can learn the task, while baselines such as vanilla federated averaging fails to do so. We anticipate that our methods to be applicable in learning from e.g., medical data, where there might be arbitrary skews on the distribution. In addition, we believe our study can further encourage the investigation of distribution shifts in FL, as this is a critical subject for learning across clients.

### Acknowledgments

This work was supported by the Hasler Foundation Program: Hasler Responsible AI (project number 21043) and by the Swiss National Science Foundation (SNSF) under grant number 200021_205011.

The work of Ali Ramezani-Kebrya was supported by the Research Council of Norway, through its Centre for Research-based Innovation funding scheme (Visual Intelligence under grant no. 309439), and Consortium Partners.

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

## A   Appendix

The appendix is organized as follows:

- Examples of $f$ for BD-based DRM are provided in Appendix A.

- Complete related work is provided in Appendix B.

- FTW-ERM with a focus on minmizing $R_1$ are provided in Appendix C.

- Details of density ratio estimation are provided in Appendix D.

- Communication costs of FTW-ERM and FITW-ERM are analyzed in in Appendix E.

- UKL, LR, and PU variants of nnBD are provided in Appendix F.

- Convergence of $\tilde{r}$ and $k$-means clustering for HDRM are provided in Appendix G.

- UKL, LR, and PU variants of nnBD for multiple clients are provided in Appendix H.

- The proof of the core Theorem 1 exists in Appendix I.

- High-probability ratio estimation error bounds for multi-layer perceptron and multiple clients are established in Appendix J.

- Additional error due to estimation of $\bar{r}_k$ with HDRM is analyzed in Appendix K.

- Lemma 1 is proved in Appendix L.

- Theorem 2 is proved in Appendix M.

- A counterexample under which FTW-ERM cannot outperform ERM is provided in Appendix N.

- Additional experimental details are included in Appendix O.

- Computational complexity of Algorithm 1 is analyzed in Appendix P.

- The limitations of our work are described in Appendix Q.

**Table 4:** Examples of $f$ for BD-based methods (Sugiyama et al., 2012; Kato & Teshima, 2021), LSIF = least-squares importance fitting, LR = logistic regression, BKL = binary Kullback–Leibler, UKL = unnormalized Kullback– Leibler, KLIEP = Kullback– Leibler importance estimation procedure, KMM = kernel mean matching , PULogLoss = positive and unlabeled learning with log Loss.

| Reference | Algorithm | $f(z)$ |
|-----------|-----------|--------|
| Basu et al. (1998) | Robust | $\frac{z^{\alpha+1}-z}{\alpha},\ \alpha > 0$ |
| Hastie et al. (2001) | LR (BKL) | $z\log(z) - (z+1)\log(z+1)$ |
| Kanamori et al. (2009) | LSIF | $\frac{(z-1)^2}{2}$ |
| Gretton et al. (2009) | KMM | $\frac{(z-1)^2}{2}$ |
| Nguyen et al. (2010) | KLIEP | $z\log(z) - z$ |
| Nguyen et al. (2010) | UKL | $z\log(z) - z$ |
| Kato et al. (2019) | PULogLoss | $C\log(1-z) + Cz(\log(z) - \log(1-z)),\ z \in (0,1), C \le \bar{r}$ |

## B   Complete related work

**Federated learning.**   One well-known method in FL is FedAvg (McMahan et al., 2017). FedAvg and its variants are extensively studied in optimization with a focus on communication efficiency and partial participation of clients while preserving privacy.

Indeed, a host of techniques, such as gradient quantization, sparsification, and local updating rules, have been proposed to improve communication efficiency in FL (Alistarh et al., 2017; Faghri et al., 2020; Ramezani-Kebrya et al., 2021; Kairouz et al., 2021; Ramezani-Kebrya et al., 2023). Furthermore, robust and secure aggregation schemes have been also proposed to provide robustness against training-time attacks launched by an adversary, and to compute aggregated values without being able to inspect the clients' local models and data, respectively (Li et al., 2020; Kairouz et al., 2021; Wang et al., 2021).

Taken together, these work largely focus on minimizing the empirical risk in the optimization objective, under the same training/test data distribution assumption over each client. Differences across clients are handled using personalization methods based on heuristics and currently do not have a statistical learning theoretical support (Smith et al., 2017; Khodak et al., 2019; Li et al., 2021b).

In contrast, we focus on learning and overall generalization performance under both intra-client and inter-client distribution shifts. Communication-efficient, robust, and secure aggregations can be viewed as complementary technologies, which can be used along with our proposed FTW-ERM method to improve the generalization performance. In our setting, clients can also all participate in every training iteration, such as cross-silo FL.

We note that (Hanzely et al., 2020; Gasanov et al., 2022) focus on minimizing the *empirical risk*, under the *same training/test data distribution assumption* over each client. Our formulation in FTW-ERM does not require specific assumptions on function $F_k$'s for $k \in [K]$ to provide an unbiased estimate of true risk minimizer. Under strong convexity and smoothness assumptions w.r.t. model parameters, similar optimal algorithms to those proposed in (Hanzely et al., 2020; Gasanov et al., 2022) will be optimal for FTW-ERM.

Different from recent FL work by Duan et al. (2021) and Li et al. (2021c), our work introduces new FTW-ERM formulation and shows *statistical consistency*.

**Importance-weighted ERM and density ratio matching.**   Density ratio estimation is an important step in various machine learning problems such as learning under covariate shift, learning under noisy labels, anomaly detection, two-sample testing, causal inference, change-pint detection, and classification from positive and unlabelled data (Qin, 1998; Shimodaira, 2000; Cheng & Chu, 2004; Keziou & Leoni-Aubin, 2005; Sugiyama et al., 2007; Kawahara & Sugiyama, 2009; Smola et al., 2009; Hido et al., 2011; Kanamori et al., 2011; Sugiyama et al., 2011; Yamada et al., 2011; Reddi et al., 2015; Liu & Tao, 2015; Kato et al., 2019; Fang et al., 2020; Uehara et al., 2020; Zhang et al., 2020; Kato & Teshima, 2021). In particular, covariate shift has been observed in real-world applications including brain-computer interfacing, emotion recognition,

human activity recognition, spam filtering, and speaker identification (Bickel & Scheffer, 2007; Li et al., 2010; Yamada et al., 2010; Hachiya et al., 2012; Jirayucharoensak et al., 2014). Shimodaira (2000) introduced covariate shift where the input train and test distributions are different while the conditional distribution of the output variable given the input variable remains unchanged. Importance-weighted ERM is widely used to improve generalization performance under covariate shift (Zadrozny, 2004; Sugiyama & Müller, 2005; Huang et al., 2006; Sugiyama et al., 2007; Kanamori et al., 2009; Sugiyama et al., 2012; Fang et al., 2020; Zhang et al., 2020; Kato & Teshima, 2021). Zhang et al. (2020) proposed a one-step approach that jointly learns the predictive model and the corresponding weights in one optimization problem. Sugiyama et al. (2012) proposed a Bregman divergence-based DRM, which unifies various DRMs. Kato & Teshima (2021) proposed a non-negative Bregman divergence-based DRM to resolve the overfitting issue when using deep neural networks for density ratio estimation. While this line of work focuses on DRM with a single train and test distributions, we consider a federated setting with multiple clients in this paper.

**Domain adaptation.** Distribution shifts between a source and a target domain have been a prominent problem in machine learning for several decades (Wang & Deng, 2018; Kouw & Loog, 2019). The premise behind such shifts is that data is frequently biased, and this results in distribution shifts that can be estimated by assuming some (unlabelled) knowledge of the target distribution. The following categories of domain adaptation methods are most closely related to our work: a) sample-based, and b) feature-based methods. In feature-based methods, the goal is to find a transformation that maps the source samples to target samples (Ganin et al., 2016; Bousmalis et al., 2017; Das & Lee, 2018; Damodaran et al., 2018). Sample-based methods aim at minimizing the target risk through data in the source domain. Importance weighting is often used in sample-based methods (Shimodaira, 2000; Jiang & Zhai, 2007; Baktashmotlagh et al., 2014). However, the focus on domain adaptation has been mainly to adapt to a single target distribution, not the overall generalization performance on multiple clients, which is addressed in this paper.

**Statistical generalization and excess risk bounds.** Understanding generalization performance of learning algorithms is one essential topic in modern machine learning. Typical techniques to establish generalization guarantees include uniform convergence by Rademacher complexity (Bartlett, 1998), and its variants (Bartlett et al., 2005), bias-variance decomposition (Geman et al., 1992; Adlam & Pennington, 2020), PAC-Bayes (McAllester, 1999), and stability-based analysis (Bousquet & Elisseeff, 2002; Shalev-Shwartz et al., 2010). Our work employs the first two techniques to analyze our density ratio estimation method in a federated setting and establish generalization guarantees for FTW-ERM, respectively. Rademacher complexity has been used in FL to obtain theoretical guarantees on the centralized model (Mohri et al., 2019) and personalized model (Mansour et al., 2020). Mohri et al. (2019) considered a scenario where a single target distribution is modeled as an unknown mixture of multiple domain distributions and obtained a global modal by minimizing the worst-case loss. This is different from our setting where we consider multiple test distributions for clients and focus on the overall test error. Mansour et al. (2020) studied personalization under the same training/test data distribution assumption over each client, which is different from our setting. Bias-variance decomposition provides a relatively refined characterization of generalization error (or excess risk), where a large bias indicates that a model is not flexible enough to learn from the data and a high variance indicates that the model performs unstably. Bias-variance decomposition is typically studied in two settings, i.e., the fixed and random design setting, which is categorized by whether the (training) data are fixed or random. This technique has been extensively applied in least squares (Hsu et al., 2012; Dieuleveut et al., 2017), analysis of SGD (Jain et al., 2018; Zou et al., 2021), and double descent (Adlam & Pennington, 2020).

Information-theoretic bounds on the generalization error and privacy leakage in federated settings were established in (Yagli et al., 2020). Under partial participation of clients, Yuan et al. (2022) proposed a framework, which distinguishes performance gaps due to unseen client data from performance gap due to unseen client distributions. Still, these work study FL under the same training/test data distribution assumption over each client.

**Table 5:** Details of scenarios described in Section 2.

| Scenario | #Clients | Assumptions on Distributions | What client 1 Knows |
|---|---|---|---|
| No-CS in (C.1) | 2 | $p_1^{tr}(\mathbf{x}) = p_1^{te}(\mathbf{x})$, $p_1^{tr}(\mathbf{x}) \neq p_2^{tr}(\mathbf{x})$ | $p_1^{tr}(\mathbf{x})/p_2^{tr}(\mathbf{x})$ |
| CS on one in (C.2) | 2 | $p_1^{tr}(\mathbf{x}) \neq p_1^{te}(\mathbf{x})$, $p_2^{tr}(\mathbf{x}) = p_2^{te}(\mathbf{x})$ | $p_1^{te}(\mathbf{x})/p_1^{tr}(\mathbf{x})$, $p_1^{te}(\mathbf{x})/p_2^{tr}(\mathbf{x})$ |
| CS on both in (C.2) | 2 | $p_1^{tr}(\mathbf{x}) \neq p_1^{te}(\mathbf{x})$, $p_2^{tr}(\mathbf{x}) \neq p_2^{te}(\mathbf{x})$ | $p_1^{te}(\mathbf{x})/p_1^{tr}(\mathbf{x})$, $p_1^{te}(\mathbf{x})/p_2^{tr}(\mathbf{x})$ |
| CS on multi. in (C.3) | $K$ | $p_k^{tr}(\mathbf{x}) \neq p_1^{te}(\mathbf{x})$ for all $k$ | $p_1^{te}(\mathbf{x})/p_k^{tr}(\mathbf{x})$ for all $k$ |

# C  FTW-ERM with a focus on minmizing $R_1$

Without loss of generality and for simplicity of notation, in this section, we set $l = 1$. We consider four typical scenarios under various distribution shifts and formulate their FTW-ERM with a focus on minmizing $R_1$. The details of these scenarios are summarized in Table 5.

*Remark* 4. Covariance shift (as well as its assumption) is the *most commonly used and studied* in *theory* and *practice* in distribution shifts (Sugiyama et al., 2007; Kanamori et al., 2009; Kato & Teshima, 2021; Uehara et al., 2020; Tripuraneni et al., 2021; Zhou & Levine, 2021). Handling covariate shift is a challenging issue, especially in federated settings (Kairouz et al., 2021).

**No intra-client covariate shift:** (No-CS) For description simplicity, we assume that there are only 2 clients but our results can be directly extended to multiple clients. This scenario assumes $p_k^{tr}(\mathbf{x}) = p_k^{te}(\mathbf{x})$ for $k = 1, 2$. Client 1 aims to learn $h_\mathbf{w}$ assuming $\frac{p_1^{tr}(x)}{p_2^{tr}(x)}$ is given. We consider the following FTW-ERM that is proved to be consistent in terms of minimizing minimizing $R_1$:

$$\min_{\mathbf{w} \in \mathbb{R}^d} \frac{1}{n_1^{tr}} \sum_{i=1}^{n_1^{tr}} \ell(h_\mathbf{w}(\mathbf{x}_{1,i}^{tr}), \mathbf{y}_{1,i}^{tr}) + \frac{1}{n_2^{tr}} \sum_{i=1}^{n_2^{tr}} \frac{p_1^{tr}(\mathbf{x}_{2,i}^{tr})}{p_2^{tr}(\mathbf{x}_{2,i}^{tr})} \ell(h_\mathbf{w}(\mathbf{x}_{2,i}^{tr}), \mathbf{y}_{2,i}^{tr}). \tag{C.1}$$

**Covariate shift only for client 1:** (CS on one) We now consider covariate shift only for client 1, i.e., $p_1^{tr}(\mathbf{x}) \neq p_1^{te}(\mathbf{x})$ and $p_2^{tr}(\mathbf{x}) = p_2^{te}(\mathbf{x})$. We consider the following FTW-ERM

$$\min_{\mathbf{w} \in \mathbb{R}^d} \frac{1}{n_1^{tr}} \sum_{i=1}^{n_1^{tr}} \frac{p_1^{te}(\mathbf{x}_{1,i}^{tr})}{p_1^{tr}(\mathbf{x}_{1,i}^{tr})} \ell(h_\mathbf{w}(\mathbf{x}_{1,i}^{tr}), \mathbf{y}_{1,i}^{tr}) + \frac{1}{n_2^{tr}} \sum_{i=1}^{n_2^{tr}} \frac{p_1^{te}(\mathbf{x}_{2,i}^{tr})}{p_2^{tr}(\mathbf{x}_{2,i}^{tr})} \ell(h_\mathbf{w}(\mathbf{x}_{2,i}^{tr}), \mathbf{y}_{2,i}^{tr}). \tag{C.2}$$

**Covariate shift for both clients:** (CS on both) We assume $p_1^{tr}(\mathbf{x}) \neq p_1^{te}(\mathbf{x})$ and $p_2^{tr}(\mathbf{x}) \neq p_2^{te}(\mathbf{x})$, *i.e.,* covariate shift for both clients. The corresponding FTW-ERM is the same as Eq. (C.2).

**Multiple clients:** (CS on multi.) Finally, we consider a general scenario with $K$ clients. We assume both intra-client and inter-client covariate shifts by the following FTW-ERM:

$$\min_{\mathbf{w} \in \mathbb{R}^d} \sum_{k=1}^{K} \frac{\lambda_k}{n_k^{tr}} \sum_{i=1}^{n_k^{tr}} \frac{p_1^{te}(\mathbf{x}_{k,i}^{tr})}{p_k^{tr}(\mathbf{x}_{k,i}^{tr})} \ell(h_\mathbf{w}(\mathbf{x}_{k,i}^{tr}), \mathbf{y}_{k,i}^{tr}) \tag{C.3}$$

where $\sum_{k=1}^{K} \lambda_k = 1$ and $\lambda_k \geq 0$.

**Proposition 2.** *Let $l \in [K]$. In above settings, FTW-ERM defined in Eqs. (C.1), (C.2), and (C.3) is consistent. i.e., the learned function converges in probability to the optimal function in terms of minimizing $R_1$.*

Proposition 2 implies that, under various settings, FTW-ERM outputs an unbiased estimate of a *minimizer of the true risk*.

*Proof.* For the scenario without intra-client covariate shift, FTW-ERM in Eq. (C.1) can be expressed as

$$
\frac{1}{n_2^{\mathrm{tr}}} \sum_{i=1}^{n_2^{\mathrm{tr}}} \frac{p_1^{\mathrm{tr}}(\mathbf{x}_{2,i}^{\mathrm{tr}})}{p_2^{\mathrm{tr}}(\mathbf{x}_{2,i}^{\mathrm{tr}})} \ell(h_{\mathbf{w}}(\mathbf{x}_{2,i}^{\mathrm{tr}}), \mathbf{y}_{2,i}^{\mathrm{tr}}) \xrightarrow{n_2^{\mathrm{tr}} \to \infty} \mathbb{E}_{p_2^{\mathrm{tr}}(\mathbf{x}, \mathbf{y})} \left[ \frac{p_1^{\mathrm{tr}}(\mathbf{x})}{p_2^{\mathrm{tr}}(\mathbf{x})} \ell(h_{\mathbf{w}}(\mathbf{x}), \mathbf{y}) \right]
$$

$$
= \int_{\mathcal{X}} \frac{p_1^{\mathrm{tr}}(\mathbf{x})}{p_2^{\mathrm{tr}}(\mathbf{x})} \mathbb{E}_{p(\mathbf{y}|\mathbf{x})} \left[ \ell(h_{\mathbf{w}}(\mathbf{x}), \mathbf{y}) \right] p_2^{\mathrm{tr}}(\mathbf{x}) \, \mathrm{d}\mathbf{x}
$$

$$
= \int_{\mathcal{X}} p_1^{\mathrm{tr}}(\mathbf{x}) \mathbb{E}_{p(\mathbf{y}|\mathbf{x})} \left[ \ell(h_{\mathbf{w}}(\mathbf{x}), \mathbf{y}) \right] \mathrm{d}\mathbf{x}
$$

$$
= \int_{\mathcal{X}} p_1^{\mathrm{te}}(\mathbf{x}) \mathbb{E}_{p(\mathbf{y}|\mathbf{x})} \left[ \ell(h_{\mathbf{w}}(\mathbf{x}), \mathbf{y}) \right] \mathrm{d}\mathbf{x}
$$

$$
= \mathbb{E}_{p_1^{\mathrm{te}}(\mathbf{x}, \mathbf{y})} \left[ \ell(h_{\mathbf{w}}(\mathbf{x}), \mathbf{y}) \right]
$$

$$
= R_1(h_{\mathbf{w}}).
$$

For the scenario with covariate shift only for client 1 or for both clients, FTW-ERM in Eq. (C.2) admits

$$
\frac{1}{n_2^{\mathrm{tr}}} \sum_{i=1}^{n_2^{\mathrm{tr}}} \frac{p_1^{\mathrm{te}}(\mathbf{x}_{2,i}^{\mathrm{tr}})}{p_2^{\mathrm{tr}}(\mathbf{x}_{2,i}^{\mathrm{tr}})} \ell(h_{\mathbf{w}}(\mathbf{x}_{2,i}^{\mathrm{tr}}), \mathbf{y}_{2,i}^{\mathrm{tr}}) \xrightarrow{n_2^{\mathrm{tr}} \to \infty} \mathbb{E}_{p_2^{\mathrm{tr}}(\mathbf{x}, \mathbf{y})} \left[ \frac{p_1^{\mathrm{te}}(\mathbf{x})}{p_2^{\mathrm{tr}}(\mathbf{x})} \ell(h_{\mathbf{w}}(\mathbf{x}), \mathbf{y}) \right]
$$

$$
= \int_{\mathcal{X}} \frac{p_1^{\mathrm{te}}(\mathbf{x})}{p_2^{\mathrm{tr}}(\mathbf{x})} \mathbb{E}_{p(\mathbf{y}|\mathbf{x})} \left[ \ell(h_{\mathbf{w}}(\mathbf{x}), \mathbf{y}) \right] p_2^{\mathrm{tr}}(\mathbf{x}) \, \mathrm{d}\mathbf{x}
$$

$$
= \int_{\mathcal{X}} p_1^{\mathrm{te}}(\mathbf{x}) \mathbb{E}_{p(\mathbf{y}|\mathbf{x})} \left[ \ell(h_{\mathbf{w}}(\mathbf{x}), \mathbf{y}) \right] \mathrm{d}\mathbf{x}
$$

$$
= \mathbb{E}_{p_1^{\mathrm{te}}(\mathbf{x}, \mathbf{y})} \left[ \ell(h_{\mathbf{w}}(\mathbf{x}), \mathbf{y}) \right]
$$

$$
= R_1(h_{\mathbf{w}}).
$$

We note that $\frac{p_1^{\mathrm{te}}(\mathbf{x})}{p_2^{\mathrm{tr}}(\mathbf{x})} = \frac{p_1^{\mathrm{te}}(\mathbf{x})}{p_1^{\mathrm{tr}}(\mathbf{x})} \frac{p_1^{\mathrm{tr}}(\mathbf{x})}{p_2^{\mathrm{tr}}(\mathbf{x})}$, which is the product of ratios due to intra-client covariate shift on client 1 and inter-client covariate shift.

For multiple clients, let $k \in [K]$. Similarly, we have

$$
\frac{1}{n_k^{\mathrm{tr}}} \sum_{i=1}^{n_k^{\mathrm{tr}}} \frac{p_1^{\mathrm{te}}(\mathbf{x}_{k,i}^{\mathrm{tr}})}{p_k^{\mathrm{tr}}(\mathbf{x}_{k,i}^{\mathrm{tr}})} \ell(h_{\mathbf{w}}(\mathbf{x}_{k,i}^{\mathrm{tr}}), \mathbf{y}_{k,i}^{\mathrm{tr}}) \xrightarrow{n_k^{\mathrm{tr}} \to \infty} R_1(h_{\mathbf{w}}).
$$

Then we have

$$
\sum_{k=1}^{K} \frac{\lambda_k}{n_k^{\mathrm{tr}}} \sum_{i=1}^{n_k^{\mathrm{tr}}} \frac{p_1^{\mathrm{te}}(\mathbf{x}_{k,i}^{\mathrm{tr}})}{p_k^{\mathrm{tr}}(\mathbf{x}_{k,i}^{\mathrm{tr}})} \ell(h_{\mathbf{w}}(\mathbf{x}_{k,i}^{\mathrm{tr}}), \mathbf{y}_{k,i}^{\mathrm{tr}}) \xrightarrow{n_1^{\mathrm{tr}}, \dots, n_K^{\mathrm{tr}} \to \infty} R_1(h_{\mathbf{w}}).
$$

The consistency of FTW-ERM, i.e., convergence in probability, is immediately followed the standard arguments in e.g., (Shimodaira, 2000)[Section 3] and (Sugiyama et al., 2007)[Section 2.2] using the law of large numbers. ∎

Note that to solve Eq. (C.3), client 1 needs to estimate $\frac{p_1^{\mathrm{te}}(\mathbf{x})}{p_k^{\mathrm{tr}}(\mathbf{x})}$ for all clients $k$ with $\lambda_k > 0$ in (C.3).

*Remark* 5. Scaling $\sum_{k=1}^{K} \lambda_k$ does not affect the optimal parameters in Eq. (C.3). For rotational simplicity, we set $\lambda_k = 1$ for $k \in [K]$.

## C.1 No intra-client shift

In this section, we consider the important and special case of the setting described in Section 2.1 under no intra-client covariate shifts but inter-client covariate shifts. For simplicity, we consider a two clients with

train/test distributions $P$ and $Q$ whose train/test densities are denoted by $p$ and $q$, respectively. We also suppose that we have a sample $z \sim P$ and $z' \sim Q$ to learn with the goal is to find an unbiased estimate of the overall risk with the smallest variance. In this setting, the classical ERM (FedAvg) objective $\ell(z, \theta) + \ell(z', \theta)$ is an unbiased estimate for the overall risk $L(\theta) = \mathbb{E}_P[\ell(z, \theta)] + \mathbb{E}_Q[\ell(z', \theta)]$[6]. In this setting, the objective of FTW-ERM, i.e., $\frac{1}{2}(\hat{L}_P(\theta) + \hat{L}_Q(\theta))$ with $\hat{L}_P(\theta) = \left(1 + \frac{q(z)}{p(z)}\right)\ell(z, \theta)$ and $\hat{L}_Q(\theta) = \left(1 + \frac{p(z')}{q(z')}\right)\ell(z', \theta)$ is an unbiased estimate for the overall risk $L(\theta)$.

We now show that the our method (FTW-ERM) has a smaller variance than FedAvg under certain conditions. Let $\mathbb{E}_P[(\ell(z, \theta) - \mathbb{E}_P[\ell(z, \theta)])^2] = \sigma_P^2$ and $\mathbb{E}_Q[(\ell(z', \theta) - \mathbb{E}_Q[\ell(z', \theta)])^2] = \sigma_Q^2$.

For FedAvg, the variance is given by

$$\mathbb{E}_{P,Q}[(\ell(z, \theta) + \ell(z', \theta) - L(\theta))^2] = \sigma_P^2 + \sigma_Q^2.$$

For FTW-ERM, the variance is given by

$$\mathbb{E}_{P,Q}[(\frac{1}{2}(\hat{L}_P(\theta) + \hat{L}_Q(\theta)) - L(\theta))^2] = V_P + V_Q$$

where $V_P = \frac{1}{4}\mathbb{E}_P[(\hat{L}_P(\theta) - L(\theta))^2]$ and $V_Q = \frac{1}{4}\mathbb{E}_Q[(\hat{L}_Q(\theta) - L(\theta))^2]$.

We now expand each term $V_P$ and $V_Q$. We can show that

$$V_P = \frac{1}{4}\mathbb{E}_P\left[\left((1 + \frac{q(z)}{p(z)})\ell(z, \theta) - \mathbb{E}_P[\ell(z, \theta)] - \mathbb{E}_Q[\ell(z', \theta)]\right)^2\right] = \frac{\sigma_P^2 + \tilde{\sigma}_P^2}{4}$$

where $\tilde{\sigma}_P^2 = \mathbb{E}_P\left[\left(\frac{q(z)}{p(z)}\ell(z, \theta) - \mathbb{E}_Q[\ell(z', \theta)]\right)^2\right] + 2\mathbb{E}_P\left[\left(\ell(z, \theta) - \mathbb{E}_P[\ell(z, \theta)]\right)\left(\frac{q(z)}{p(z)}\ell(z, \theta) - \mathbb{E}_Q[\ell(z', \theta)]\right)\right]$. Similarly, we have

$$V_Q = \frac{1}{4}\mathbb{E}_Q\left[\left((1 + \frac{p(z')}{q(z')})\ell(z', \theta) - \mathbb{E}_P[\ell(z, \theta)] - \mathbb{E}_Q[\ell(z', \theta)]\right)^2\right] = \frac{\sigma_Q^2 + \tilde{\sigma}_Q^2}{4}$$

where $\tilde{\sigma}_Q^2 = \mathbb{E}_Q\left[\left(\frac{p(z')}{q(z')}\ell(z', \theta) - \mathbb{E}_P[\ell(z, \theta)]\right)^2\right] + 2\mathbb{E}_Q\left[\left(\ell(z', \theta) - \mathbb{E}_Q[\ell(z', \theta)]\right)\left(\frac{p(z')}{q(z')}\ell(z', \theta) - \mathbb{E}_P[\ell(z, \theta)]\right)\right]$.

We note if $\tilde{\sigma}_P^2 + \tilde{\sigma}_Q^2 \leq 3(\sigma_P^2 + \sigma_Q^2)$ then, FTW-ERM will have smaller variance than FedAvg, i.e., $V_P + V_Q \leq \sigma_P^2 + \sigma_Q^2$. The exact condition depends on the loss and densities. To show a concrete example, for the more general and practical case with both intra/inter-client , in Section 4.2, we show that FTW-ERM results in smaller excess risk compared to FedAvg through a refined bias-variance decomposition. Given two distributions, considering the case of no intra-client shift is a special case, where it is true that FedAvg is an unbiased estimate of the overall risk. However, this unbiasedness breaks as soon as there is only one client whose test and train distributions are different, which is very common in theory and practice. Please note that FTW-ERM is an unbiased estimate of the overall risk in a general FL setting without requiring any prior knowledge/assumptions on the potential covariate shifts.

# D   Ratio estimation

## D.1   nnBD DRM for a single client

For simplicity, we firstly focus on the problem of estimating $r(\mathbf{x}) = \frac{p^{\text{te}}(\mathbf{x})}{p^{\text{tr}}(\mathbf{x})}$ and then extend our consideration to the estimation of $r_k(\mathbf{x})$ in Eq. (3.1). Let $r^*$ denote the true density ratio. Our goal is to estimate $r^*$ by optimizing our ratio model $r$. The discrepancy between $r$ and $r^*$ is measured by $\mathbb{E}_{p^{\text{tr}}}[\text{BD}_f(r^*(\mathbf{x}) \parallel r(\mathbf{x}))]$. We note that $\mathbb{E}_{p^{\text{tr}}}[\text{BD}_f(r^*(\mathbf{x}) \parallel r(\mathbf{x}))] = \mathcal{E}_f(r) + \mathbb{E}_{p^{\text{tr}}}[f(r^*(\mathbf{x}))]$ where $\mathcal{E}_f(r) = \mathbb{E}_{p^{\text{tr}}}[\nabla f(r(\mathbf{x}))r(\mathbf{x}) - f(r(\mathbf{x}))] - \mathbb{E}_{p^{\text{te}}}[\nabla f(r(\mathbf{x}))]$. Note that $\mathbb{E}_{p^{\text{tr}}}[f(r^*(\mathbf{x}))]$ is constant w.r.t. $r$. Let $\{\mathbf{x}_i^{\text{tr}}\}_{i=1}^{n^{\text{tr}}}$ and $\{\mathbf{x}_j^{\text{te}}\}_{j=1}^{n^{\text{te}}}$ denote unlabelled

---

[6]For notational simplicity, we overload $\ell(z, \theta)$ to denote the loss of model $\theta$ on example $z$.

samples drawn i.i.d. from distributions $p^{\mathrm{tr}}$ and $p^{\mathrm{te}}$, respectively. Let $\mathcal{H}_r \subset \{r : \mathcal{X} \to \mathcal{B}_f\}$ denote a hypothesis class for our model $r$. Using an empirical approximation of $\mathcal{E}_f(r^*(\mathbf{x}) \| r(\mathbf{x}))$, Sugiyama et al. (2012) formulated BD-based DRM problem as $\min_{r \in \mathcal{H}_r} \hat{\mathcal{E}}_f(r)$ where

$$\hat{\mathcal{E}}_f(r) = \frac{1}{n^{\mathrm{tr}}} \sum_{i=1}^{n^{\mathrm{tr}}} \left( \nabla f(r(\mathbf{x}_i^{\mathrm{tr}}))r(\mathbf{x}_i^{\mathrm{tr}}) - f(r(\mathbf{x}_i^{\mathrm{tr}})) \right) - \frac{1}{n^{\mathrm{te}}} \sum_{j=1}^{n^{\mathrm{te}}} \nabla f(r(\mathbf{x}_j^{\mathrm{te}})). \tag{D.1}$$

Sugiyama et al. (2012) showed that BD-based DRM unifies well-known density ratio estimation methods by substituting an appropriate $f$ in (D.1). However, it is shown that solving BD-based DRM with highly flexible models such as neural networks typically leads to an over-fitting issue (Kato & Teshima, 2021; Kiryo et al., 2017). In particular, Kato & Teshima (2021) called such issue "train-loss hacking" where $-\frac{1}{n^{\mathrm{te}}} \sum_{j=1}^{n^{\mathrm{te}}} \nabla f(r(\mathbf{x}_j^{\mathrm{te}}))$ in (D.1) diverges if there is no lower bound on this term. Even when there exists a lower bound, the model $r$ tends to increase to the largest possible values of its output range at points $\{\mathbf{x}_j^{\mathrm{te}}\}_{j=1}^{n^{\mathrm{te}}}$. To resolve such issue, Kato & Teshima (2021) proposed to use non-negative BD (nnBD) DRM, i.e., $\min_{r \in \mathcal{H}_r} \hat{\mathcal{E}}_f^+(r)$ where

$$\hat{\mathcal{E}}_f^+(r) = \mathrm{ReLU}\left( \frac{1}{n^{\mathrm{tr}}} \sum_{i=1}^{n^{\mathrm{tr}}} \ell_1(r(\mathbf{x}_i^{\mathrm{tr}})) - \frac{C}{n^{\mathrm{te}}} \sum_{j=1}^{n^{\mathrm{te}}} \ell_1(r(\mathbf{x}_j^{\mathrm{te}})) \right) + \frac{1}{n^{\mathrm{te}}} \sum_{j=1}^{n^{\mathrm{te}}} \ell_2(r(\mathbf{x}_j^{\mathrm{te}})), \tag{D.2}$$

$\mathrm{ReLU}(z) = \max\{0, z\}$, $0 < C < \frac{1}{\bar{r}}$, $\bar{r} = \sup_{\mathbf{x} \in \mathcal{X}^{\mathrm{tr}}} r^*(\mathbf{x})$, $\ell_1(z) = \nabla f(z)z - f(z)$, and $\ell_2(z) = C(\nabla f(z)z - f(z)) - \nabla f(z)$. Substituting $f(z) = \frac{(z-1)^2}{2}$ into (D.2), the least-squares importance fitting (LSIF) variant of nnBD is given by

$$\hat{\mathcal{E}}_{\mathrm{LSIF}}^+(r) = \mathrm{ReLU}\left( \frac{1}{2n^{\mathrm{tr}}} \sum_{i=1}^{n^{\mathrm{tr}}} r^2(\mathbf{x}_i^{\mathrm{tr}}) - \frac{C}{2n^{\mathrm{te}}} \sum_{j=1}^{n^{\mathrm{te}}} r^2(\mathbf{x}_j^{\mathrm{te}}) \right) - \frac{1}{n^{\mathrm{te}}} \sum_{j=1}^{n^{\mathrm{te}}} \left( r(\mathbf{x}_j^{\mathrm{te}}) - \frac{C}{2} r^2(\mathbf{x}_j^{\mathrm{te}}) \right).$$

In Appendix F, we show explicit expressions for unnormalized Kullback–Leibler (UKL), logistic regression (LR), and positive and unlabeled learning (PU) variants of nnBD.

Estimating $\bar{r} = \sup_{\mathbf{x} \in \mathcal{X}^{\mathrm{tr}}} r^*(\mathbf{x})$ is a key step for density ratio estimation. It is shown that underestimating $C$ leads to significant performance degradation (Kato & Teshima, 2021, Section 5). Kato & Teshima (2021) considered $C$ as a hyper-parameter, which can be tuned. However, obtaining an efficient estimate of $\bar{r}$ is desirable, in particular when training a deep model.

Let $\mathcal{B} \subset \mathcal{X}^{\mathrm{tr}}$. Assume $p^{\mathrm{tr}}$ and $p^{\mathrm{te}}$ are continuous. Since $\mathcal{B}$ is connected and Lebesgue-measurable with finite measure, by applying intermediate value theorem (Russ, 1980), there exist $\tilde{x}^{\mathrm{tr}}$ and $\hat{x}^{\mathrm{te}}$ such that $\Pr\{X^{\mathrm{tr}} \in \mathcal{B}\} = p^{\mathrm{tr}}(\tilde{x}^{\mathrm{tr}})\mathrm{Vol}(\mathcal{B})$ and $\Pr\{X^{\mathrm{te}} \in \mathcal{B}\} = p^{\mathrm{te}}(\hat{x}^{\mathrm{te}})\mathrm{Vol}(\mathcal{B})$ where $\mathrm{Vol}(\mathcal{B}) = \int_{\mathbf{x} \in \mathcal{B}} d\mathbf{x}$. We note that $\sup_{\mathbf{x} \in \mathcal{B}} r^*(\mathbf{x}) \leq \frac{\sup_{\mathbf{x} \in \mathcal{B}} p^{\mathrm{te}}(\mathbf{x})}{\inf_{\mathbf{x} \in \mathcal{B}} p^{\mathrm{tr}}(\mathbf{x})}$ and $\frac{p^{\mathrm{te}}(\hat{x}^{\mathrm{te}})}{p^{\mathrm{tr}}(\tilde{x}^{\mathrm{tr}})} \leq \frac{\sup_{\mathbf{x} \in \mathcal{B}} p^{\mathrm{te}}(\mathbf{x})}{\inf_{\mathbf{x} \in \mathcal{B}} p^{\mathrm{tr}}(\mathbf{x})}$. We partition $\mathcal{X}^{\mathrm{tr}}$ into $M$ bins where for each bin $\mathcal{B}_m$, if there exists some $\mathbf{x}_i^{\mathrm{tr}} \in \mathcal{B}_m$, then we define $\tilde{r}_m := \frac{\Pr\{X^{\mathrm{te}} \in \mathcal{B}_m\}}{\Pr\{X^{\mathrm{tr}} \in \mathcal{B}_m\}} \simeq \frac{\frac{1}{n^{\mathrm{te}}} \sum_{j=1}^{n^{\mathrm{te}}} \mathbb{1}(\mathbf{x}_j^{\mathrm{te}} \in \mathcal{B}_m)}{\frac{1}{n^{\mathrm{tr}}} \sum_{i=1}^{n^{\mathrm{tr}}} \mathbb{1}(\mathbf{x}_i^{\mathrm{tr}} \in \mathcal{B}_m)}$ for $m \in [M]$. Otherwise, $\tilde{r}_m = 0$. Finally, we propose to use $C \leq \frac{1}{\tilde{r}}$ where $\tilde{r} = \max\{\tilde{r}_1, \ldots, \tilde{r}_M\}$. Convergence of $\tilde{r}$ to $\bar{r}$ is established in Appendix G.

Now, suppose there are $K$ clients where each client provides $n^{\mathrm{te}}$ unlabelled test samples to the pool of samples. Our goal is to estimate $r_k$ in Eq. (3.1) for $k = 1, \ldots, K$. The BD-based DRM for client $k$ is given by $\min_{r_k \in \mathcal{H}_r} \hat{\mathcal{E}}_f(r_k)$ where $\hat{\mathcal{E}}_f(r_k) = \frac{1}{n_k^{\mathrm{tr}}} \sum_{i=1}^{n_k^{\mathrm{tr}}} \left( \nabla f(r_k(\mathbf{x}_{k,i}^{\mathrm{tr}}))r_k(\mathbf{x}_{k,i}^{\mathrm{tr}}) - f(r_k(\mathbf{x}_{k,i}^{\mathrm{tr}})) \right) - \frac{1}{n^{\mathrm{te}}} \sum_{j=1}^{n^{\mathrm{te}}} \sum_{l=1}^{K} \nabla f(r_k(\mathbf{x}_{l,j}^{\mathrm{te}}))$. The nnBD DRM problem for client $k$ is $\min_{r_k \in \mathcal{H}_r} \hat{\mathcal{E}}_f^+(r_k)$ where

$$\hat{\mathcal{E}}_f^+(r_k) = \mathrm{ReLU}(\hat{S}_{1,\ell_1}) + \frac{1}{n^{\mathrm{te}}} \sum_{j=1}^{n^{\mathrm{te}}} \sum_{l=1}^{K} \ell_2(r_k(\mathbf{x}_{l,j}^{\mathrm{te}})), \tag{D.3}$$

$\hat{S}_{1,\ell_1} = \frac{1}{n_k^{\text{tr}}} \sum_{i=1}^{n_k^{\text{tr}}} \ell_1(r_k(\mathbf{x}_{k,i}^{\text{tr}})) - \frac{C_k}{n^{\text{te}}} \sum_{j=1}^{n^{\text{te}}} \sum_{l=1}^{K} \ell_1(r_k(\mathbf{x}_{l,j}^{\text{te}}))$, $0 < C_k < \frac{1}{\bar{r}_k}$, and $\bar{r}_k = \sup_{\mathbf{x} \in \mathcal{X}^{\text{tr}}} r_k^*(\mathbf{x})$. Substituting $f(z) = \frac{(z-1)^2}{2}$ into (D.3), the LSIF variant of nnBD for client $k$ is given by $\min_{r_k \in \mathcal{H}_r} \hat{\mathcal{E}}_{\text{LSIF}}^+(r_k)$ where

$$\hat{\mathcal{E}}_{\text{LSIF}}^+(r_k) = \text{ReLU}(\hat{S}_{\text{LSIF}}) - \frac{1}{n^{\text{te}}} \sum_{j=1}^{n^{\text{te}}} \sum_{l=1}^{K} \left( r_k(\mathbf{x}_{l,j}^{\text{te}}) - \frac{C_k}{2} r_k^2(\mathbf{x}_{l,j}^{\text{te}}) \right), \tag{D.4}$$

and $\hat{S}_{\text{LSIF}} = \frac{1}{2n_k^{\text{tr}}} \sum_{i=1}^{n_k^{\text{tr}}} r_k^2(\mathbf{x}_{k,i}^{\text{tr}}) - \frac{C_k}{2n^{\text{te}}} \sum_{j=1}^{n^{\text{te}}} \sum_{l=1}^{K} r_k^2(\mathbf{x}_{l,j}^{\text{te}})$. We provide explicit expressions for UKL, LR, and PU variants of nnBD for client $k$ in Appendix H.

Our goal is to estimate $\bar{r}_k = \sup_{\mathbf{x} \in \mathcal{X}^{\text{tr}}} \frac{\sum_{l=1}^{K} p_l^{\text{te}}(\mathbf{x})}{p_k^{\text{tr}}(\mathbf{x})}$. For HDRM method, we first partition $\mathcal{X}^{\text{tr}}$ into $M$ bins where for each bin $\mathcal{B}_m$, if there exists some $\mathbf{x}_{k,i}^{\text{tr}} \in \mathcal{B}_m$, then we define $\tilde{r}_{k,m} := \frac{\sum_{l=1}^{K} \Pr\{X_l^{\text{te}} \in \mathcal{B}_m\}}{\Pr\{X_k^{\text{tr}} \in \mathcal{B}_m\}} \simeq \frac{\frac{1}{n^{\text{te}}} \sum_{j=1}^{n^{\text{te}}} \sum_{l=1}^{K} \mathbb{1}(\mathbf{x}_{l,j}^{\text{te}} \in \mathcal{B}_m)}{\frac{1}{n_k^{\text{tr}}} \sum_{i=1}^{n_k^{\text{tr}}} \mathbb{1}(\mathbf{x}_{k,i}^{\text{tr}} \in \mathcal{B}_m)}$ for $m \in [M]$. Otherwise, $\tilde{r}_{k,m} = 0$. Finally, we propose to use $C_k = \frac{1}{\tilde{r}_k}$ where $\tilde{r}_k = \max\{\tilde{r}_{k,1}, \ldots, \tilde{r}_{k,M}\}$.

## D.2  BD-based DRM for FL

Our goal is to estimate $r_k$ by minimizing the discrepancy $\mathbb{E}_{p_k^{\text{tr}}}[\text{BD}_f(r_k^*(\mathbf{x}) \parallel r_k(\mathbf{x}))]$, which is equivalent to $\min_{r_k \in \mathcal{H}_r} \mathcal{E}_f(r_k)$ where

$$\mathcal{E}_f(r_k) = \mathbb{E}_{p_k^{\text{tr}}}[\nabla f(r_k(\mathbf{x})) r_k(\mathbf{x}) - f(r_k(\mathbf{x}))] - \sum_{l=1}^{K} \mathbb{E}_{p_l^{\text{te}}}[\nabla f(r_k(\mathbf{x}))], \tag{D.5}$$

since $\mathbb{E}_{p_k^{\text{tr}}}[\text{BD}_f(r_k^*(\mathbf{x}) \parallel r_k(\mathbf{x}))] = \mathcal{E}_f(r_k) + \mathbb{E}_{p_k^{\text{tr}}}[f(r_k^*(\mathbf{x}))]$ and $\mathbb{E}_{p_k^{\text{tr}}}[f(r_k^*(\mathbf{x}))]$ is constant w.r.t. $r_k$. Let $\{\mathbf{x}_{k,i}^{\text{tr}}\}_{i=1}^{n_k^{\text{tr}}}$ and $\{\mathbf{x}_{l,j}^{\text{te}}\}_{j=1}^{n^{\text{te}}}$ denote unlabelled samples drawn i.i.d. from distributions $p_k^{\text{tr}}$ and $p_l^{\text{te}}$, respectively, for $l \in [K]$. A natural way to solve $\min_{r_k \in \mathcal{H}_r} \mathcal{E}_f(r_k)$ is to substitute empirical averages in Eq. (D.5) (Sugiyama et al., 2012), leading to BD-based DRM for FL: $\min_{r_k \in \mathcal{H}_r} \hat{\mathcal{E}}_f(r_k)$ where

$$\hat{\mathcal{E}}_f(r_k) = \frac{1}{n_k^{\text{tr}}} \sum_{i=1}^{n_k^{\text{tr}}} \left( \nabla f(r_k(\mathbf{x}_{k,i}^{\text{tr}})) r_k(\mathbf{x}_{k,i}^{\text{tr}}) - f(r_k(\mathbf{x}_{k,i}^{\text{tr}})) \right) - \frac{1}{n^{\text{te}}} \sum_{j=1}^{n^{\text{te}}} \sum_{l=1}^{K} \nabla f(r_k(\mathbf{x}_{l,j}^{\text{te}})).$$

## E  Communication costs and FITW-ERM

To estimate density ratios for FTW-ERM, clients require to send a few unlabelled test samples *only once*. The server shuffles those samples and broadcasts the shuffled version to clients *only once*. The communication overhead to estimate ratios is negligible compared to the communication costs for sharing high-dimensional stochastic gradients over the course of training.

Consider the example of CIFAR10 consisting of 32 by 32 images with 3 channels represented by 8 bits. If one shares 1000 unlabelled images[7], the communication amounts to sharing roughly $3 \times 10^6$ values each with 8 bits, i.e., $25 \times 10^6$ total communication bits or 3.1MB. In contrast, during training, the network size alone easily surpasses this size (e.g. the common ResNet-18 has 11 million parameters, each represented by a 32-bit floating point). Standard training of ResNet-18 requires $8 \times 10^4$ iterations and aggregations, which amounts to $2.816 \times 10^{13}$ total communicated bits per client, i.e., 3.5TB during training.

In other words, the number of communication bits needed during training in standard federated learning is usually many *orders of magnitudes* larger than the size of samples shared for estimating the ratios. To further

---

[7]A total number of 1000 images are shown to be sufficient to learn density ratios on CIFAR10 (Kato & Teshima, 2021)[10, Section 5.1].

reduce communication costs of density ratio estimation and gradient aggregation, compression methods such as quantization, sparsification, and local updating rules, can be used along with FTW-ERM *on the fly* (Alistarh et al., 2017).

Alternatively, to eliminate any privacy risks, clients may minimize the following surrogate objective, which we name FITW-ERM:

$$\min_{\mathbf{w}\in\mathbb{R}^d} \tilde{F}(\mathbf{w}) := \sum_{k=1}^{K} \tilde{F}_k(\mathbf{w}) \tag{E.1}$$

where $\tilde{F}_k(\mathbf{w}) = \frac{1}{n_k^{\mathrm{tr}}} \sum_{i=1}^{n_k^{\mathrm{tr}}} \frac{p_k^{\mathrm{te}}(\mathbf{x}_{k,i}^{\mathrm{tr}})}{p_k^{\mathrm{tr}}(\mathbf{x}_{k,i}^{\mathrm{tr}})} \ell(h_{\mathbf{w}}(\mathbf{x}_{k,i}^{\mathrm{tr}}), \mathbf{y}_{k,i}^{\mathrm{tr}})$.

We note that privacy risks are eliminated by solving E.1. However, to exploit the entire data distributed among all clients and achieve the optimal global model in terms of overall test error, clients need to compromise some level of privacy and share unlabelled samples generated from their test distribution with the server. Hence, in this paper, we focus on the original objective $F(\mathbf{w})$ in FTW-ERM, which is different from $\tilde{F}(\mathbf{w})$.

## F    Variants of nnBD

In this section, we show explicit expressions for unnormalized Kullback–Leibler (UKL), logistic regression (LR), and positive and unlabeled learning (PU) variants of nnBD.

Substituting $f(z) = z\log(z) - z$ into Eq. (D.2), we have $\ell_1(z) = z$ and $\ell_2(z) = zC - \log(z)$, and the UKL variant of nnBD is given by

$$\hat{\mathcal{E}}_{\mathrm{UKL}}^+(r) = \mathrm{ReLU}\Big(\frac{1}{n^{\mathrm{tr}}} \sum_{i=1}^{n^{\mathrm{tr}}} r(\mathbf{x}_i^{\mathrm{tr}}) - \frac{C}{n^{\mathrm{te}}} \sum_{j=1}^{n^{\mathrm{te}}} r(\mathbf{x}_j^{\mathrm{te}})\Big)$$
$$- \frac{1}{n^{\mathrm{te}}} \sum_{j=1}^{n^{\mathrm{te}}} \big( \log(r(\mathbf{x}_j^{\mathrm{te}})) - Cr(\mathbf{x}_j^{\mathrm{te}}) \big). \tag{F.1}$$

Substituting $f(z) = z\log(z) - (z+1)\log(z+1)$ into Eq. (D.2), we have $\ell_1(z) = \log(z+1)$ and $\ell_2(z) = C\log(z+1) + \log\left(\frac{z+1}{z}\right)$, and the LR (BKL) variant of nnBD is given by

$$\hat{\mathcal{E}}_{\mathrm{LR}}^+(r) = \mathrm{ReLU}\Big(\frac{1}{n^{\mathrm{tr}}} \sum_{i=1}^{n^{\mathrm{tr}}} \log(r(\mathbf{x}_i^{\mathrm{tr}}) + 1) - \frac{C}{n^{\mathrm{te}}} \sum_{j=1}^{n^{\mathrm{te}}} \log(r(\mathbf{x}_j^{\mathrm{te}}) + 1)\Big)$$
$$- \frac{1}{n^{\mathrm{te}}} \sum_{j=1}^{n^{\mathrm{te}}} \left( \log\left(\frac{r(\mathbf{x}_j^{\mathrm{te}})}{r(\mathbf{x}_j^{\mathrm{te}}) + 1}\right) - C\log(r(\mathbf{x}_j^{\mathrm{te}}) + 1) \right). \tag{F.2}$$

Substituting $f(z) = C\log(1-z) + Cz(\log(z) - \log(1-z))$ into Eq. (D.2), we have $\ell_1(z) = -C\log(1-z)$ and $\ell_2(z) = -C\log(z) + (C - C^2)\log(1-z)$, and the PU variant of nnBD is given by

$$\hat{\mathcal{E}}_{\mathrm{PU}}^+(r) = \mathrm{ReLU}\Big(\frac{-C}{n^{\mathrm{tr}}} \sum_{i=1}^{n^{\mathrm{tr}}} \log(1 - r(\mathbf{x}_i^{\mathrm{tr}})) + \frac{C^2}{n^{\mathrm{te}}} \sum_{j=1}^{n^{\mathrm{te}}} \log(1 - r(\mathbf{x}_j^{\mathrm{te}}))\Big)$$
$$- \frac{1}{n^{\mathrm{te}}} \sum_{j=1}^{n^{\mathrm{te}}} \big( C\log(r(\mathbf{x}_j^{\mathrm{te}})) - (C - C^2)\log(1 - r(\mathbf{x}_j^{\mathrm{te}})) \big). \tag{F.3}$$

## G    Convergence of $\tilde{r}$ and $k$-means clustering

**Lemma 2.** *If $n_k^{\mathrm{tr}}$, $n^{\mathrm{te}}$, and $M$ go to infinity with $\sup_m \mathrm{Vol}(\mathcal{B}_m) \to 0$, then $\tilde{r}_k \to \bar{r}_k$.*

*Proof.* Let $\mathbf{x} \in \mathcal{X}^{\text{tr}}$. Note that when $n_k^{\text{tr}}$, $n^{\text{te}}$, and $M$ go to infinity, the numerator and denominator of $\tilde{r}_k$ become $\sum_{l=1}^{K} p_l^{\text{te}}(\mathbf{x})\text{Vol}(\mathcal{B}_m)$ and $p_k^{\text{tr}}(\mathbf{x})\text{Vol}(\mathcal{B}_m)$, respectively, where $\mathbf{x} \in \mathcal{B}_m$. ∎

Please note that our density ratio in Eq. (3.1) is in the form of a sum of test densities over own train density. So even if one or a few number of ratios are poorly estimated, it will not impact the entire ratio in Eq. (3.1) as nested estimation errors. The error does not propagate in a multiplicative manner but in an additive way.

**$k$-means clustering for HDRM.** We note that partitioning the space and counting the number of samples in each bin is not necessarily an easy task when data is high dimensional. In practice, one simple method is to cluster train and test samples using an efficient implementation of $k$-means clustering with $M$ clusters and count the number of train and test samples in each cluster (Lloyd, 1982). To estimate the ratios, we need a batch of samples from the test distribution of each client in addition to a batch of samples from the train distribution for each estimating client. The running time of Lloyd's algorithm with $M$ clusters is $O(nd_{\mathbf{x}}M)$ where $n$ is the total number of samples with dimension $d_{\mathbf{x}}$.

## H UKL, LR, and PU variants of nnBD for multiple clients

In this section, we provide explicit expressions for UKL, LR, and PU variants of nnBD for client $k$.

The UKL variant of nnBD for client $k$ is given by $\min_{r_k \in \mathcal{H}_r} \hat{\mathcal{E}}_{\text{UKL}}^+(r_k)$ where

$$\hat{\mathcal{E}}_{\text{UKL}}^+(r_k) = \text{ReLU}\Big( \frac{1}{n_k^{\text{tr}}} \sum_{i=1}^{n_k^{\text{tr}}} r_k(\mathbf{x}_{k,i}^{\text{tr}}) - \frac{C_k}{n^{\text{te}}} \sum_{j=1}^{n^{\text{te}}} \sum_{l=1}^{K} r_k(\mathbf{x}_{l,j}^{\text{te}}) \Big)$$
$$- \frac{1}{n^{\text{te}}} \sum_{j=1}^{n^{\text{te}}} \sum_{l=1}^{K} \big( \log(r_k(\mathbf{x}_{l,j}^{\text{te}})) - C_k r_k(\mathbf{x}_{l,j}^{\text{te}}) \big). \tag{H.1}$$

The LR variant of nnBD for client $k$ is given by $\min_{r_k \in \mathcal{H}_r} \hat{\mathcal{E}}_{\text{LR}}^+(r_k)$ where

$$\hat{\mathcal{E}}_{\text{LR}}^+(r_k) = \text{ReLU}\Big( \frac{1}{n_k^{\text{tr}}} \sum_{i=1}^{n_k^{\text{tr}}} \log(r_k(\mathbf{x}_{k,i}^{\text{tr}}) + 1) - \frac{C_k}{n^{\text{te}}} \sum_{j=1}^{n^{\text{te}}} \sum_{l=1}^{K} \log(r_k(\mathbf{x}_{l,j}^{\text{te}}) + 1) \Big)$$
$$- \frac{1}{n^{\text{te}}} \sum_{j=1}^{n^{\text{te}}} \sum_{l=1}^{K} \left( \log\left( \frac{r_k(\mathbf{x}_{l,j}^{\text{te}})}{r_k(\mathbf{x}_{l,j}^{\text{te}}) + 1} \right) - C_k \log(r_k(\mathbf{x}_{l,j}^{\text{te}}) + 1) \right). \tag{H.2}$$

The PU variant of nnBD for client $k$ is given by $\min_{r_k \in \mathcal{H}_r} \hat{\mathcal{E}}_{\text{PU}}^+(r_k)$ where

$$\hat{\mathcal{E}}_{\text{PU}}^+(r_k) = \text{ReLU}\Big( \frac{-C_k}{n_k^{\text{tr}}} \sum_{i=1}^{n_k^{\text{tr}}} \log(1 - r_k(\mathbf{x}_{k,i}^{\text{tr}})) + \frac{C_k^2}{n^{\text{te}}} \sum_{j=1}^{n^{\text{te}}} \sum_{l=1}^{K} \log(1 - r_k(\mathbf{x}_{l,j}^{\text{te}})) \Big)$$
$$- \frac{1}{n^{\text{te}}} \sum_{j=1}^{n^{\text{te}}} \sum_{l=1}^{K} \big( C_k \log(r_k(\mathbf{x}_{l,j}^{\text{te}})) - (C_k - C_k^2) \log(1 - r_k(\mathbf{x}_{l,j}^{\text{te}})) \big). \tag{H.3}$$

## I Proof of Theorem 1

In this section, we prove Theorem 1, which establishes an upper bound on the ratio estimation error of nnBD DRM (HDRM method with an arbitrary $f$) for client $k$ in terms of BD risk, which holds with high probability along the lines of (Kiryo et al., 2017; Lu et al., 2020; Kato & Teshima, 2021).

We remind that client $k$'s goal is to estimate this ratio:

$$r_k(\mathbf{x}) = \frac{\sum_{l=1}^{K} p_l^{\text{te}}(\mathbf{x})}{p_k^{\text{tr}}(\mathbf{x})}. \tag{I.1}$$

For client $k$, the BD risk is given by

$$\mathcal{E}_f(r_k) = \tilde{\mathbb{E}}_k(\mathbf{x})[\ell_1(r_k(\mathbf{x}))] + \sum_{l=1}^{K} \mathbb{E}_{p_l^{\text{te}}}[\ell_2(r_k(\mathbf{x}))] \tag{I.2}$$

where $\tilde{\mathbb{E}}_k := \mathbb{E}_{p_k^{\text{tr}}} - C_k \sum_{l=1}^{K} \mathbb{E}_{p_l^{\text{te}}}$, $0 < C_k < \frac{1}{\bar{r}_k}$, $\bar{r}_k = \sup_{\mathbf{x} \in \mathcal{X}^{\text{tr}}} = \frac{\sum_{l=1}^{K} p_l^{\text{te}}(\mathbf{x})}{p_k^{\text{tr}}(\mathbf{x})}$, $\ell_1(z) = \nabla f(z)z - f(z)$, and $\ell_2(z) = C(\nabla f(z)z - f(z)) - \nabla f(z)$. We note that the definition of $C_k$ implies $\tilde{p}_k = p_k^{\text{tr}} - C_k \sum_{l=1}^{K} p_l^{\text{te}} > 0$. We remind that $f : \mathcal{B}_f \to \mathbb{R}$ is a strictly convex function with bounded gradient $\nabla f$ where $\mathcal{B}_f \subset [0, \infty)$, and $\mathcal{H}_r \subset \{r : \mathcal{X} \to \mathcal{B}_f\}$ denotes a hypothesis class for our model $r$.

The nnBD DRM problem for client $k$ is $\min_{r_k \in \mathcal{H}_r} \hat{\mathcal{E}}_f^+(r_k)$ where

$$\hat{\mathcal{E}}_f^+(r_k) = \text{ReLU}\Big( (\hat{\mathbb{E}}_{p_k^{\text{tr}}} - C_k \sum_{l=1}^{K} \hat{\mathbb{E}}_{p_l^{\text{te}}})[\ell_1(r_k(\mathbf{x}))] \Big) + \sum_{l=1}^{K} \hat{\mathbb{E}}_{p_l^{\text{te}}}[\ell_2(r_k(\mathbf{x}))] \tag{I.3}$$

with $\hat{\mathbb{E}}_{p_k^{\text{tr}}}$ is the sample average over $\{\mathbf{x}_{k,i}^{\text{tr}}\}_{i=1}^{n_k^{\text{tr}}}$, and $\hat{\mathbb{E}}_{p_l^{\text{te}}}$ is the sample average over $\{\mathbf{x}_{l,j}^{\text{te}}\}_{j=1}^{n^{\text{te}}}$. In the following, we denote $\hat{\mathbb{E}}_k := \hat{\mathbb{E}}_{p_k^{\text{tr}}} - C_k \sum_{l=1}^{K} \hat{\mathbb{E}}_{p_l^{\text{te}}}$ for notational simplicity.

Let $\hat{r}_k := \arg\min_{r_k \in \mathcal{H}_r} \hat{\mathcal{E}}_f^+(r_k)$ and $r_k^* := \arg\min_{r_k \in \mathcal{H}_r} \mathcal{E}_f(r_k)$. We first decompose the ratio estimation error into maximal deviation and bias terms:

$$\begin{aligned}
\mathcal{E}_f(\hat{r}_k) - \mathcal{E}_f(r_k^*) &\leq \mathcal{E}_f(\hat{r}_k) - \hat{\mathcal{E}}_f^+(\hat{r}_k) + \hat{\mathcal{E}}_f^+(\hat{r}_k) - \mathcal{E}_f(r_k^*) \\
&\leq \mathcal{E}_f(\hat{r}_k) - \hat{\mathcal{E}}_f^+(\hat{r}_k) + \hat{\mathcal{E}}_f^+(r_k^*) - \mathcal{E}_f(r_k^*) \\
&\leq 2 \sup_{r_k \in \mathcal{H}_r} |\mathcal{E}_f(r_k) - \hat{\mathcal{E}}_f^+(r_k)| \\
&\leq 2 \sup_{r_k \in \mathcal{H}_r} |\hat{\mathcal{E}}_f^+(r_k) - \mathbb{E}[\hat{\mathcal{E}}_f^+(r_k)]| + 2 \sup_{r_k \in \mathcal{H}_r} |\mathbb{E}[\hat{\mathcal{E}}_f^+(r_k)] - \mathcal{E}_f(r_k)|
\end{aligned} \tag{I.4}$$

where the second inequality holds since $\hat{r}_k := \arg\min_{r_k \in \mathcal{H}_r} \hat{\mathcal{E}}_f^+(r_k)$. The first term in the RHS of (I.4) is the maximal derivation and the second term is the bias.

In the following two lemmas, we find an upper bound on the maximal deviation $\sup_{r_k \in \mathcal{H}_r} |\hat{\mathcal{E}}_f^+(r_k) - \mathbb{E}[\hat{\mathcal{E}}_f^+(r_k)]|$ and bias $\sup_{r_k \in \mathcal{H}_r} |\mathbb{E}[\hat{\mathcal{E}}_f^+(r_k)] - \mathcal{E}_f(r_k)|$, respectively.

**Lemma 3** (Maximal deviation bound). *Denote $\Delta_\ell := \sup_{z \in \mathcal{B}_f} \max_{i \in \{1,2\}} |\ell_i(z)|$, then for any $0 < \delta < 1$, the maximal deviation term is upper bounded with probability at least $1 - \delta$*

$$\begin{aligned}
\sup_{r_k \in \mathcal{H}_r} |\hat{\mathcal{E}}_f^+(r_k) - \mathbb{E}[\hat{\mathcal{E}}_f^+(r_k)]| &\leq 4L_1 R_{n_k^{\text{tr}}}^{p_k^{\text{tr}}}(\mathcal{H}_r) + 4(C_k L_1 + L_2) \sum_{l=1}^{K} R_{n^{\text{te}}}^{p_l^{\text{te}}}(\mathcal{H}_r) \\
&\quad + \Delta_\ell \sqrt{2\Big(\frac{1}{n_k^{\text{tr}}} + \frac{K(1 + C_k)^2}{n^{\text{te}}}\Big) \log \frac{1}{\delta}}.
\end{aligned} \tag{I.5}$$

*Proof.* Denote $\Phi(\mathcal{S}_k) := \sup_{r_k \in \mathcal{H}_r} |\hat{\mathcal{E}}_f^+(r_k) - \mathbb{E}[\hat{\mathcal{E}}_f^+(r_k)]|$ with $\mathcal{S}_k = \{\mathbf{x}_{k,1}^{\text{tr}}, \ldots, \mathbf{x}_{n_k^{\text{tr}},1}^{\text{tr}}, \mathbf{x}_{1,1}^{\text{te}}, \ldots, \mathbf{x}_{K,n^{\text{te}}}^{\text{te}}\}$. Let $\mathcal{S}_k^{(i)}$ be obtained by replacing element $i$ of set $\mathcal{S}_k$ by an independent data point taking values from the set $\mathcal{X}^{\text{tr}}$. We now measure the absolute value of the difference caused by changing one data point in the maximal deviation term (I.5), i.e., $|\Phi(\mathcal{S}_k) - \Phi(\mathcal{S}_k^{(i)})|$. If the changed point is sampled from $p_k^{\text{tr}}$, then the absolute value of the difference caused in the maximal deviation term is upper bounded by $\frac{2\Delta_\ell}{n_k^{\text{tr}}}$. If the changed point is sampled from $p_l^{\text{te}}$, the the absolute value of the difference caused in the maximal deviation term is upper bounded by $\frac{2\Delta_\ell(C_k+1)}{n^{\text{te}}}$ for $l = 1, \ldots, K$. Applying McDiarmid's inequality (McDiarmid et al., 1989), with probability at

least $1 - \delta$, we have

$$\sup_{r_k \in \mathcal{H}_r} |\hat{\mathcal{E}}_f^+(r_k) - \mathbb{E}[\hat{\mathcal{E}}_f^+(r_k)]| \leq \mathbb{E}[\sup_{r_k \in \mathcal{H}_r} |\hat{\mathcal{E}}_f^+(r_k) - \mathbb{E}[\hat{\mathcal{E}}_f^+(r_k)]|]$$
$$+ \Delta_\ell \sqrt{2 \Big( \frac{1}{n_k^{\mathrm{tr}}} + \frac{K(1 + C_k)^2}{n^{\mathrm{te}}} \Big) \log \frac{1}{\delta}}.$$

In the following, we establish an upper bound on the expected maximal deviation $\mathbb{E}[\sup_{r_k \in \mathcal{H}_r} |\hat{\mathcal{E}}_f^+(r_k) - \mathbb{E}[\hat{\mathcal{E}}_f^+(r_k)]|]$ by generalization the symmetrization argument in (Kiryo et al., 2017; Lu et al., 2020) followed by applying Talagrand's contraction lemma for two-sided Rademacher complexity.

Let $m \in [M]$ and $N_m \in \mathbb{Z}_+$ for $M \in \mathbb{Z}_+$. Let $g_m : \mathbb{R} \to \mathbb{R}$ be a $L_{g_m}$-Lipschitz function. Let $p_{m,p}$ denote a probability distribution over $\mathcal{X}^{\mathrm{tr}}$. Suppose that $\{\mathbf{x}_i\}_{i=1}^{n_{m,p}}$ are drawn i.i.d. from $p_{m,p}$ for $p \in [N_m]$ and $m \in [M]$. Let $\ell_{m,p} : \mathcal{B}_f \to \mathbb{R}_+$ be a $L_{m,p}$-Lipschitz function and $\tilde{C}_{m,p}$ be a constant $\forall\, m, p$. Consider the following stochastic process:

$$\hat{R}_k(r_k) := \sum_{m=1}^{M} g_m \Big( \sum_{p=1}^{N_m} \tilde{C}_{m,p} \hat{\mathbb{E}}_{m,p}[\ell_{(m,p)}(r_k(\mathbf{x}))] \Big)$$

where $\hat{\mathbb{E}}_{m,p}$ denotes sample average over $\{\mathbf{x}_i\}_{i=1}^{n_{m,p}}$. In the rest of the proof, we show that

$$\mathbb{E}[\sup_{r_k \in \mathcal{H}_r} |\hat{R}_k(r_k) - \mathbb{E}[\hat{R}_k(r_k)]|] \leq 4 \sum_{m=1}^{M} \sum_{p=1}^{N_m} L_{g_m} |\tilde{C}_{m,p}| L_{m,p} R_{n_{m,p}}^{p_{m,p}}(\mathcal{H}_r). \tag{I.6}$$

To prove (I.6), we consider a continuous extension of $\ell_{(m,p)}$ defined on the origin. We note that such extension does not change $\hat{R}_k(r_k)$ since $\ell_{(m,p)}$ takes values only in $\mathcal{B}_f$. If $\mathcal{B}_f = \{(z_1, z_2)\}$ for some $0 \leq z_1 < z_2$, then for any $z \in [0, z_1]$, we define $\ell_{(m,p)}(z) = \lim_{z \downarrow z_1} \ell_{(m,p)}(z)$ where $\lim_{z \downarrow z_1} \ell_{(m,p)}(z)$ exists since $\ell_{(m,p)}$ is uniformly continuous due to Lipschitz continuity. Then $\ell_{(m,p)}$ will be $L_{m,p}$-Lipschitz on $z \in [0, z_2]$. Let $\{\tilde{\mathbf{x}}_i\}_{i=1}^{n_{m,p}}$ be an independent copy of $\{\mathbf{x}_i\}_{i=1}^{n_{m,p}}$. Let denote $\delta_{\hat{R}} := \mathbb{E}[\sup_{r_k \in \mathcal{H}_r} |\hat{R}_k(r_k) - \mathbb{E}[\hat{R}_k(r_k)]|]$. Following a symmetrization argument (Vapnik, 1999), an upper bound on the symmetrized process can be established by Rademacher complexity:

$$
\begin{aligned}
\delta_{\hat{R}} &\leq \mathbb{E}\left[ \sup_{r_k \in \mathcal{H}_r} \sum_{m=1}^{M} \Big| g_m \Big( \sum_{p=1}^{N_m} \tilde{C}_{m,p} \hat{\mathbb{E}}_{m,p}[\ell_{(m,p)}(r_k(\mathbf{x}))] \Big) - \tilde{\mathbb{E}}\Big[ g_m \Big( \sum_{p=1}^{N_m} \tilde{C}_{m,p} \hat{\tilde{\mathbb{E}}}_{m,p}[\ell_{(m,p)}(r_k(\mathbf{x}))] \Big) \Big] \Big| \right] \\
&\leq \mathbb{E}\tilde{\mathbb{E}}\left[ \sup_{r_k \in \mathcal{H}_r} \sum_{m=1}^{M} \Big| g_m \Big( \sum_{p=1}^{N_m} \tilde{C}_{m,p} \hat{\mathbb{E}}_{m,p}[\ell_{(m,p)}(r_k(\mathbf{x}))] \Big) - g_m \Big( \sum_{p=1}^{N_m} \tilde{C}_{m,p} \hat{\tilde{\mathbb{E}}}_{m,p}[\ell_{(m,p)}(r_k(\mathbf{x}))] \Big) \Big| \right] \\
&\leq \sum_{m=1}^{M} L_{g_m} \sum_{p=1}^{N_m} |\tilde{C}_{m,p}| \mathbb{E}\tilde{\mathbb{E}}\left[ \sup_{r_k \in \mathcal{H}_r} \Big| \hat{\mathbb{E}}_{m,p}[\ell_{(m,p)}(r_k(\mathbf{x}))] - \hat{\tilde{\mathbb{E}}}_{m,p}[\ell_{(m,p)}(r_k(\mathbf{x}))] \Big| \right] \\
&= \sum_{m=1}^{M} L_{g_m} \sum_{p=1}^{N_m} |\tilde{C}_{m,p}| \mathbb{E}\tilde{\mathbb{E}}\Big[ \sup_{r_k \in \mathcal{H}_r} \Big| \hat{\mathbb{E}}_{m,p}[\ell_{(m,p)}(r_k(\mathbf{x})) - \ell_{(m,p)}(0)] - \hat{\tilde{\mathbb{E}}}_{m,p}[\ell_{(m,p)}(r_k(\mathbf{x})) - \ell_{(m,p)}(0)] \Big| \Big] \\
&\leq 4 \sum_{m=1}^{M} L_{g_m} \sum_{p=1}^{N_m} |\tilde{C}_{m,p}| \mathbb{E}\left[ \sup_{r_k \in \mathcal{H}_r} \Big| \hat{\mathbb{E}}_{m,p}[\sigma_{m,p}(\ell_{(m,p)}(r_k(\mathbf{x})) - \ell_{(m,p)}(0))] \Big| \right] \\
&\leq 4 \sum_{m=1}^{M} L_{g_m} \sum_{p=1}^{N_m} |\tilde{C}_{m,p}| R_{n_{m,p}}^{p_{m,p}}(\mathcal{H}_r)
\end{aligned}
\tag{I.7}
$$

where $\sigma_{m,p}$ are Rademacher variables uniformly chosen from $\{-1, 1\}$, $\tilde{\mathbb{E}}$ and $\hat{\tilde{\mathbb{E}}}_{m,p}$ denote the expectation and sample average over data distribution $p_{m,p}$ and the independent copy $\{\tilde{\mathbf{x}}_i\}_{i=1}^{n_{m,p}}$, respectively, the third

inequality holds by the Lipschitz continuous property of $g_m$, and the last inequality is obtained by applying Talagrand's contraction lemma for two-sided Rademacher complexity (Ledoux & Talagrand, 1991; Bartlett & Mendelson, 2002).

Applying (I.6), we can show that

$$\mathbb{E}[\sup_{r_k \in \mathcal{H}_r} |\hat{\mathcal{E}}_f^+(r_k) - \mathbb{E}[\hat{\mathcal{E}}_f^+(r_k)]|] \leq 4L_1 R_{n_k^{\mathrm{tr}}}^{p_k^{\mathrm{tr}}}(\mathcal{H}_r) + 4(C_k L_1 + L_2) \sum_{l=1}^{K} R_{n^{\mathrm{te}}}^{p_l^{\mathrm{te}}}(\mathcal{H}_r),$$

which completes the proof. ∎

Next we find an upper bound on the bias $\sup_{r_k \in \mathcal{H}_r} |\mathbb{E}[\hat{\mathcal{E}}_f^+(r_k)] - \mathcal{E}_f(r_k)|$.

**Lemma 4** (Bias bound). *Denote $\Delta_\ell := \sup_{z \in \mathcal{B}_f} \max_{i \in \{1,2\}} |\ell_i(z)|$. Assume $\inf_{r \in \mathcal{H}_r} \mathbb{E}[\hat{\mathbb{E}}_k[\ell_1(r_k(\mathbf{x}))]] > 0$ for $k \in [K]$. Then, an upper bound on the bias term is given by*

$$\sup_{r_k \in \mathcal{H}_r} |\mathbb{E}[\hat{\mathcal{E}}_f^+(r_k)] - \mathcal{E}_f(r_k)| \leq (1 + KC_k)\Delta_\ell \exp\left(\frac{-2\eta_k^2}{\Delta_\ell^2/n_k^{\mathrm{tr}} + KC_k^2\Delta_\ell^2/n^{\mathrm{te}}}\right) \qquad (\mathrm{I.8})$$

*for some constant $\eta_k > 0$.*

*Proof.* Let $\hat{\bar{\mathbb{E}}}_k := \hat{\mathbb{E}}_{p_k^{\mathrm{tr}}} - C_k \sum_{l=1}^{K} \hat{\mathbb{E}}_{p_l^{\mathrm{te}}}$. We first note that

$$
\begin{aligned}
|\mathbb{E}[\hat{\mathcal{E}}_f^+(r_k)] - \mathcal{E}_f(r_k)| &= |\mathbb{E}[\hat{\mathcal{E}}_f^+(r_k) - \hat{\mathcal{E}}_f(r_k)]| \\
&= \left| \mathbb{E}\left[ \mathrm{ReLU}\left(\hat{\bar{\mathbb{E}}}_k[\ell_1(r_k(\mathbf{x}))]\right) - \hat{\bar{\mathbb{E}}}_k[\ell_1(r_k(\mathbf{x}))] \right] \right| \\
&\leq \mathbb{E}\left[ \left| \mathrm{ReLU}\left(\hat{\bar{\mathbb{E}}}_k[\ell_1(r_k(\mathbf{x}))]\right) - \hat{\bar{\mathbb{E}}}_k[\ell_1(r_k(\mathbf{x}))] \right| \right] \\
&= \mathbb{E}\left[ 1\left\{ \mathrm{ReLU}\left(\hat{\bar{\mathbb{E}}}_k[\ell_1(r_k(\mathbf{x}))]\right) \neq \hat{\bar{\mathbb{E}}}_k[\ell_1(r_k(\mathbf{x}))] \right\} \right] \\
&\quad \cdot \left| \mathrm{ReLU}\left(\hat{\bar{\mathbb{E}}}_k[\ell_1(r_k(\mathbf{x}))]\right) - \hat{\bar{\mathbb{E}}}_k[\ell_1(r_k(\mathbf{x}))] \right| \\
&= \mathbb{E}\left[ 1\left\{ \mathrm{ReLU}\left(\hat{\bar{\mathbb{E}}}_k[\ell_1(r_k(\mathbf{x}))]\right) \neq \hat{\bar{\mathbb{E}}}_k[\ell_1(r_k(\mathbf{x}))] \right\} \right] \sup_{z:|z| \leq (1+KC_k)\Delta_\ell} (\mathrm{ReLU}(z) - z)
\end{aligned}
$$

where the third inequality holds due to Jensen's inequality.

We note that $\hat{\bar{\mathbb{E}}}_k[\ell_1(r_k(\mathbf{x}))] \leq (1 + KC_k)\Delta_\ell$ implies

$$\sup_{z:|z| \leq (1+KC_k)\Delta_\ell} (\mathrm{ReLU}(z) - z) \leq (1 + KC_k)\Delta_\ell.$$

Due to the assumption $\inf_{r \in \mathcal{H}_r} \mathbb{E}[\hat{\mathbb{E}}_k[\ell_1(r_k(\mathbf{x}))]] > 0$, there exists an $\eta_k > 0$ such that $\mathbb{E}[\hat{\mathbb{E}}_k[\ell_1(r_k(\mathbf{x}))]] \geq \eta_k$ for all $r_k \in \mathcal{H}_r$. Then we have

$$
\begin{aligned}
\mathbb{E}\left[ 1\left\{ \mathrm{ReLU}\left(\hat{\bar{\mathbb{E}}}_k[\ell_1(r_k(\mathbf{x}))]\right) \neq \hat{\bar{\mathbb{E}}}_k[\ell_1(r_k(\mathbf{x}))] \right\} \right] &= \Pr\left\{ \hat{\bar{\mathbb{E}}}_k[\ell_1(r_k(\mathbf{x}))] \in \mathrm{supp}(\widetilde{\mathrm{ReLU}}) \right\} \\
&= \Pr\left\{ \hat{\bar{\mathbb{E}}}_k[\ell_1(r_k(\mathbf{x}))] < 0 \right\} \\
&= \Pr\left\{ \hat{\bar{\mathbb{E}}}_k[\ell_1(r_k(\mathbf{x}))] < \mathbb{E}[\hat{\bar{\mathbb{E}}}_k[\ell_1(r_k(\mathbf{x}))]] - \eta_k \right\}
\end{aligned}
$$

where $\widetilde{\mathrm{ReLU}}(z) = \mathrm{ReLU}(z) - z$.

Denote $\tilde{\Phi}(\mathcal{S}_k) := \hat{\bar{\mathbb{E}}}_k[\ell_1(r_k(\mathbf{x}))]$ where $\mathcal{S}_k = \{\mathbf{x}_{k,1}^{\mathrm{tr}}, \ldots, \mathbf{x}_{n_k^{\mathrm{tr}},1}^{\mathrm{tr}}, \mathbf{x}_{1,1}^{\mathrm{te}}, \ldots, \mathbf{x}_{K,n^{\mathrm{te}}}^{\mathrm{te}}\}$. Let $\mathcal{S}_k^{(i)}$ be obtained by replacing element $i$ of set $\mathcal{S}_k$ by an independent data point taking values from the set $\mathcal{X}^{\mathrm{tr}}$. We now measure

the absolute value of the difference caused by changing one data point in $|\tilde{\Phi}(\mathcal{S}_k) - \tilde{\Phi}(\mathcal{S}_k^{(i)})|$. If the changed point is sampled from $p_k^{\text{tr}}$, the the absolute value of the difference caused in the maximal deviation term is upper bounded by $\frac{\Delta_\ell}{n_k^{\text{tr}}}$. If the changed point is sampled from $p_l^{\text{te}}$, the the absolute value of the difference caused in the maximal deviation term is upper bounded by $\frac{\Delta_\ell C_k}{n^{\text{te}}}$ for $l = 1, \dots, K$. Finally, McDiarmid's inequality (McDiarmid et al., 1989) implies:

$$\Pr\left\{\hat{\bar{\mathbb{E}}}_k[\ell_1(r_k(\mathbf{x}))] < \mathbb{E}[\hat{\bar{\mathbb{E}}}_k[\ell_1(r_k(\mathbf{x}))]] - \eta_k\right\} \le \exp\left(\frac{-2\eta_k^2}{\Delta_\ell^2/n_k^{\text{tr}} + KC_k^2\Delta_\ell^2/n^{\text{te}}}\right),$$

which completes the proof. ∎

Substituting the upper bounds in (I.5) and (I.8) into (I.4), with probability at least $1 - \delta$, we have

$$\mathcal{E}_f(\hat{r}_k) - \mathcal{E}_f(r_k^*) \le 8L_1 R_{n_k^{\text{tr}}}^{p_k^{\text{tr}}}(\mathcal{H}_r) + \Psi(\delta, \Delta_\ell, n_k^{\text{tr}}, n^{\text{te}}) + 8(C_k L_1 + L_2)\sum_{l=1}^{K} R_{n^{\text{te}}}^{p_l^{\text{te}}}(\mathcal{H}_r) \qquad (\text{I.9})$$

where $\Psi = \Delta_\ell\sqrt{8\left(\frac{1}{n_k^{\text{tr}}} + \frac{K(1+C_k)^2}{n^{\text{te}}}\right)\log\frac{1}{\delta}} + 2(1 + KC_k)\Delta_\ell\exp\left(\frac{-2\eta_k^2}{\Delta_\ell^2/n_k^{\text{tr}} + KC_k^2\Delta_\ell^2/n^{\text{te}}}\right)$ for some constant $\eta_k > 0$. This completes the proof.

## J  Ratio estimation error bound for multi-layer perceptron and multiple clients

Our high-probability ratio estimation error bound for client $k$ depends on the Rademacher complexity of the hypothesis class for our density ratio model $\mathcal{H}_r \subset \{r : \mathcal{X} \to \mathcal{B}_f\}$ w.r.t. client $k$ train distribution $p_k^{\text{tr}}$ and all client's test distributions $p_l^{\text{te}}$ for $l \in [K]$. By restricting a function class for density ratios and substituting an upper bounds on its Rademacher complexity, we can obtain explicit ratio estimation error bounds in terms of $n_k^{\text{tr}}, n^{\text{te}}$ in a special case. As an example, the following corollary establishes a ratio estimation error bound for multi-layer perceptron density ratio models in terms of the Frobenius norms of weight matrices.

**Example J.1** (Complexity for multi-layer perceptron class (Golowich et al., 2018))**.** Assume that distribution $p$ has a bounded support $S_p := \sup_{\mathbf{x}\in\text{supp}(p)} \|\mathbf{x}\| < \infty$. Let $\mathcal{H}$ be the class of real-valued neural networks with depth $L$ over the domain $\mathcal{X}^{\text{tr}}$, $\mathbf{W}_i$ be the network weight matrix $i$. Suppose that each weight matrix has a bounded Frobenius norm $\|\mathbf{W}_i\|_F \le \Delta_{\mathbf{W}_i}$ for $i \in [L]$ and the activation $\phi$ is 1-Lipschitz, and positive-homogeneous function, *i.e.,* $\phi(\alpha z) = \alpha\phi(z)$, which is applied element-wise. Then we have

$$R_n^p(\mathcal{H}) \le \frac{S_p(\sqrt{2L\log 2} + 1)\prod_{i=1}^{L}\Delta_{\mathbf{W}_i}}{\sqrt{n}}.$$

*Remark* 6. To control the upper bound $\Delta_{\mathbf{W}_i}$ for $i \in [L]$, it is natural to employ the sparsity of the weights, *e.g.,* (Golowich et al., 2018, Section 4) and (Hanin & Rolnick, 2019). We consider a special network architecture where $\text{diag}(\mathbf{W}_i)$'s are close to 1-sparse unit vectors for $i \in [L]$, which implies that the matrices $\mathbf{W}_i$'s will be *almost* rank-1. Then $\|\mathbf{W}_i\|_F$ is upper bounded by 1 for $i \in [L]$.

**Corollary 1** (High-probability ratio estimation error bound under Example J.1)**.** *For Example J.1 and loss functions described in Theorem 1, with probability at least $1 - \delta$, we have*

$$\mathcal{E}_f(\hat{r}_k) - \mathcal{E}_f(r_k^*) \le \frac{K_k^{\text{tr}}}{\sqrt{n_k^{\text{tr}}}} + \sum_{l=1}^{K}\frac{K_l^{\text{te}}}{\sqrt{n^{\text{te}}}} + \Psi(\delta, \Delta_\ell, n_k^{\text{tr}}, n^{\text{te}})$$

*where* $K_k^{\text{tr}} = O(L_1 S_{p_k^{\text{tr}}}\sqrt{L}\prod_{i=1}^{L}\Delta_{\mathbf{W}_i})$, $K_l^{\text{te}} = O(\max\{L_1, L_2\}S_{p_l^{\text{te}}}\sqrt{L}\prod_{i=1}^{L}\Delta_{\mathbf{W}_i})$, *and* $\Psi = \Delta_\ell\sqrt{8\left(\frac{1}{n_k^{\text{tr}}} + \frac{K(1+C_k)^2}{n^{\text{te}}}\right)\log\frac{1}{\delta}} + 2(1 + KC_k)\Delta_\ell\exp\left(\frac{-2\eta_k^2}{\Delta_\ell^2/n_k^{\text{tr}} + KC_k^2\Delta_\ell^2/n^{\text{te}}}\right)$ *for some constant* $\eta_k > 0$.

Finally, we apply union bound and obtain a global ratio estimation error bound that holds for all clients:

**Corollary 2** (High-probability ratio estimation error bound for multiple clients)**.** *Let $0 < \delta_k < 1$ for $k \in [K]$. Let $\overline{K}^{\mathrm{tr}} = \max_{k \in [K]} K_k^{\mathrm{tr}}$. For Example J.1 and loss functions described in Theorem 1, with probability at least $1 - \sum_{k=1}^{K} \delta_k$, we have*

$$\max_{k \in [K]} \{ \mathcal{E}_f(\hat{r}_k) - \mathcal{E}_f(r_k^*) \} \leq \frac{\overline{K}^{\mathrm{tr}}}{\sqrt{\underline{n}^{\mathrm{tr}}}} + \sum_{l=1}^{K} \frac{K_l^{\mathrm{te}}}{\sqrt{n^{\mathrm{te}}}}$$
$$+ \overline{\Psi}(\delta, \Delta_\ell, \underline{n}^{\mathrm{tr}}, n^{\mathrm{te}})$$

*where $\overline{\Psi} = \Delta_\ell \sqrt{8(\frac{1}{\underline{n}^{\mathrm{tr}}} + \frac{K(1+\overline{C})^2}{n^{\mathrm{te}}}) \log \frac{1}{\underline{\delta}}} + 2(1 + K\overline{C})\Delta_\ell \exp\left(\frac{-2\underline{\eta}^2}{\Delta_\ell^2 / \underline{n}^{\mathrm{tr}} + K\overline{C}^2 \Delta_\ell^2 / n^{\mathrm{te}}}\right)$, $\overline{C} = \max_{k \in [K]} C_k$, $\underline{n}^{\mathrm{tr}} = \min_{k \in [K]} n_k^{\mathrm{tr}}$, $\underline{\delta} = \min_{k \in [K]} \delta_k$, and $\underline{\eta} = \min_{k \in [K]} \eta_k$.*

The rates match the optimal minimax rates for example for a density estimation problem when the density belongs to the Hölder function class (Tsybakov, 2008)[Section 2] with a sufficiently large $\beta$ based on Definition 1.2 of Tsybakov (2008). The $\Omega(1/\sqrt{n})$ lower bounds are obtained for important problems including nonparametric regression, estimation of functionals, nonparametric testing, and finding a linear combination of $M$ functions to be as close as the target data generating function (Nemirovski, 1998)[Section 5.3].

# K   Additional error due to estimation of $\overline{r}_k$

In this section, we consider a practical scenario where we have access to only in imperfect estimate of $\overline{r}_k = \sup_{\mathbf{x} \in \mathcal{X}^{\mathrm{tr}}} r_k^*(\mathbf{x})$ to find $C_k$ in Eq. (3.2). In particular, we find additional error when using $\tilde{C}_k = \frac{1}{\tilde{r}_k}$ where $\tilde{r}_k$ is obtained by HDRM in Section 3. The nnBD DRM problem for client $k$ using $\tilde{C}_k$ is $\min_{r_k \in \mathcal{H}_r} \hat{\mathcal{E}}_f^+(r_k)$ where

$$\hat{\mathcal{E}}_f^+(r_k) = \mathrm{ReLU}\Big( (\hat{\mathbb{E}}_{p_k^{\mathrm{tr}}} - \tilde{C}_k \sum_{l=1}^{K} \hat{\mathbb{E}}_{p_l^{\mathrm{te}}})[\ell_1(r_k(\mathbf{x}))] \Big) + \sum_{l=1}^{K} \hat{\mathbb{E}}_{p_l^{\mathrm{te}}}[\ell_2(r_k(\mathbf{x}))]. \tag{K.1}$$

Along the lines of the proof of Lemma 3, we can show that the maximal deviation term using $\tilde{C}_k$ is upper bounded with probability at least $1 - \delta$:

$$\sup_{r_k \in \mathcal{H}_r} |\hat{\mathcal{E}}_f^+(r_k) - \mathbb{E}[\hat{\mathcal{E}}_f^+(r_k)]| \leq 4L_1 R_{n_k^{\mathrm{tr}}}^{p_k^{\mathrm{tr}}}(\mathcal{H}_r) + 4(\tilde{C}_k L_1 + L_2) \sum_{l=1}^{K} R_{n^{\mathrm{te}}}^{p_l^{\mathrm{te}}}(\mathcal{H}_r)$$
$$+ \Delta_\ell \sqrt{2\Big(\frac{1}{n_k^{\mathrm{tr}}} + \frac{K(1 + \tilde{C}_k)^2}{n^{\mathrm{te}}}\Big) \log \frac{1}{\delta}}. \tag{K.2}$$

Under perfect estimate of $\overline{r}_k = \sup_{\mathbf{x} \in \mathcal{X}^{\mathrm{tr}}} r_k^*(\mathbf{x})$ with $C_k = \frac{1}{\overline{r}_k}$, the nnBD DRM problem for client $k$ is $\min_{r_k \in \mathcal{H}_r} \hat{\mathcal{E}}_f^{++}(r_k)$ where

$$\hat{\mathcal{E}}_f^{++}(r_k) = \mathrm{ReLU}\Big( (\hat{\mathbb{E}}_{p_k^{\mathrm{tr}}} - C_k \sum_{l=1}^{K} \hat{\mathbb{E}}_{p_l^{\mathrm{te}}})[\ell_1(r_k(\mathbf{x}))] \Big) + \sum_{l=1}^{K} \hat{\mathbb{E}}_{p_l^{\mathrm{te}}}[\ell_2(r_k(\mathbf{x}))]. \tag{K.3}$$

Applying triangle inequality, we first decompose the bias term

$$\sup_{r_k \in \mathcal{H}_r} |\mathbb{E}[\hat{\mathcal{E}}_f^+(r_k)] - \mathcal{E}_f(r_k)| \leq \sup_{r_k \in \mathcal{H}_r} |\mathbb{E}[\hat{\mathcal{E}}_f^+(r_k) - \hat{\mathcal{E}}_f^{++}(r_k)]|$$
$$+ \sup_{r_k \in \mathcal{H}_r} |\mathbb{E}[\hat{\mathcal{E}}_f^{++}(r_k)] - \mathcal{E}_f(r_k)|. \tag{K.4}$$

An upper bound on $\sup_{r_k \in \mathcal{H}_r} |\mathbb{E}[\hat{\mathcal{E}}_f^{++}(r_k)] - \mathcal{E}_f(r_k)|$ is established similar to the proof of Lemma 4:

$$\sup_{r_k \in \mathcal{H}_r} |\mathbb{E}[\hat{\mathcal{E}}_f^{++}(r_k)] - \mathcal{E}_f(r_k)| \leq (1 + KC_k)\Delta_\ell \exp\left(\frac{-2\eta_k^2}{\Delta_\ell^2/n_k^{\text{tr}} + KC_k^2\Delta_\ell^2/n^{\text{te}}}\right).$$

Substituting Eq. (K.1) and Eq. (K.3) into $|\mathbb{E}[\hat{\mathcal{E}}_f^+(r_k)] - \hat{\mathcal{E}}_f^{++}(r_k)]|$, we have

$$|\mathbb{E}[\hat{\mathcal{E}}_f^+(r_k) - \hat{\mathcal{E}}_f^{++}(r_k)]|$$
$$= |\mathbb{E}[\text{ReLU}((\hat{\mathbb{E}}_{p_k^{\text{tr}}} - \tilde{C}_k \sum_{l=1}^K \hat{\mathbb{E}}_{p_l^{\text{te}}})[\ell_1(r_k(\mathbf{x}))]) - \text{ReLU}((\hat{\mathbb{E}}_{p_k^{\text{tr}}} - C_k \sum_{l=1}^K \hat{\mathbb{E}}_{p_l^{\text{te}}})[\ell_1(r_k(\mathbf{x}))])]|,$$

which together with $\text{ReLU}(a) - \text{ReLU}(b) \leq |a - b|$ is used to establish the following upper bound:

$$\left|\mathbb{E}\left[\hat{\mathcal{E}}_f^+(r_k) - \hat{\mathcal{E}}_f^{++}(r_k)\right]\right| \leq K\Delta_\ell|\tilde{C}_k - C_k|. \tag{K.5}$$

Let $m^* = \arg\max_{m \in [M]} \tilde{r}_{k,m}$. We note that by the construction of HDRM, there is a constant lower bound on the numerator of $\tilde{r}_k$, i.e., $\frac{1}{n^{\text{te}}} \sum_{j=1}^{n^{\text{te}}} \sum_{l=1}^K \mathbb{1}(\mathbf{x}_{l,j}^{\text{te}} \in \mathcal{B}_{m^*}) \geq \bar{c}$, that is achieved when $\{\mathbf{x}_{l,j}^{\text{te}}\}_{j=1}^{n^{\text{te}}}$ are distributed uniformly across $M$ bins. Let $\mathbf{x} \in \mathcal{X}$ and let $\hat{p}_k^{\text{tr}}(\mathbf{x}; M)$ denote a histogram-based density estimate of $p_k^{\text{tr}}(\mathbf{x})$ with $M$ bins. The maximum value of $\tilde{C}_k$ is attained when $\frac{1}{n^{\text{te}}} \sum_{j=1}^{n^{\text{te}}} \sum_{l=1}^K \mathbb{1}(\mathbf{x}_{l,j}^{\text{te}} \in \mathcal{B}_{m^*})$ meets its lower bound, which leads to the maximum deviation from $C_k \leq \tilde{C}_k$. Assuming $p_k^{\text{tr}}(\mathbf{x})$ is $L_k$-Lipschitz with $\sup_{\mathbf{x} \in \mathcal{X}} p_k^{\text{tr}}(\mathbf{x}) < \infty$, the mean squared error of a histogram-based density estimate with $M$ bins is upper bounded by (Wasserman, 2006, Section 6): $\mathbb{E}|\hat{p}_k^{\text{tr}}(\mathbf{x}; M) - p_k^{\text{tr}}(\mathbf{x})|^2 = \mathcal{O}(L_k^2/M^2 + M/n_k^{\text{tr}})$. Putting together with a constant lower bound on the numerator of $\tilde{r}_k$ and applying Jensen's inequality, we have:

$$\mathbb{E}[|\tilde{C}_k - C_k|] \lesssim \frac{1}{M} + \sqrt{\frac{M}{n_k^{\text{tr}}}}.$$

## L   Proof of Lemma 1

We first note that

$$\mathbb{E}[L(\hat{\boldsymbol{\theta}})] - L(\boldsymbol{\theta}_*) = \mathbb{E}\|\hat{\boldsymbol{\theta}} - \boldsymbol{\theta}_*\|_{\boldsymbol{\Sigma}^{\text{te}}}^2 = \text{Bias} + \text{Variance}.$$

We first find the expression for $\hat{\boldsymbol{\theta}}$ considering the ridge regression problem assuming $\frac{p^{\text{te}}(\mathbf{x})}{p^{\text{tr}}(\mathbf{x})}$ is given. FTW-ERM problem with Tikhonov regularization is given by

$$\hat{\boldsymbol{\theta}} = \arg\min_{\boldsymbol{\theta}} \sum_{i=1}^n w_i(\boldsymbol{\theta}^\top \mathbf{x}_i - y_i)^2 + \lambda\|\boldsymbol{\theta}\|_2^2$$

where $w_i = \frac{p^{\text{te}}(\mathbf{x}_i)}{p^{\text{tr}}(\mathbf{x}_i)}$ and $\lambda$ is the regularization parameter. This is a reweighted least squares problem.

The objective function above is strongly convex and differentiable. Applying the fist-order condition, the unique minimum is as follows:

$$\hat{\boldsymbol{\theta}} = \left(\mathbf{X}^\top \mathbf{W} \mathbf{X} + \lambda \mathbf{I}_d\right)^{-1} \mathbf{X}^\top \mathbf{W} \mathbf{y} \tag{L.1}$$

where $\mathbf{W} = \text{diag}(w_1, \ldots, w_n)$.

Substituting $\mathbf{y} = \mathbf{X}\boldsymbol{\theta}_* + \boldsymbol{\epsilon}$ into (L.1), we note that

$$\hat{\boldsymbol{\theta}} = \left(\mathbf{X}^\top \mathbf{W} \mathbf{X} + \lambda \mathbf{I}_d\right)^{-1} \mathbf{X}^\top \mathbf{W} \mathbf{X} \boldsymbol{\theta}_* + \left(\mathbf{X}^\top \mathbf{W} \mathbf{X} + \lambda \mathbf{I}_d\right)^{-1} \mathbf{X}^\top \mathbf{W} \boldsymbol{\epsilon}$$

and

$$\mathbb{E}_{\mathbf{X},\epsilon}[\hat{\boldsymbol{\theta}}] = \mathbb{E}_{\mathbf{X}}\left[\left(\mathbf{X}^\top\mathbf{W}\mathbf{X} + \lambda\mathbf{I}_d\right)^{-1}\mathbf{X}^\top\mathbf{W}\mathbf{X}\boldsymbol{\theta}_*\right].$$

We now characterize the bias $\mathtt{B}(\hat{\boldsymbol{\theta}})$ and variance $\mathtt{V}(\hat{\boldsymbol{\theta}})$ terms when the model estimate is given by (L.1).

Let $\|\mathbf{x}\|_\mathbf{A}^2 := \mathbf{x}^\top\mathbf{A}\mathbf{x}$. Substituting the expression for $\hat{\boldsymbol{\theta}}$ into $R(\hat{\boldsymbol{\theta}})$, the excess risk can be decomposed into a bias and a variance term as follows:

$$
\begin{aligned}
R(\hat{\boldsymbol{\theta}}) &= \mathbb{E}_{\mathbf{X},\epsilon,\mathbf{x},\epsilon_{\mathrm{te}}}[(y - \hat{\boldsymbol{\theta}}^\top\mathbf{x})^2 - (y - \boldsymbol{\theta}_*^\top\mathbf{x})^2]\\
&= \mathbb{E}_{\mathbf{X},\epsilon,\mathbf{x},\epsilon_{\mathrm{te}}}[\left(y - \boldsymbol{\theta}_*^\top\mathbf{x} + (\boldsymbol{\theta}_* - \hat{\boldsymbol{\theta}})^\top\mathbf{x}\right)^2 - (y - \boldsymbol{\theta}_*^\top\mathbf{x})^2]\\
&= \mathbb{E}_{\mathbf{X},\epsilon,\mathbf{x}}\left[\left((\boldsymbol{\theta}_* - \hat{\boldsymbol{\theta}})^\top\mathbf{x}\right)^2\right]\\
&= \mathbb{E}_{\mathbf{X},\epsilon}[\|\boldsymbol{\theta}_* - \hat{\boldsymbol{\theta}}\|_{\boldsymbol{\Sigma}^{\mathrm{te}}}^2]\\
&= \mathtt{B} + \mathtt{V}
\end{aligned}
$$

where the bias is given by

$$
\begin{aligned}
\mathtt{B} &= \mathbb{E}_{\mathbf{X},\epsilon}\left[\left\|\left(\mathbf{X}^\top\mathbf{W}\mathbf{X} + \lambda\mathbf{I}_d\right)^{-1}\mathbf{X}^\top\mathbf{W}\mathbf{X}\boldsymbol{\theta}_* - \boldsymbol{\theta}_*\right\|_{\boldsymbol{\Sigma}^{\mathrm{te}}}^2\right]\\
&= \mathbb{E}_{\mathbf{X}}\left[\left\|\left(\mathbf{X}^\top\mathbf{W}\mathbf{X} + \lambda\mathbf{I}_d\right)^{-1}\lambda\boldsymbol{\theta}_*\right\|_{\boldsymbol{\Sigma}^{\mathrm{te}}}^2\right].\\
&= \lambda^2\boldsymbol{\theta}_*^\top\mathbb{E}_{\mathbf{X}}[\boldsymbol{\Delta}_{\mathbf{W},\lambda}\boldsymbol{\Sigma}^{\mathrm{te}}\boldsymbol{\Delta}_{\mathbf{W},\lambda}]\boldsymbol{\theta}_*
\end{aligned}
$$

with

$$\boldsymbol{\Delta}_{\mathbf{W},\lambda} = \left[\left(\mathbf{X}^\top\mathbf{W}\mathbf{X} + \lambda\mathbf{I}_d\right)^{-1}\right],$$

and the variance is given by

$$
\begin{aligned}
\mathtt{V} &= \mathbb{E}_{\mathbf{X},\epsilon}\left[\left\|\left(\mathbf{X}^\top\mathbf{W}\mathbf{X} + \lambda\mathbf{I}_d\right)^{-1}\mathbf{X}^\top\mathbf{W}\epsilon\right\|_{\boldsymbol{\Sigma}^{\mathrm{te}}}^2\right]\\
&= \sigma_\epsilon^2\mathbb{E}_{\mathbf{X}}\left[\mathrm{tr}\left(\Phi_V\right)\right].
\end{aligned}
$$

where $\Phi_V = \left(\mathbf{X}^\top\mathbf{W}\mathbf{X} + \lambda\mathbf{I}_d\right)^{-1}\mathbf{X}^\top\mathbf{W}^2\mathbf{X}\left(\mathbf{X}^\top\mathbf{W}\mathbf{X} + \lambda\mathbf{I}_d\right)^{-1}\boldsymbol{\Sigma}^{\mathrm{te}}$.

## M  Proof of Theorem 2

In the one-hot case, it is clear that $\mathbf{X}^\top\mathbf{X} = \sum_{i=1}^n \mathbf{x}_i\mathbf{x}_i^\top$ and $\mathbf{X}^\top\mathbf{W}\mathbf{X} = \sum_{i=1}^n w_i\mathbf{x}_i\mathbf{x}_i^\top$ are diagonal matrices. For bias in the one-hot setting, we have

$$
\begin{aligned}
\mathtt{B}(\hat{\boldsymbol{\theta}}) &= \lambda^2\left[\boldsymbol{\theta}_*^\top\left(\mathbf{X}^\top\mathbf{W}\mathbf{X} + \lambda\mathbf{I}\right)^{-1}\boldsymbol{\Sigma}^{\mathrm{te}}\left(\mathbf{X}^\top\mathbf{W}\mathbf{X} + \lambda\mathbf{I}\right)^{-1}\boldsymbol{\theta}_*\right]\\
&= \lambda^2\sum_{i=1}^d\frac{[(\boldsymbol{\theta}_*)_i]^2\lambda_i'}{(\lambda_i(\mathbf{X}^\top\mathbf{W}\mathbf{X}) + \lambda)^2}\\
&= \lambda^2\sum_{i=1}^d\frac{[(\boldsymbol{\theta}_*)_i]^2\lambda_i'}{[\mu_i w_i + \lambda]^2}
\end{aligned}
$$

where the equation holds by the fact that, all matrices are diagonal including $\mathbf{X}^\top\mathbf{X}$, $\mathbf{X}^\top\mathbf{W}\mathbf{X}$, and $\boldsymbol{\Sigma}^{\mathrm{te}}$. Accordingly, we have $\lambda_i(\mathbf{X}^\top\mathbf{W}\mathbf{X}) = \lambda_i(\mathbf{X}^\top\mathbf{X})\lambda_i(\mathbf{W})$ with $i \in [d]$. For the classical ERM, the bias is

$$\mathtt{B}(\boldsymbol{\theta}^{\mathrm{v}}) = \lambda^2\sum_{i=1}^d\frac{[(\boldsymbol{\theta}_*)_i]^2\lambda_i}{[\mu_i + \lambda]^2}$$

where $\lambda_i$ is the eigenvalue of $\mathbf{\Sigma}^{\mathrm{tr}}$. To achieve $\mathtt{B}(\hat{\boldsymbol{\theta}}) \leqslant \mathtt{B}(\boldsymbol{\theta}^{\mathrm{v}})$, we have to make some assumptions on the relationship between $\lambda_i$, $\lambda_i'$ and $w_i$. Our analysis of error bound requires

$$\frac{\lambda_i'}{[\mu_i w_i + \lambda]^2} \leqslant \frac{\lambda_i}{[\mu_i + \lambda]^2} \Leftrightarrow \frac{\mu_i + \lambda}{\mu_i w_i + \lambda} \leqslant \sqrt{\frac{\lambda_i}{\lambda_i'}}, \tag{M.1}$$

which implies

$$w_i \geqslant \sqrt{\frac{\lambda_i'}{\lambda_i}} - 1, \tag{M.2}$$

such that Eq. (M.1) holds where we use the inequality $\frac{a+c}{b+c} \leqslant \frac{a}{b} + 1$ for any $a, b, c > 0$.

For the vanilla ERM, the variance is

$$\mathtt{V}(\boldsymbol{\theta}^{\mathrm{v}}) = \sigma_\epsilon^2 \sum_{i=1}^{d} \frac{\lambda_i \mu_i}{[\mu_i + \lambda]^2}.$$

For FTW-ERM, the variance is

$$\begin{aligned}
\mathtt{V}(\hat{\boldsymbol{\theta}}) &= \sigma_\epsilon^2 \left[ \left( \mathbf{X}^\top \mathbf{W} \mathbf{X} + \lambda \mathbf{I} \right)^{-1} \mathbf{X}^\top \mathbf{W}^2 \mathbf{X} \left( \mathbf{X}^\top \mathbf{W} \mathbf{X} + \lambda \mathbf{I} \right)^{-1} \mathbf{\Sigma}^{\mathrm{te}} \right] \\
&= \sigma_\epsilon^2 \sum_{i=1}^{d} \frac{\lambda_i' \lambda_i (\mathbf{X}^\top \mathbf{W}^2 \mathbf{X})}{(\lambda_i (\mathbf{X}^\top \mathbf{W} \mathbf{X}) + \lambda)^2} \\
&= \sigma_\epsilon^2 \sum_{i=1}^{d} \frac{\lambda_i' \mu_i w_i^2}{[\mu_i w_i + \lambda]^2}.
\end{aligned}$$

We note that $\mathtt{V}(\hat{\boldsymbol{\theta}}) \leq \mathtt{V}(\boldsymbol{\theta}^{\mathrm{v}})$ can be achieved by

$$\frac{\lambda_i \mu_i}{[\mu_i + \lambda]^2} \geq \frac{\lambda_i' \mu_i w_i^2}{[\mu_i w_i + \lambda]^2}.$$

This can be obtained by

$$\frac{\mu_i + \frac{\lambda}{w_i}}{\mu_i + \lambda} \geq \frac{\frac{\lambda}{w_i}}{\mu_i + \lambda} \geq \sqrt{\frac{\lambda_i'}{\lambda_i}}, \tag{M.3}$$

which implies $w_i \leqslant \xi_i \sqrt{\frac{\lambda_i}{\lambda_i'}}$. Combining Eqs. (M.2) and (M.3), the proof is complete.

## N   When FTW-ERM cannot outperform ERM

In this section, we provide a counterexample to show that, under which certain case, FTW-ERM cannot provably outperform ERM.

**Proposition 3.** *Under the same setting of Theorem 2, i.e., the fixed-design setting and label noise assumption, under the following condition*

$$\sqrt{\frac{\lambda_i'}{\lambda_i}} \geqslant \max\{\xi, 1 - \xi\}.$$

*If the ratio satisfies*

$$w_i \leqslant \min \left\{ \frac{1}{\frac{\sqrt{\lambda_i'/\lambda_i - 1}}{\xi} + 1}, \sqrt{\frac{\lambda_i'}{\lambda_i}} + \frac{\lambda}{\mu_i} \sqrt{\frac{\lambda_i'}{\lambda_i}} - \frac{\lambda}{\mu_i} \right\}, \tag{N.1}$$

*then we have*

$$R(\boldsymbol{\theta}^{\mathrm{v}}) \leqslant R(\hat{\boldsymbol{\theta}}).$$

*Proof.* According to Eq. (M.1), $B(\boldsymbol{\theta}^{\mathrm{v}}) \leqslant B(\hat{\boldsymbol{\theta}})$ holds by

$$\frac{\mu_i + \lambda}{\mu_i w_i + \lambda} \geqslant \sqrt{\frac{\lambda_i}{\lambda_i'}},$$

which is equivalent to

$$w_i \leqslant \sqrt{\frac{\lambda_i'}{\lambda_i}} + \frac{\lambda}{\mu_i}\sqrt{\frac{\lambda_i'}{\lambda_i}} - \frac{\lambda}{\mu_i}. \tag{N.2}$$

According to Eq. (M.3), $V(\boldsymbol{\theta}^{\mathrm{v}}) \leqslant V(\hat{\boldsymbol{\theta}})$ holds by

$$\frac{\mu_i + \frac{\lambda}{w_i}}{\mu_i + \lambda} \leq \sqrt{\frac{\lambda_i'}{\lambda_i}},$$

which is equivalent to

$$w_i \leqslant \frac{1}{\frac{\sqrt{\frac{\lambda_i'}{\lambda_i}-1}}{\xi}+1}. \tag{N.3}$$

Combining Eqs. (N.2) and (N.3), the proof is complete. To validate the condition in Eq. (N.1), we require each term in the RHS to be nonnegative. That implies

$$\sqrt{\frac{\lambda_i'}{\lambda_i}} \geqslant \max\{\xi, 1-\xi\},$$

which is our condition in Proposition 3.

By checking Eqs. (N.2) and (N.3), in both cases $\sqrt{\frac{\lambda_i'}{\lambda_i}} \geq 1$ and $\sqrt{\frac{\lambda_i'}{\lambda_i}} \leq 1$, we have

$$w_i \leqslant 1.$$

$\blacksquare$

## O   Experimental details and additional experiments

**Datasets**: We make use of three datasets in the experiments: *MNIST* (LeCun et al., 1998), *Fashion MNIST*[8] (Xiao et al., 2017), and *CIFAR10* (Krizhevsky). *MNIST* consists of images depicting handwritten digits from 0 to 9. The resolution of each image is $28 \times 28$. The dataset includes $60,000$ images for training. Similarly *Fashion MNIST* includes grayscale images of clothing of resolution $28 \times 28$. The training set consists of $60,000$ examples, and the test set of $10,000$ examples. *CIFAR10* consists of colored images with a resolution of $32 \times 32$. The training set contains $50,000$ examples while the test set contains $10,000$ examples.

**Experimental setup**: For all experiments we use the cross entropy loss. The stochastic gradient for each of the clients are computed with a batch size of 64 and aggregated on the server, which uses the Adam optimizer. Experiments on *MNIST* and *Fashion MNIST* uses a LeNet (LeCun et al., 1998), a learning rate of 0.001, no weight decay, and runs for $5,000$ iterations. For *CIFAR10* experiments we use the larger ResNet-18 (He et al., 2016). Batch normalization in ResNet-18 is treated by averaging the statistics on the server and subsequently broadcasting to the workers. A learning rate of 0.0001 and weight decay of 0.0001 are used. We report the best iterate in terms of average test accuracy after $20,000$ iterations in Table 7. FedBN Li et al. (2021c) is given a 10 times larger horizon due to slower convergence. The partial client participation experiment in Table 2 uses $200,000$ iterations.

All reported mean and standard deviations are computed over 5 independent runs except for CIFAR10 which uses 3 independent runs. For target shift the randomisation is also over the realization of the class distributions to ensure that the conclusions are not due to the particularities of the sub-sampled images. All experiments are carried out on an internal cluster using one GPU.

---

[8]*Fashion MNIST* is provided under the MIT license.

**Table 6:** CIFAR10 target shift distribution across 100 clients where groups of 10 clients shares the same distribution.

| | | Class 0 | 1 | 2 | 3 | 4 | 5 | 6 | 7 | 8 | 9 |
|---|---|---|---|---|---|---|---|---|---|---|---|
| Client 1-10 | Train | $95/100$ | $5/9$ | $5/9$ | $5/9$ | $5/9$ | $5/9$ | $5/9$ | $5/9$ | $5/9$ | $5/9$ |
| | Test | $5/9$ | $5/9$ | $5/9$ | $5/9$ | $5/9$ | $5/9$ | $5/9$ | $5/9$ | $5/9$ | $95/100$ |
| Client 11-20 | Train | $5/9$ | $95/100$ | $5/9$ | $5/9$ | $5/9$ | $5/9$ | $5/9$ | $5/9$ | $5/9$ | $5/9$ |
| | Test | $5/9$ | $5/9$ | $5/9$ | $5/9$ | $5/9$ | $5/9$ | $5/9$ | $5/9$ | $95/100$ | $5/9$ |
| Client 21-30 | Train | $5/9$ | $5/9$ | $95/100$ | $5/9$ | $5/9$ | $5/9$ | $5/9$ | $5/9$ | $5/9$ | $5/9$ |
| | Test | $5/9$ | $5/9$ | $5/9$ | $5/9$ | $5/9$ | $5/9$ | $5/9$ | $95/100$ | $5/9$ | $5/9$ |
| Client 31-40 | Train | $5/9$ | $5/9$ | $5/9$ | $95/100$ | $5/9$ | $5/9$ | $5/9$ | $5/9$ | $5/9$ | $5/9$ |
| | Test | $5/9$ | $5/9$ | $5/9$ | $5/9$ | $5/9$ | $5/9$ | $95/100$ | $5/9$ | $5/9$ | $5/9$ |
| Client 41-50 | Train | $5/9$ | $5/9$ | $5/9$ | $5/9$ | $95/100$ | $5/9$ | $5/9$ | $5/9$ | $5/9$ | $5/9$ |
| | Test | $5/9$ | $5/9$ | $5/9$ | $5/9$ | $5/9$ | $95/100$ | $5/9$ | $5/9$ | $5/9$ | $5/9$ |
| Client 51-60 | Train | $5/9$ | $5/9$ | $5/9$ | $5/9$ | $5/9$ | $95/100$ | $5/9$ | $5/9$ | $5/9$ | $5/9$ |
| | Test | $5/9$ | $5/9$ | $5/9$ | $5/9$ | $95/100$ | $5/9$ | $5/9$ | $5/9$ | $5/9$ | $5/9$ |
| Client 61-70 | Train | $5/9$ | $5/9$ | $5/9$ | $5/9$ | $5/9$ | $5/9$ | $95/100$ | $5/9$ | $5/9$ | $5/9$ |
| | Test | $5/9$ | $5/9$ | $5/9$ | $95/100$ | $5/9$ | $5/9$ | $5/9$ | $5/9$ | $5/9$ | $5/9$ |
| Client 71-80 | Train | $5/9$ | $5/9$ | $5/9$ | $5/9$ | $5/9$ | $5/9$ | $5/9$ | $95/100$ | $5/9$ | $5/9$ |
| | Test | $5/9$ | $5/9$ | $95/100$ | $5/9$ | $5/9$ | $5/9$ | $5/9$ | $5/9$ | $5/9$ | $5/9$ |
| Client 81-90 | Train | $5/9$ | $5/9$ | $5/9$ | $5/9$ | $5/9$ | $5/9$ | $5/9$ | $5/9$ | $95/100$ | $5/9$ |
| | Test | $5/9$ | $95/100$ | $5/9$ | $5/9$ | $5/9$ | $5/9$ | $5/9$ | $5/9$ | $5/9$ | $5/9$ |
| Client 91-100 | Train | $5/9$ | $5/9$ | $5/9$ | $5/9$ | $5/9$ | $5/9$ | $5/9$ | $5/9$ | $5/9$ | $95/100$ |
| | Test | $95/100$ | $5/9$ | $5/9$ | $5/9$ | $5/9$ | $5/9$ | $5/9$ | $5/9$ | $5/9$ | $5/9$ |

### O.1 Target shift

For the target shift experiments on *Fashion MNIST* in Table 1, we summarize the different number of data points for each dataset split in Table 9. A similar distribution across clients is used for the additional experiments for FTW-ERM and FedAvg on *CIFAR10* (Table 8). *CIFAR10* differs from *Fashion MNIST* in the number of examples due to the training set being smaller. The results for *CIFAR10* in Table 7 shows that FTW-ERM uniformly improves the accuracy over FedAvg on this difficult target shift instance. We additionally include a two-client setting in Table 10 with the associated distribution described in Table 11.

To compute the exact ratio $r(\mathbf{x})$ we will assume that the distributions are separable.

**Definition 3** (Separability). *A distribution over $\mathcal{X} \times \mathcal{Y}$ is separable if there exists a partition $(\mathcal{X}_i)_{i=1}^m$ of $\mathcal{X}$ such that $p(y_i|\mathcal{X}_i) = 1$ for some $y_i \in \mathcal{Y}$ and all $i \in [m]$. We denote the associated deterministic label assignment as $g : \mathcal{X} \to \mathcal{Y}$.*

**Proposition 4.** *Assume that the distributions $p^{\text{te}}(\mathbf{x}, y)$ and $p^{\text{tr}}(\mathbf{x}, y)$ are both separable. Then the ratio can be computed based on the associated label $y := g(\mathbf{x})$ as follows,*

$$r(x) = \frac{p^{\text{te}}(y)}{p^{\text{tr}}(y)}. \tag{O.1}$$

*Proof.* Due to separability, $p^{\text{te}}(y|x) = p^{\text{tr}}(y|x)$. So

$$r(\mathbf{x}) := \frac{p^{\text{te}}(\mathbf{x})}{p^{\text{tr}}(\mathbf{x})} = \frac{p^{\text{te}}(\mathbf{x})p^{\text{te}}(y|\mathbf{x})}{p^{\text{tr}}(\mathbf{x})p^{\text{tr}}(y|\mathbf{x})} = \frac{p^{\text{te}}(\mathbf{x}, y)}{p^{\text{tr}}(\mathbf{x}, y)}. \tag{O.2}$$

It follows that,

$$\frac{p^{\text{te}}(\mathbf{x}, y)}{p^{\text{tr}}(\mathbf{x}, y)} = \frac{p^{\text{te}}(\mathbf{x}|y)p^{\text{te}}(y)}{p^{\text{tr}}(\mathbf{x}|y)p^{\text{tr}}(y)}. \tag{O.3}$$

Using the definition of the target shift assumption, $p^{\text{te}}(\mathbf{x}|y) = p^{\text{tr}}(\mathbf{x}|y)$, the conditional distributions cancel and we obtain the claim. ∎

**Table 7:** Target shift on CIFAR10 with ResNet-18. Note that FedBN is given a 10 times larger horizon due to slower convergence.

|                   | FTW-ERM              | FedAvg             | FedBN             |
| ----------------- | -------------------- | ------------------ | ----------------- |
| Average accuracy  | $\mathbf{0.6004} \pm 0.0076$ | $0.4426 \pm 0.0291$ | $0.5081 \pm 0.0520$ |
| Client 1 accuracy | $\mathbf{0.6714} \pm 0.0153$ | $0.3984 \pm 0.1497$ | $0.6699 \pm 0.1283$ |
| Client 2 accuracy | $\mathbf{0.8196} \pm 0.0962$ | $0.7307 \pm 0.1533$ | $0.7213 \pm 0.0810$ |
| Client 3 accuracy | $\mathbf{0.5412} \pm 0.0776$ | $0.3333 \pm 0.2251$ | $0.1584 \pm 0.1222$ |
| Client 4 accuracy | $\mathbf{0.5087} \pm 0.0827$ | $0.3030 \pm 0.1106$ | $0.2907 \pm 0.1859$ |
| Client 5 accuracy | $0.4610 \pm 0.0508$ | $0.4476 \pm 0.3649$ | $\mathbf{0.7003} \pm 0.0386$ |

**Table 8:** CIFAR10 target shift distribution.

|          |       | \multicolumn{10}{c}{Class} | | | | | | | | | |
| -------- | ----- | --- | --- | --- | --- | --- | ---- | ---- | ---- | ---- | ---- |
|          |       | 0   | 1   | 2   | 3   | 4   | 5    | 6    | 7    | 8    | 9    |
| Client 1 | Train | 28  | 28  | 28  | 28  | 28  | 4885 | 28   | 28   | 28   | 28   |
|          | Test  | 977 | 5   | 5   | 5   | 5   | 5    | 5    | 5    | 5    | 5    |
| Client 2 | Train | 28  | 28  | 28  | 28  | 28  | 28   | 4885 | 28   | 28   | 28   |
|          | Test  | 5   | 977 | 5   | 5   | 5   | 5    | 5    | 5    | 5    | 5    |
| Client 3 | Train | 28  | 28  | 28  | 28  | 28  | 28   | 28   | 4885 | 28   | 28   |
|          | Test  | 5   | 5   | 977 | 5   | 5   | 5    | 5    | 5    | 5    | 5    |
| Client 4 | Train | 28  | 28  | 28  | 28  | 28  | 28   | 28   | 28   | 4885 | 28   |
|          | Test  | 5   | 5   | 5   | 977 | 5   | 5    | 5    | 5    | 5    | 5    |
| Client 5 | Train | 28  | 28  | 28  | 28  | 28  | 28   | 28   | 28   | 28   | 4885 |
|          | Test  | 5   | 5   | 5   | 5   | 977 | 5    | 5    | 5    | 5    | 5    |

**Table 9:** Fashion MNIST target shift distribution.

|          |       | \multicolumn{10}{c}{Class} | | | | | | | | | |
| -------- | ----- | --- | --- | --- | --- | --- | ---- | ---- | ---- | ---- | ---- |
|          |       | 0   | 1   | 2   | 3   | 4   | 5    | 6    | 7    | 8    | 9    |
| Client 1 | Train | 34  | 34  | 34  | 34  | 34  | 5862 | 34   | 34   | 34   | 34   |
|          | Test  | 977 | 5   | 5   | 5   | 5   | 5    | 5    | 5    | 5    | 5    |
| Client 2 | Train | 34  | 34  | 34  | 34  | 34  | 34   | 5862 | 34   | 34   | 34   |
|          | Test  | 5   | 977 | 5   | 5   | 5   | 5    | 5    | 5    | 5    | 5    |
| Client 3 | Train | 34  | 34  | 34  | 34  | 34  | 34   | 34   | 5862 | 34   | 34   |
|          | Test  | 5   | 5   | 977 | 5   | 5   | 5    | 5    | 5    | 5    | 5    |
| Client 4 | Train | 34  | 34  | 34  | 34  | 34  | 34   | 34   | 34   | 5862 | 34   |
|          | Test  | 5   | 5   | 5   | 977 | 5   | 5    | 5    | 5    | 5    | 5    |
| Client 5 | Train | 34  | 34  | 34  | 34  | 34  | 34   | 34   | 34   | 34   | 5862 |
|          | Test  | 5   | 5   | 5   | 5   | 977 | 5    | 5    | 5    | 5    | 5    |

Proposition 4 provides a way to compute the ratio $r(\mathbf{x})$ when the labels are available and the shift is known.

## O.2 Covariate shift

The color flipping probability used to generate each of the colored MNIST datasets for the covariate shift experiment can be found in Table 12. We consider an asymmetric client setup where client 1 in addition has 40 times less training examples than client 2.

**Table 10:** Fashion MNIST with target shift across two clients.

|                   | FTW-ERM         | FITW-ERM        | FedAvg          |
| ----------------- | --------------- | --------------- | --------------- |
| Average accuracy  | **0.82** ± 0.00 | 0.76 ± 0.01     | 0.76 ± 0.01     |
| Client 1 accuracy | 0.89 ± 0.01     | 0.80 ± 0.02     | **0.94** ± 0.00 |
| Client 2 accuracy | **0.74** ± 0.01 | 0.71 ± 0.02     | 0.58 ± 0.01     |

**Table 11:** Two-client Fashion MNIST. The number of samples for each class across the different datasets.

|          |       | \multicolumn{10}{c}{Class} |
| -------- | ----- | --- | --- | --- | --- | --- | ---- | ---- | ---- | ---- | ---- |
|          |       | 0   | 1   | 2   | 3   | 4   | 5    | 6    | 7    | 8    | 9    |
| Client 1 | Train | 100 | 100 | 100 | 100 | 100 | 100  | 100  | 100  | 100  | 100  |
|          | Test  | 9   | 9   | 9   | 9   | 9   | 990  | 990  | 990  | 990  | 990  |
| Client 2 | Train | 39  | 39  | 39  | 39  | 39  | 3986 | 3986 | 3986 | 3986 | 3986 |
|          | Test  | 990 | 990 | 990 | 990 | 990 | 9    | 9    | 9    | 9    | 9    |

# P    Computational complexity of Algorithm 1

We note that clients compute the ratios *in parallel* where each client needs to estimate one ratio. To estimate density ratios for FTW-ERM, clients require to send a few unlabelled test samples *only once*. The server shuffles those samples and broadcasts the shuffled version to clients *only once*. Compared to FedAvg, the additional computational cost per client is $\mathcal{O}(TN_k)$ where $T$ is the number of iterations for Algorithm 1 to converge and $N_k$ is the number of batches for ratio estimation. Compared to baseline FedAvg, the additional computation of FTW-ERM is negligible but leads to substantial improvements of the overall generalization in settings under challenging distribution shifts.

# Q    Limitations

In this paper, we focus on settings where ratio estimation is required *once* prior to model training. Handling distribution shifts in complex non-stationary settings where ratio estimation is an ongoing process is an interesting problem for future work.

In addition, various personalization methods have been proposed to improve fairness in terms of uniformity of model performance across clients (Li et al., 2021a;b). To meet specific requirements of each client, our global model can be combined with a personalized model on each client. Developing new variants of FTW-ERM with a focus on fairness is an interesting problem for future work.

To estimate $\{r_k(\mathbf{x})\}_{k=1}^K$, clients need to send unlabelled samples $\mathbf{x}_{l,j}^{\text{te}}$ for $l \in [K]$ and $j \in [n^{\text{te}}]$ from their test distributions. We note that instead of their true samples, clients can alternatively send samples generated from a *generative model* (Goodfellow et al., 2020).

Note that training GANs may be computationally extensive due to required computational resources and availability of representative samples. However, we propose to use GANs as an alternative method with clear caveats, only when 1) clients have sufficient computational resources and 2) they are unwilling to share unlabelled data with the server.

**Table 12:** For covariate shift the datasets for each of the client are constructed using different probabilities.

|                            | $p_1^{\text{tr}}(\mathbf{x})$ | $p_1^{\text{te}}(\mathbf{x})$ | $p_2^{\text{tr}}(\mathbf{x})$ | $p_2^{\text{te}}(\mathbf{x})$ |
| -------------------------- | ----- | ----- | ----- | ----- |
| Probability of flipping color | 0.5 | 0.2 | 0.2 | 0.8 |

**Table 13:** Estimating ratio upper bound with $k$-means clustering. We consider the target shift setup, such that a tight upper bound is known, and construct a single client variant for simplicity. We specifically consider MNIST with a label distribution during training and testing to be $q^{\text{tr}} \propto (1/20, 1/20, 1/20, 1/20, 1/20, 1, 1, 1, 1, 1)^\top$ and $q^{\text{te}} \propto (1, 1, 1, 1, 1, 1/20, 1/20, 1/20, 1/20, 1/20)^\top$ respectively. The table shows the estimated upper bound on the ratio ($\tilde{r}$) for a range of clustering sizes. A reasonable estimate of the true maximal ratio of 20 is obtained for a wide range of clustering sizes. Whereas naively binning the space can be problematic due to division by zero, the clustering approach is less prone to this issue as long as #(clusters) $\ll$ #(datapoints).

| #(clusters) | 10 | 20 | 40 | 50 | 100 | 200 | 500 |
|---|---|---|---|---|---|---|---|
| $\tilde{r}$ | 10.31 | 15.48 | 19.08 | 27.41 | 31.47 | 32.84 | 206.76 |

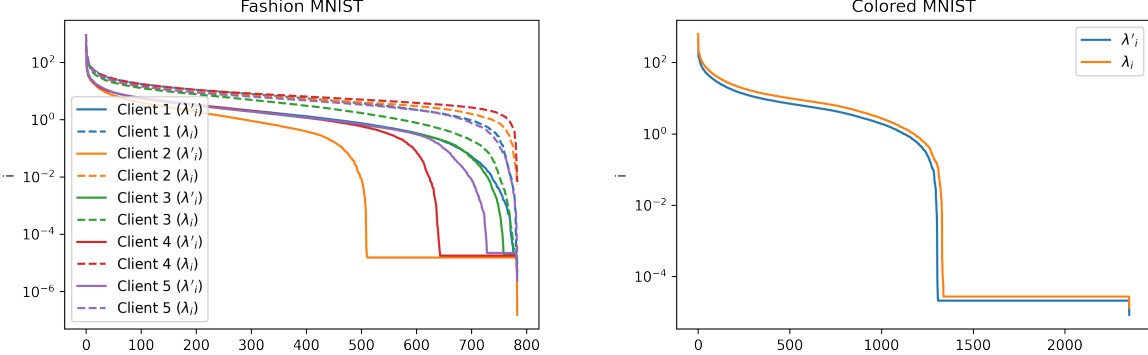

**Figure 3:** The sudden increase in the eigenvalue ratio observed in Figure 2 for $i$ larger than 780 and 1300 in Fashion MNIST and CMNIST respectively is most likely due to numerical error. We plot the eigenvalues under the test distributions ($\lambda'$) and under the training distribution ($\lambda$) and observe that indeed the eigenvalues suddenly drop very close to zero at those exact indices.

As a partial mitigation of privacy risks, we introduced FITW-ERM. FITW-ERM does not require any data sharing among clients and does not require any GAN training. In this paper, we focus on FTW-ERM since it outputs an unbiased estimate of a minimizer of the overall *true risk*, and enables us to theoretically show the benefit of importance weighting in generalization.

One particular challenge in real-world cross-device FL is to estimate ratios on real-world datasets such as WILDS (Koh et al., 2021) and LEAF (Caldas et al., 2019). WILDS has been mostly used for domain generalization, where the setting is not similar to ours. We still have to decide on an arbitrary test/train split. LEAF mainly captures inter-client distribution shifts and settings where different clients have different numbers of examples over thousands of clients. This work is not about scalability to thousands of clients experimentally using our single GPU simulated setup. While we anticipate efficient ratio estimation will improve over time, our FTW-ERM and FITW-ERM formulations along with improved ratio estimates will provide reasonable solutions to learn an effective global model in real-world cross-device FL under covariate shifts.

