# OpenReview forum: "Federated Learning under Covariate Shifts with Generalization Guarantees"
_TMLR — Accepted by TMLR_

### Review · Reviewer_i3bH · 2023-03-07

**Summary Of Contributions:**

In this paper authors study generalization properties in Federated Learning. They focus on intra-client and inter-client covariate shifts. To overcome the problems caused by the existence of the shifts, authors propose a general framework called Federated Importance-weighteD Empirical risk Minimization. Moreover, the authors propose a communication-efficient variant with the same level of privacy guarantees. At the end of the paper, experimental results are presented.

More specifically, the authors propose a framework to minimize the average test error for the Federated Learning problem and design a technique to control privacy leakages and also have communication-efficient properties.

The paper provides a theoretical analysis of high-probability guarantees on ratio estimation error with multiple clients. In this case, the supremum is calculated not exactly. The analysis shows the advantage of importance weighting in terms of excess risk decoupled from density ratio estimation through bias-variance decomposition.

The authors demonstrate experimental results that show more than 16% overall test accuracy improvement over existing FL baselines for the ResNet-18 model and CIFAR10 dataset.



**Audience:**

Yes

**Broader Impact Concerns:**

In this paper, the Broader Impact Statement is not presented. This paper is of theoretical nature, and I do not think that obtained results can lead to ethical issues.

**Claims And Evidence:**

Yes

**Requested Changes:**

I suggest changing the name of the framework and reducing the percentage of text with italics.
Also, I suggest providing more details for Bregman divergence and experiments.

**Strengths And Weaknesses:**

I am not an expert in privacy theory and in generalization studies, so my review is an educated guess.

Strengths:
1)This paper has a clear structure, and it is easy to read. The paper has good storytelling and flow.
2) The introduction part is well-written, and the motivation of the research is clearly stated.
3) All assumptions, lemmas, and theorems are rigorously described.
4) Contribution seems to be significant.
5) Theoretical analysis seems to be correct. However, I might miss something.

Weaknesses:
1) I think that FIDEM is a good name for the framework. Since this framework is connected to ERM, it must contain ERM in the name. Otherwise, it is confusing for readers. Also, having capitalized letters at the end of the word (weighteD) is questionable.
2) I do not like that a large part of the text is written in italics. It is suitable for theorems or assumptions, but if the whole section is written in italics, it is hard to follow.
3) The definition of Bregman divergence for strictly convex function with bounded gradients is not clear. The set should be bounded. Otherwise, strict convexity and bounded gradients will contradict each other. This should be described.
4) Figure 2 is not clear to readers. The nature of peaks is not explained.

---

> ### Author Response · Authors · 2023-03-15
> **Response to Reviewer i3bH**
>
> We thank the reviewer for their thoughtful comments which we address one by one below.
>
> ------
>
>
> -  I think that FIDEM is a good name for the framework. Since this framework is connected to ERM, it must contain ERM in the name. Otherwise, it is confusing for readers. Also, having capitalized letters at the end of the word (weighteD) is questionable:
>
> In the revision, we renamed FIDEM and FIIDEM to Federated Importance-Weighted Empirical Risk Minimization (FTW-ERM) and Federated Independent Importance-Weighted Empirical Risk Minimization (FITW-ERM), respectively.
>
> Similar acronyms have been used in the literature (Jaggi et al., 2014; Smith et al., 2017; Hongyu et al., 2020).
>
> M. Jaggi, V. Smith, J. Terhorst, S. Krishnan, T. Hofmann, and M. I. Jordan. Communication-Efficient Distributed Dual Coordinate Ascent. In Advances in Neural Information Processing Systems (NeurIPS), 2014.
>
> V. Smith, C.-K. Chiang, M. Sanjabi, and A. S. Talwalkar. Federated multi-task learning. In Advances in Neural Information Processing Systems (NeurIPS), 2017.
>
> R. Hongyu, Y. Zhu, J. Leskovec, A. Anandkumar, and A. Garg. OCEAN: Online Task Inference for Compositional Tasks with Context Adaptation. In Conference on Uncertainty in Artificial Intelligence (UAI), 2020.
>
>
> ------
>
> -  Reduce the percentage of text with italics:
>
> Thanks for your comment. In the revision, we made sure the entire section is not written in italics and reduced the percentage of text italics in particular in Section 2.
>
> ------
>
> -  The definition of Bregman divergence for strictly convex function with bounded gradients is not clear. The set should be bounded:
>
> In the revision, we clarify that ${\cal B}_f$ in Definition 1 is a bounded set and add the following remark in Section 3:
>
> The bounded ${\cal B}_f$ is a standard assumption (Kato & Teshima, 2021), which holds in our problem since the density ratios that are inputs of BD  are bounded. We estimate the supremum over true ratios in Section 3.2 and provide examples of $f$ commonly used for BD-based methods in Table 4 of Appendix A.
>
>
> ------
>
> -  Explain the nature of peaks in Figure 2:
>
> We have elaborated on the peaks and added an additional figure in the appendix (fig. 3) which plots the eigenvalues. The peaks in the eigenvalue ratios occur when the eigenvalues are very small in which case the ratio becomes sensitive to the numerical precision.

---

> ### Author Response · Authors · 2023-03-27
> **Follow-up**
>
> Dear reviewer,
>
> We would like to check whether you have any questions/concerns? We also welcome any suggestions on improving our paper if the reviewer believes that it will make it easier for the reader to understand.

---

### Review · Reviewer_hsZ4 · 2023-03-12

**Summary Of Contributions:**

This paper proposes a new method to solve the covariate shifts in federated learning, called FIDEM. It shows the generalization guarantees of FIDEM and conducts experiments on MNIST, CIFAR-10 and ColoredMNIST. The experiments show advantages over FedAvg.

**Audience:**

Yes

**Broader Impact Concerns:**

I don't have ethical concerns.

**Claims And Evidence:**

No

**Requested Changes:**

See above.

**Strengths And Weaknesses:**

Strengths:
1. covariate shift is an important problem in FL and in general machine learning. This paper proposes a new method based on importance sampling to address the covariate shift.
2. The literature survey is quite complete.
3. This paper shows generalization guarantees of error estimation and gives supporting experiments to show the advantages of FIDEM.

Weaknesses:
1. Clarity:
 * After Sec 2 the font suddenly becomes italic, which makes it hard to read.
 * Sec 1: Page 1, $p^{\rm tr}_k$ and $p^{\rm te}_k$ are not defined.
 * Sec 1.1: BD and DRM are not defined
* "Our work is the first step towards ... distribution shifts": cannot agree on this. There are existing works on it, such as https://arxiv.org/abs/2205.11101, https://arxiv.org/pdf/2206.09979.pdf, https://arxiv.org/abs/2102.07623, etc.
2. Minor:
* Page 2: CIFAR10 (Krizhevsky)
* Page 3: our setting our setting
* Page 4: i.i.d -> i.i.d.
* Page 5: sum test over own train: what does it mean?
* Sec 3.2: what is $C$?
3. Novelty:
* The nnBD DRM method seems like a direct application of Kato and Teshima 2021. What is the novel part?
4. Soundness:
* Algorithm 1 requires additional communication of samples, which violates privacy constraints more than FedAvg, and it also violates the original design of FL: samples should be kept in local clients.
* Sec 4.1: the original paper on Rademacher complexity should be Koltchinskii, V. (2001). Rademacher penalties and structural risk minimization. IEEE Transactions on Information Theory, 47(5):1902–1914.
5. Questions:
* eq. (2.1): the loss depends on the density ratio estimation. It also requires whenever $p_k^{\rm tr} = 0$, $p_l^{\rm te} = 0$. How can this be ensured?
* Sec 2.3: why is it impossible to apply a gradient inversion attack? It seems possible as long as we can find the gradient.
* In Theorem 2, could you explain why FIDEM is better than ERM? If we minimize the total loss, it seems that ERM always obtains the optimal solution for the empirical risk. Is it due to the sampling error?
* In Figure 2, there is a sudden increase and the authors explain it is due to numerical error. However, such an explanation is not quite convincing and it will violate the assumption in Theorem 2.
* Additional baseline algorithms that address covariate shifts should be compared, such as FedBN, FedNova and the related work mentioned above. Also, FIDEM requires additional samples to be sent to the server, which makes the comparison with FedAvg quite unfair.

---

> ### Author Response · Authors · 2023-03-22
> **Response to Reviewer hsZ4 Part 1**
>
> We thank the reviewer for their thoughtful comments which we address one by one below. The manuscript is revised, and the changes are shown in blue.
>
> ------
>
>
> - Comparison with  (Wang et al., 2020; Li et al., 2021; de Luca et al., 2022; Gupta et al., 2022).
>
> We thank the reviewer for the remark. In the Section 1.2 of the revision, we have cited (Wang et al., 2020; Li et al., 2021; de Luca et al., 2022; Gupta et al., 2022). We clarify FedNova of Wang et al. (2020) tackles update drifts considering variations in the number of local updates performed by each client in each communication round and focuses on minimizing the empirical risk under the same training/test data distribution assumption over each client (non-IID datasets across clients). In contrast, we focus on learning and overall generalization performance under both intra/inter-client covariate shifts. Under our setting with one local update for each client,  FedNova reduces to FedAvg. This holds since ${\bf a}_i$ in Eq. (12) of (Wang et al., 2020) becomes  a non-negative scalar for all $i$’s so the final update will become Eq. (2) of (Wang et al., 2020).
>
> Li et al. (2021) propose FedBD to tackle inter-client feature shift by updating Batch Normalization (BN) layers locally and updating non-BN layers using FedAvg. They consider both inter-client covariate shift and concept shift but under the same training/test data distribution assumption over each client. Note that for models without BN layers, FedBD reduces to FedAvg. We handle intra-client and inter-client covariate shifts and do not require BN layers.
>
> De Luca et al. (2022) consider Federated Domain Generalization and propose data augmentation to learn a model that generalizes to in-domain datasets of the participating clients and an out-of-domain dataset of a non-participating client. They propose to use FedAvg after proper data augmentation, which is orthogonal to the algorithmic design, e.g., our work. In particular, true labels are directly used for the structural  causal models in (De Luca et al., 2022). Our FTW-ERM uses *unlabelled* and shuffled test samples just to estimate density ratios. Given ratios, each client computes stochastic gradients using its own original local training data. We also propose FITW-ERM  that does not require any form of data sharing among clients and preserves the same level of privacy and same communication costs as those of FedAvg. Their method can possibly complement our work.
>
> Gupta et al. (2022) propose FL Games, a game-theoretic framework for learning causal features that are invariant across clients by using ensembles over clients’ historical actions and increasing the local computation under the same training/test data distribution assumption over each client (non-IID datasets across clients). Their method requires each client to maintain a buffer and store its historically played actions, which is not required in our work. We note the setting and the objective of Invariant Risk Minimization (IPM) Games in (Gupta et al., 2022, Eq. 4) are different from our setting with intra/inter-client covariate shift and our objective of minimizing the average test error, respectively.
>
> Overall, different from these work, we focus on learning and overall generalization performance, i.e.,  minimizing the average test error over all clients, under both intra/inter-client covariate shifts.
>
> FedBN is the closest work that explicitly considers covariate shifts in their problem formulation, for which we provide **numerical comparison in the revised version**.  We have included FedBN as an additional baseline in the CIFAR10 experiments where batch normalization is used. Under full participation the method improves over FedAvg, but remains uncompetitive with our method FTW-ERM (Table 7 in Appendix O). This is true even after giving FedBN the advantage of 10 times more iterations, which we did due to observing slow convergence of the method. Under partial participation (Table 2) FedBN performs worse than even FedAvg, which is maybe not surprising given that the batch normalization parameters are only updated locally by the method. Consequently, the statistics on a given client are only updated the number of times that client is sampled.
>
>
> J. Wang, Q. Liu, H. Liang, G. Joshi, and H. V. Poor. Tackling the objective inconsistency problem in heterogeneous federated optimization. In Advances in Neural Information Processing Systems (NeurIPS), 2020.
>
> X. Li, M. Jiang, X. Zhang, M. Kamp, and Q. Dou. FedBN: Federated learning on non-iid features via local batch normalization. In International Conference on Learning Representations (ICLR), 2021.
>
> A. B. de Luca, G. Zhang, X. Chen, and Y. Yu. Mitigating Data Heterogeneity in Federated Learning with Data Augmentation. arXiv preprint arXiv:2206.09979, 2022.
>
> S. Gupta, K. Ahuja, M. Havaei, N. Chatterjee, and Y. Bengio. FL Games: A federated learning framework for distribution shifts. arXiv preprint arXiv:2205.11101, 2022.
>
> ------

---

> ### Author Response · Authors · 2023-03-22
> **Response to Reviewer hsZ4 Part 2**
>
> -  Italic font and typos:
>
> Thanks for carefully reading our paper. We have fixed them in the revision.
>
>
>
> ------
>
> -  Define $p_{k}^{\mathrm{tr}}$ and $p_{k}^{\mathrm{te}}$ on page 1:
>
> In the revision, we clarified “covariate
> shift assumes marginal train distributions $p_{k}^{\mathrm{tr}}({\bf x})$ and marginal test distributions $p_{k}^{\mathrm{te}}({\bf x})$ can be arbitrarily different; while …”
>
>
> ------
>
> -  Define BD and DRM in Section 1.1:
>
>
> DRM is defined in the paragraph above Section 1.1. We defined BD in Section 1.1 of the revision.
>
> ------
>
> -  Clarify “sum test over own train” on page 5:
>
> We changed it to “sum of test densities over own train density, e.g., $\sum_{l=1}^K p_l^{\mathrm{te}}/p_k^{\mathrm{tr}}$ for client $k$ ”
>
> ------
>
> -  What is $C$:
>
> In Section 3.2 of th revision, we have clarified “$0<C<\frac{1}{\overline r}$”.
>
> ------
>
> -  Clarify the novelty of density ratio matching compared to Kato & Teshima (2021):
>
> ​​While Kato & Teshima (2021) have proposed nnBD DRM for a single train and test distributions, we consider a federated setting with multiple clients.
>
> A key step for the nnBD DRM method is to obtain an estimate of the supremum over true ratio, in particular when training a deep neural network. While Kato & Teshima (2021) regards this supremum  as an additional hyper-parameter, we efficiently estimate *multiple* ratios and relax the requirement to have a perfect estimate of $\overline r_k=\sup_{{\bf x}\in {\cal X}^{\mathrm{tr}}}r_k^*({\bf x})$ while controlling privacy leakage to other clients, which is non-trivial.
>
> ------
>
> -  Algorithm 1 requires additional communication of samples, which violates privacy constraints and the design of FedAvg.
>
> To estimate density ratios, if clients can tolerate some level of privacy leakage, clients send **unlabelled** samples from their test distributions. To control privacy leakage to other clients, we propose that the server *randomly shuffles* these unlabelled samples before broadcasting to clients. In Appendix Q, we discuss an alternative method instead of sending original unlabelled samples and discuss its limitations.
>
> To fully **eliminate any privacy risks**, we introduce FITW-ERM in Section 2.3. It does not require any form of data sharing among clients and preserves the *same level of privacy* and *same communication costs* as those of baseline FedAvg.
>
>
>
> ------
>
> -  Citation for Rademacher complexity:
>
> In Section 4.1 of the revision, we have  “.. Rademacher complexity (Koltchinskii, 2001) …”
>
>
> ------
>
> -  Eq. (2.1) requires $p_l^{\mathrm{te}}=0$ whenever $p_k^{\mathrm{tr}}=0$:
>
> We need a common data domain with strictly positive train density $p_k^{\mathrm{tr}}({\bf x}^{\mathrm{tr}})>0$ in the support and ${\cal X}^{\mathrm{te}}\subseteq{\cal X}^{\mathrm{tr}}$
> similar to (Kato & Teshima, 2021), (Kanamori et al., 2009, Section 2.1).
>
> In FL, this is quite mild, e.g., the common data domain can include images with $W\times H\times C$ pixels where $W$, $H$, and $C$ denote the width, height, and number of channels.
>
> We have addressed this in Section 2.2 of the revision.
>
>
> ------
>
> - Why is it impossible to apply a gradient inversion attack?
>
>
> The ratio is unknown to the adversary (it is estimated locally on the client side). If there is a mismatch between what the adversary estimates as the ratio for client $k$ in FTW-ERM and what the ratio client $k$ applies when outputting gradients, the optimization problem for the gradient inversion will be perturbed.
>
> In particular, even if the adversary knows the model, the loss $\ell$, and obtains $\nabla_{\bf w} F_k({\bf w}_{t})$  in Eq (2.2),
>
> it still needs to know the exact ratio  $\sum_{l=1}^K p_l^{\mathrm{te}}/p_k^{\mathrm{tr}}$ that client $k$ applied when computing $\nabla_{\bf w} F_k({\bf w}_{t})$. Otherwise, the adversary will not be able to formulate the correct optimization problem to recover client $k$’s data. We have clarified this in Section 2.3 of the revision.
>
> ------
>
> -  Why is FTW-ERM better than ERM in Theorem 2?:
>
> The expected risk takes the expectation over the test data, of which the distribution is different from the training data due to the covariate shift.
> In this case, though ERM always obtains the optimal solution for the empirical risk,  it is not optimal to the expected risk. Instead, our FTW-ERM is consistent, ie., the estimator converges to the optimal function that minimizes the expected risk, see Section 2.2 for details.
> That means, FTW-ERM is able to generalize better than ERM.
>
> We also validate this experimentally in Section 5.
>
> ------
>
> -  Elaborate on the the sudden increase in Figure 2:
>
> We have elaborated on the sudden increase and added an additional figure in the appendix (Fig. 3) which plots the eigenvalues. The increase in the eigenvalue ratios occur when the eigenvalues are very small in which case the ratio becomes sensitive to the numerical precision.
>
> ------

---

> ### Author Response · Authors · 2023-04-04
> **Follow-up**
>
> Dear Reviewer,
>
> We would like to check whether you have any questions/concerns? We also welcome any suggestions on improving our paper if the reviewer believes that it will make it easier for the reader to understand.

---

### Review · Reviewer_1yN7 · 2023-03-22

**Summary Of Contributions:**

This paper address the intra- and inter- client covariate shift in the federated learning by correcting the standard empirical risk minimization formulation with additional importance weights. The required density ratio is obtained via a histogram based estimating scheme. The objective after this reweighting procedure is consistent, i.e. it matches the testing error.

**Audience:**

No

**Broader Impact Concerns:**

Since the proposed paradigm requires test data during the training phase, it does not fit the standard machine learning set up and may requires additional attention if such a framework is implemented in real world system.

**Claims And Evidence:**

No

**Requested Changes:**

Please address the points raised in the weakness section, especially points 2-4.

**Strengths And Weaknesses:**

Weakness:

1. The concept of adjusting the training loss based on the train/test ratio seems intuitive, and one would expect it to yield a consistent objective a priori.
2. However, despite being a logical solution for the covariate shift issue, this reweighting approach is not practical because it requires information from the testing dataset during the training phase, which goes against the standard machine learning setup where testing data is not accessible during training.
3. Additionally, even if the testing data were available, the histogram-based density ratio estimation method could still pose challenges since it may not scale well with the dimensionality of the problem.
4. The authors assert that the proposed paradigm provides an equivalent level of privacy protection as the standard ERM formulation. However, this statement lacks rigorous analysis, such as a guarantee based on Differential Privacy. Moreover, the density ratio estimation, which is the fundamental aspect of the proposed scheme, seems to introduce additional privacy risks, but there is no analysis of its potential impact.

---

> ### Author Response · Authors · 2023-03-27
> **Response to Reviewer 1yN7**
>
> We thank the reviewer for their thoughtful comments which we address one by one below. The manuscript is revised, and the changes are shown in blue.
>
> ------
>
> - The concept of adjusting the training loss based on the train/test ratio seems intuitive.
>
> We are thankful for recognizing that this concept is intuitive. We also strive for this, and we are glad that the reviewer agrees with us.  Having said that, our work is the first one to find the unique expression in FTW-ERM where each client requires to estimate a ratio of the sum of test densities over own train density, e.g., $\sum_{l=1}^K p_l^{\mathrm{te}}/p_k^{\mathrm{tr}}$ for client $k$.
>
> ------
>
> - This reweighting approach is not practical because it requires information from the testing dataset during the training phase.
>
> It is true that we use unlabelled test samples to estimate density ratios. Importance-weighted ERM and density ratio estimation are widely used in various machine learning problems such as learning under covariate shift that is observed in real-world applications including emotion recognition and spam filtering, anomaly detection, two-sample testing, causal inference, and classification from positive and unlabelled data  (Zadrozny, 2004; Cheng & Chu, 2004; Sugiyama & Müller, 2005; Keziou & Leoni-Aubin, 2005; Huang et al., 2006; Sugiyama et al., 2007; Kanamori et al., 2009; Kawahara & Sugiyama, 2009; Smola et al., 2009; Hido et al., 2011; Kanamori et al., 2011; Sugiyama et al., 2011; Yamada et al., 2011; Sugiyama et al., 2012; Reddi et al., 2015; Liu & Tao, 2015; Kato et al., 2019; Fang et al., 2020; Uehara et al., 2020; Zhang et al., 2020; Kato & Teshima, 2021).
>
> If the reviewer has any further concerns, we would be happy to discuss them or modify the parts that are not clear to the reader.
>
>
>
> ------
>
> - The histogram-based density ratio estimation method may not scale well with the dimensionality of the problem.
>
> For high-dimensional data, an efficient implementation of histogram-based density ratio estimation using $k$-means clustering is discussed in Appendix G. The running time of Lloyd's algorithm with $M$ clusters is ${\cal O}(n d_{\bf x} M)$ where $n$ is  the total number of samples with dimension $d_{\bf x}$. Even under perfect estimates of $C_k$’s, the number of elementary operations for computing the objective nnBD DRM in (3.2) and its gradients (at each step) is already in $\Omega(n d_{\bf x})$. Even without any importance weighting, the same complexity w.r.t. dimensionality of samples holds for computing the gradients of ERM at each training step. So the histogram-based density ratio estimation does not add any additional complexity w.r.t. dimensionality of samples. We have elaborated on this in Section 3.2 of the revision.
>
>
> ------
>
> - The authors assert that the proposed paradigm provides an equivalent level of privacy protection as the standard ERM formulation.
>
> We respectfully disagree. We **did not claim** that our proposed FTW-ERM in Section 2.2 provides the same level of privacy as ERM. To estimate density ratios for FTW-ERM, if clients can tolerate some level of privacy leakage, clients send **unlabelled** samples from their test distributions. To control privacy leakage to other clients, we propose that the server *randomly shuffles* these unlabelled samples before broadcasting to clients. In Appendix Q, we discuss an alternative method instead of sending original unlabelled samples and discuss its limitations.
>
> To fully eliminate any privacy risks, we introduce FITW-ERM in Section 2.3 (**an alternative method that is different from FTW-ERM**). It does not require any form of data sharing among clients and preserves the *same level of privacy* as those of baseline ERM. Regarding formal differential privacy guarantees for FITW-ERM, the same guarantees as those for ERM (Kairouz et al., 2021 and references therein) hold because **ratios for FITW-ERM are obtained using local data (no data sharing)** and clients share only gradient information without sharing any data. Clients estimate and apply ratios using their own local data, which essentially modifies their local loss function. This modified local loss for FITW-ERM can be directly substituted in any formal differential privacy results for ERM such as those in (Kairouz et al., 2021). In Section 2.3 of the revised version, we have elaborated on this.

---

### Decision · Action_Editors · 2023-05-20

**Recommendation:** Accept with minor revision

**Comment:**

The authors adapted some existing covariate shift techniques to the federated learning setting, and proposed two algorithms that in theory could accommodate shift in marginal training and test distributions. There is some concern over sending unlabeled test samples to the server, which I agree is one of the limitations of this work. Nevertheless, my hope is that the authors have experimentally shown the advantage of their approach in an ideal setting, and future work could possibly address any remaining limitation. For instance, the authors mentioned in Appendix Q that the clients could train a generative model and send generated samples instead. It is also conceivable that some clients may be willing to estimate and send the test density function, or a nested algorithm can be developed to eliminate the need of sending test samples altogether. Thus, I believe overall this work is interesting and relevant to the FL community, considering especially the ubiquity of domain shift in practical problems.

In the final revision, please make sure to incorporate the reviewers' comments (if not already done so in the revision), some of which we recall (and add) below:

-- histogram-based density estimation may not scale well w.r.t. dimension: I believe what the reviewer meant is the sample complexity (not computational complexity that the authors responded to), i.e., how many samples one would need in order to estimate the density (ratio) well. Typically, the dependence on dimension is exponential. It'd be good if the authors add some discussion around this (see also the next comment).

-- error bound in Theorem 1: note that the bound is about the difference between our estimate and the best estimator from the function class H_k (i.e., left-hand side of Eq (4.2)). To exaggerate a bit, if H_k consists of a single function, we get 0 error trivially. So, the usefulness of Theorem 1 really boils down to how likely H_k contains the true density ratio, which is itself a subtle issue. Perhaps it is worthwhile to add that Theorem 1 is mostly about estimation error, and we do not touch approximation error. Or we simply put our belief on H_k containing the true ratio, which is why one might "skip" the curse of dimensionality as alluded to in the previous point.

-- Before Section 2.3, "both FTW-ERM and classical ERM result in the same solution, which is a minimizer of the overall empirical risk": this is true in the limit (i.e., when sample size goes to infinity) but not necessarily true in the finite sample setting (e.g., Eq (2.2))

-- FITW-ERM: note that this reduces to the usual FL setting where the local weights are entirely determined by each client (and hence can be abstracted away, e.g., absorbed into the local loss). It is perhaps better to not claim it as new as what the current text implies.

-- End of page 6: "since the density ratios that are inputs of BD are bounded." This is an assumption, not a fact?

**Audience:**

Yes. Domain shift is a common problem in practice and it is bound to happen in the federated learning setting.

**Claims And Evidence:**

Yes, the authors provided precise mathematical formulation of their problem and offered some theoretical analysis of the underlying generalization, as well as experimental verifications.

---

> ### Author Response · Authors · 2023-05-30
> **Camera Ready Revision**
>
> Dear Action Editor,
>
> We would like to thank you for  handling and carefully reading our paper, and your thoughtful comments. In the camera ready version, we have addressed comments of the Action Editor and anonymous reviewers:
>
> ------
>
> - I believe what the reviewer meant is the sample complexity. Theorem 1 is mostly about estimation error, and we do not touch approximation error ... why one might "skip" the curse of dimensionality.
>
> In Section 4.1. of the camera ready revision, we have clarified "Typically, the required number of samples to accurately estimate density ratios scales  exponentially with the dimensionality of data due to the curse of dimensionality. Theorem 1 bounds estimation error without considering approximation error. The curse of dimensionality is avoided when e.g., the ratio model ${\cal H}_r$ is rich enough and contains the true ratio that is smooth enough."
>
> ------
>
> -  Before Section 2.3 "both FTW-ERM and classical ERM result in the same solution, which is a minimizer of the overall empirical risk": this is true in the limit (i.e., when sample size goes to infinity).
>
>
> Thanks for carefully reading our paper. In Section 2.2 of the camera ready revision, we have have changed the final paragraph to "Under no covariate shift, both FTW-ERM with *true ratios* and …."
>
>
> ------
>
> - FITW-ERM: note that this reduces to the usual FL setting where the local weights are entirely determined by each client.
>
> In Section 2.3 of the camera ready revision, we have added Footnote 5 and clarified "By estimating the ratios locally and absorbing into local losses, FITW-ERM can be viewed as a variant of classical ERM."
>
>
> ------
>
> - End of page 6: "since the density ratios that are inputs of BD are bounded." This is an assumption, not a fact.
>
> In the camera ready version just above Section 3.1, we have clarified that "inputs of BD are bounded following the assumption in Section 2.2."
>
> ------